# REASONING BO: ENHANCING BAYESIAN OPTIMIZATION WITH THE LONG-CONTEXT REASONING POWER OF LLMS

## ABSTRACT

Many real-world scientific and industrial applications require the optimization of expensive black-box functions. Bayesian Optimization (BO) provides an effective framework for such problems. However, traditional BO methods are prone to get trapped in local optima and often lack interpretable insights. To address this issue, this paper designs Reasoning BO, a novel framework that leverages reasoning models to guide the sampling process in BO while incorporating multi-agent systems and knowledge graphs for online knowledge accumulation. We systematically evaluate our approach across 10 diverse tasks encompassing synthetic mathematical functions and complex real-world applications. The framework demonstrates its capability to progressively refine sampling strategies through real-time insights and hypothesis evolution, effectively identifying higher-performing regions of the search space for focused exploration. This process highlights the powerful reasoning and context-learning abilities of LLMs in optimization scenarios. For example, in the Direct Arylation task (a chemical reaction yield optimization problem), our method increased the yield to 60.7%, whereas traditional BO achieved only a 25.2% yield. Furthermore, our investigation reveals that smaller LLMs, after post-training, can attain comparable performance to their larger counterparts.

## 1 INTRODUCTION

Real-world scientific optimization problems often involve complex, expensive-to-evaluate functions across diverse domains, from chemical synthesis (Startt et al., 2024; Liang et al., 2024; Chattoraj et al., 2025) to hyperparameter tuning (Snoek et al., 2012; Feurer et al., 2015; Onorato, 2024). Bayesian Optimization provides an efficient framework for such black-box problems by constructing probabilistic surrogate models and intelligently guiding experiments through acquisition functions (Shahriari et al., 2016). While BO outperforms traditional methods in sample efficiency, it faces key limitations: susceptibility to local optima (Cissé et al., 2024), sensitivity to initial sampling (Huang et al., 2022), and lack of interpretable scientific insights (González-Duque et al., 2024). These challenges motivate our integration of reasoning capabilities with BO.

Recent research has explored integrating LLMs to create hybrid intelligent optimization frameworks that overcome BO's inherent limitations. These frameworks leverage three key LLM capabilities: First, their cross-domain knowledge provides global heuristics to avoid local optima (Cissé et al., 2025). Second, domain priors like chemical reaction rules can be naturally injected through language to reduce sensitivity to initial sampling. Most importantly, LLMs generate scientifically plausible mechanistic hypotheses and iteratively refine them through experiments (Cissé et al., 2024). However, current LLM-BO integration approaches suffer from two critical limitations: (1) potentially hallucinated optimization suggestions from LLMs that may violate domain constraints and compromise safety, and (2) inability to dynamically incorporate experimental discoveries into subsequent optimization cycles. These shortcomings necessitate the development of memory-augmented closed-loop optimization systems.

To address these challenges, we present the Reasoning BO framework (illustrated in Figure 2), which incorporates three mechanisms for trustworthy optimization. First, we embed a Reasoning

Model within the BO loop: (a) users describe experiments in natural language via the Experiment Compass to define the search space; (b) the BO algorithm proposes candidate points, which are then evaluated by the LLMleveraging domain priors, historical data, and knowledge graphsto generate scientific hypotheses and assign confidence scores to each candidate; (c) candidates are filtered based on confidence and consistency with prior results to ensure scientific plausibility. Second, we construct a dynamic knowledge management system that integrates structured domain rules in knowledge graphs and unstructured literature in vector databases, enabling both expert knowledge injection and real-time assimilation of new findings. Third, we fine-tune lightweight models using reinforcement learning on reasoning trajectories to further enhance optimization performance.

For ease of exposition, we summarize our main contributions as

- Incorporation of a reasoning model that leverages LLMs' inference abilities to automatically generate and evolve scientific hypotheses, ensuring scientific plausibility via confidence-based filtering.

- Development of a full-cycle knowledge management system enabling dynamic updating of domain expertise throughout optimization, featuring flexible knowledge architectures with customizable storage formats and open agent interfaces for extensibility.

- Demonstrated superior performance across diverse optimization tasks, particularly in chemical yield prediction and high-dimensional benchmarks. In our Direct Arylation benchmark, Reasoning BO achieves a 23.3% higher final yield (94.39% vs 76.60%) and 44.6% higher initial performance (66.08% vs 21.62%) compared to Vanilla BO, showcasing both superior initialization and continuous optimization capabilities.

## 2 METHODOLOGY

### 2.1 MOTIVATION

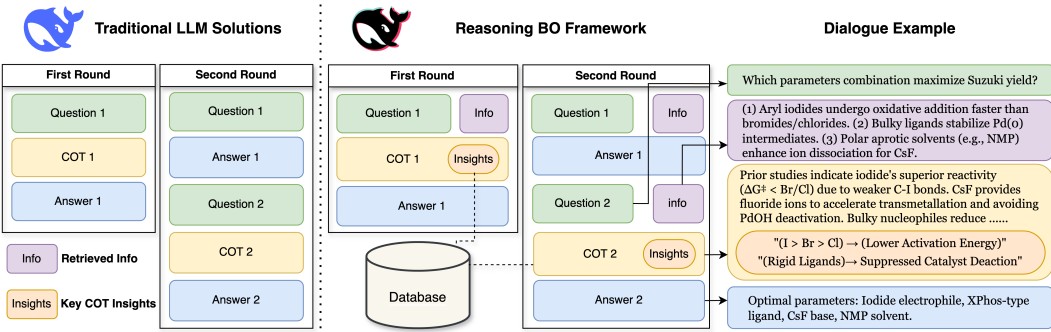

Figure 1: Comparison of classic methods versus our algorithm in utilizing Chain-of-Thought (CoT) data. "Info" represents database information retrieved based on question keywords, "Insights" represents critical insights extracted from Chain-of-Thought data analysis.

The integration of LLMs with traditional BO addresses three fundamental limitations: (1) ineffective utilization of domain-specific prior knowledge, (2) lack of interpretability in mathematical optimization, and (3) weak cross-domain adaptability (González-Duque et al., 2024). Classical BO defines search spaces as rigid parametric constructs that are inadequate for domains like chemistry where constraints are implicit and difficult to formulate as explicit conditions (Hickman et al., 2022). Moreover, as illustrated in Figure 1, traditional approaches do not retain Chain-of-Thought (CoT) data within the context window, discarding valuable reasoning insights from previous rounds. To address this, our framework extracts structured notes from each round of CoT data and stores them, enabling retrieval of relevant information during subsequent queries. Reasoning BO addresses these challenges through three core technical components illustrated in Figure 2: (1) reasoning-enhanced BO that incorporates natural language specifications and domain knowledge, (2) multi-agent knowledge management for dynamic information extraction and storage, and (3) post-training strategies for model enhancement. This framework operates as an end-to-end system where users provide

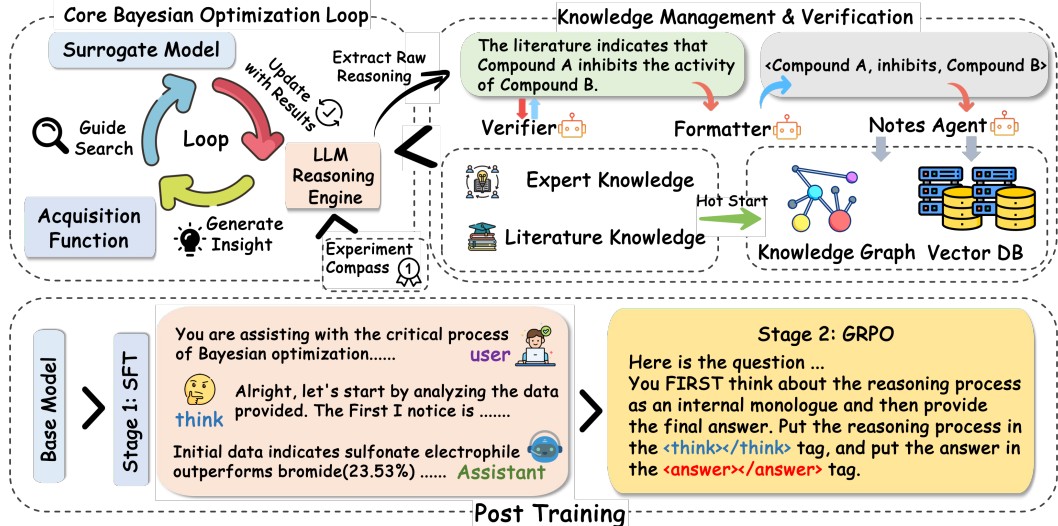

Figure 2: Overview of Reasoning BO's algorithmic framework showcasing the interaction between Bayesian optimization, reasoning models, and knowledge graphs.

experimental objectives through natural language, and the system delivers optimization results with interpretable insights.

## 2.2 REASONING-ENHANCED BAYESIAN OPTIMIZATION

Our Reasoning BO framework integrates reasoning models with Bayesian optimization through a knowledge-guided process. At each iteration $t$, the algorithm maintains: (1) a dataset $D_t = \{(x_i, y_i = f(x_i))\}_{i=1}^{n_t}$ of evaluated points, (2) an insight history $H_t = \{I_0, I_1, \ldots, I_{t-1}\}$ containing structured reasoning outputs from previous iterations, (3) a knowledge base comprising a vector database $V$ and knowledge graph $G$ for domain knowledge retrieval, and (4) an Experiment Compass $C$ that defines the optimization problem in natural language.

The algorithm proceeds in three phases: LLM initialization, iterative optimization with hypothesis generation, and candidate selection, detailed below. Algorithm 1 provides the complete procedure.

**LLM Insight and Hypothesis Generation.** At each optimization iteration $t \geq 1$, the Bayesian acquisition function first proposes a set of $q = 5$ candidate points $C_t = \{x_{\text{BO}}^{(j)}\}_{j=1}^q$ by maximizing the acquisition function over the Gaussian Process surrogate model. The reasoning LLM then receives a prompt containing the current dataset $D_{t-1}$, insight history $H_{t-1}$, Experiment Compass $C$, and candidate set $C_t$.

The LLM is prompt-engineered to generate a structured Insight object $I_t$ that includes: (1) a textual comment analyzing optimization progress and identifying potential stagnation regions, (2) a set of hypotheses $\mathcal{H}_t = \{h_i\}_{i=1}^{|\mathcal{H}_t|}$ where each hypothesis $h_i$ contains a meaningful name, rationale, confidence score $c_i \in [0, 1]$, and associated candidate points, and (3) reasoning data $D_t^{\text{reason}}$ capturing the internal reasoning process.

Formally, the LLM models the conditional distribution $p(x|y > y_{\max}; D_{t-1}, H_{t-1}, C, C_t)$ where $y_{\max} = \max_{(x,y) \in D_{t-1}} y$ is the current best observation. This modeling extends standard LLM regression capabilities (Liu et al., 2024) by incorporating both historical data $D_{t-1}$ and accumulated insights $H_{t-1}$, enabling the model to generate hypotheses that explore regions likely to improve upon $y_{\max}$ while respecting domain constraints specified in $C$.

To enhance reasoning quality, the system retrieves relevant knowledge from the dual-channel knowledge base: keywords extracted from the previous insight $I_{t-1}$ are used to query the knowledge graph $G$ for structured relationships and the vector database $V$ for contextual patterns. This

retrieved knowledge $K_t$ is incorporated into the LLM prompt, effectively modeling $p(x|y > y_{\max}; D_{t-1}, H_{t-1}, C, C_t, K_t)$.

**LLM Initialization.** The initialization phase begins with a user-provided Experiment Compass $C$, a structured natural language template that specifies: the optimization objective, input variable descriptions, target variable to maximize, and domain constraints. The Experiment Compass leverages the LLM's semantic understanding to define the search space $\mathcal{X}$.

Given $C$, the LLM generates an experiment overview $O$ that synthesizes key experimental parameters, domain constraints, and optimization objectives into a structured knowledge representation. Key entities $E$ are extracted from $O$ and stored in both the vector database $V$ and knowledge graph $G$ to initialize the knowledge base.

The reasoning LLM then generates initial reasoning data $D_0^{\text{reason}}$ and an initial Insight $I_0$ based on $O$ and $C$. The Insight $I_0$ contains initial hypotheses $\mathcal{H}_0$ with associated candidate points. High-confidence candidate points $P_0 = \{x_i\}_{i=1}^{n_{\text{init}}} \subseteq I_0$ are selected, where $n_{\text{init}}$ is the number of initial samples. These points are evaluated to form the initial dataset $D_0 = \{(x_i, y_i = f(x_i))\}_{i=1}^{n_{\text{init}}}$.

**Selection Mechanism.** Our sampling strategy avoids direct generation of candidate points by the LLM to prevent test data contamination from its pretraining corpus and potential hallucinations. Instead, we adopt a BO-guided approach where the Bayesian acquisition function proposes candidates, and the LLM serves as a reasoning-based filter.

Specifically, at iteration $t$, the acquisition function proposes $q = 5$ candidates $C_t = \{x_{\text{BO}}^{(j)}\}_{j=1}^q$. The LLM analyzes each candidate in $C_t$ by generating hypotheses and assigning confidence scores, producing Insight $I_t$ with confidence scores $\{c_j\}_{j=1}^q$ for each candidate. The selection mechanism then selects $n = 3$ candidates $P_t = \{x_t^{(i)}\}_{i=1}^n \subseteq C_t$ with the highest confidence scores:

$$P_t = \operatorname*{arg\,max}_{S \subseteq C_t, |S|=n} \sum_{x \in S} c(x, I_t)$$

where $c(x, I_t)$ denotes the confidence score assigned to candidate $x$ in Insight $I_t$. This design leverages the LLM's strengths in knowledge integration and interpretable reasoning while delegating numerical optimization to the Bayesian framework.

The selected candidates $P_t$ are evaluated to obtain results $R_t = \{(x, f(x)) : x \in P_t\}$, which update the dataset $D_t = D_{t-1} \cup R_t$. The insight $I_t$ is added to the history $H_t = H_{t-1} \cup \{I_t\}$, and structured notes $N_t$ extracted from $D_t^{\text{reason}}$ are stored into $V$ and $G$ for future retrieval. **A complete end-to-end case study is provided in Appendix D using the Buchwald-Hartwig benchmark (Shields et al., 2021).**

### 2.3 MULTI AGENTS SYSTEM FOR DYNAMIC KNOWLEDGE PRECIPITATION

Our dynamic knowledge precipitation mechanism uses a multi-agent framework for efficient extraction, verification, and integration of knowledge. Verifier agent extracts variable relationships and domain knowledge from LLM-generated reasoning data, identifying key entities, their interactions, and causal dependencies within the reasoning trajectories. Formatter agent then structures the verified knowledge for storage by parsing reasoning trajectories into verified `<entity, relation, entity>` triples, ensuring consistency and compatibility with both structured storage formats. Finally, Notes Agent integrates this structured knowledge into the vector database and knowledge graph, where triples are incrementally incorporated into the knowledge base. Additionally, the system incorporates expert and prior knowledge to enhance domain-specific comprehensiveness. Verified knowledge is systematically stored in a dual-channel knowledge management system: vector databases store high-dimensional semantic features for efficient similarity searches, while knowledge graphs maintain interpretable networks of conceptual relationships. During optimization, the system triggers knowledge retrieval through keyword queries: first traversing the knowledge graph for structured knowledge paths, then querying the vector database for contextual information.

---

**Algorithm 1:** Reasoning BO

---

**Input:** Experiment Compass $C$, Number of samples $n$ per iteration, Maximum iterations $T$
**Output:** Dataset $D_T$, Insight History $H_T$, Summary and Conclusion

1 Initialize: $D_0 = \{\}, H_0 = \{\}, V = \{\}, G = \{\}$;
2 Generate overview $O$ from $C$ using LLM; Extract entities $E$ from $O$; Store $E$ into $V$ and $G$;
3 Generate initial reasoning data $D_0^{\text{reason}}$ and Insight $I_0$ from $(O, C)$; Select high-confidence
   candidate points $P_0$ from $I_0$;
4 Evaluate $P_0$ to form $D_0 = \{(x, f(x)) : x \in P_0\}$; $H_0 = \{I_0\}$;
5 **for** $t = 1$ **to** $T$ **do**
6     Propose candidates: $C_t = \{x_{\text{BO}}^{(j)}\}_{j=1}^{q=5}$ by maximizing acquisition function over GP
     surrogate;
7     Retrieve knowledge: Extract keywords from $I_{t-1}$; Query $K_t$ from $(V, G)$;
8     Generate reasoning data and Insight: $(D_t^{\text{reason}}, I_t) \sim p(\cdot | D_{t-1}, H_{t-1}, C, C_t, K_t)$ where $I_t$
     contains hypotheses $\mathcal{H}_t$ and confidence scores;
9     Select candidates: $P_t = \arg\max_{S \subseteq C_t, |S|=n=3} \sum_{x \in S} c(x, I_t)$;
10    Evaluate: $R_t = \{(x, f(x)) : x \in P_t\}$; Update $D_t = D_{t-1} \cup R_t$, $H_t = H_{t-1} \cup \{I_t\}$;
11    Extract notes $N_t$ from $D_t^{\text{reason}}$; Store $N_t$ into $V$ and $G$;
12 Generate Summary and Conclusion from $(D_T, H_T)$;

---

## 2.4 POST TRAINING STRATEGIES

To investigate whether additional post-training can enhance reasoning quality and thereby improve Reasoning BO performance, we develop an RL-14B variant through a two-stage training process. This variant is evaluated separately in an ablation study (Figure 8) on two synthetic mathematical benchmarks and is not part of the main experimental pipeline.

**1. Supervised Fine-Tuning (SFT):** We apply Low-Rank Adaptation (LoRA) (Hu et al., 2022) using a dataset constructed from reasoning trajectories. The training dataset is structured in a three-component format: `<user>` contains the optimization prompt and experimental context, `<think>` includes the internal reasoning process generated by reasoning models, and `<assistant>` provides the final structured response with candidate recommendations. The SFT stage serves two purposes: (1) ensuring the model generates correctly formatted JSON responses that can be reliably parsed by the BO loop, and (2) enhancing the model's reasoning capabilities through exposure to structured reasoning patterns. Critically, the SFT dataset contains only format demonstrations and reasoning patternsno task-specific knowledge that could leak information about test benchmarks in our evaluation.

**2. Reinforcement Learning from Human Feedback (RLHF):** Following SFT, we further enhance the RL-14B model using Group Relative Policy Optimization (GRPO) (Shao et al., 2024) on the GSM8K mathematical reasoning dataset. Although GSM8K focuses on mathematical reasoning rather than scientific optimization, training on this dataset activates multi-step inference capabilities that transfer to optimization tasks.

During runtime optimization, the trained model dynamically integrates information from the knowledge graph and vector database (Sections 3.2-3.3) through a dual retrieval process: first querying the knowledge graph for structured relationships, then searching the vector space for contextual patterns. This knowledge integration occurs at inference time and does not involve further model training.

## 3 EXPERIMENTS

### 3.1 EXPERIMENTAL SETUP

**Real-world BO benchmarks**. We evaluate three real-world BO benchmarks in our main experiments: (1) **Suzuki-Miyaura Reaction:** Maximizing product yield by optimizing electrophile-nucleophile pairs, ligand, base, and solvent under catalyst compatibility constraints (Perera et al., 2018); (2) **Direct Arylation Reaction:** Maximizing isolated yield through optimal selection of lig-

and, base, solvent, concentration and temperature (McNally et al., 2011); (3) **Lunar Lander:** Optimizing control policies for spacecraft landing with mixed continuous-discrete variables, balancing fuel efficiency and landing precision under physical constraints (Maus et al., 2024).

**Synthetic Function Benchmarks**. Following (Cissé et al., 2025; Mahammadli, 2024; Hvarfner et al., 2023), we use synthetic functions to systematically evaluate optimization algorithms' ability to handle different landscape characteristics: (1) **Ackley:** A d-dimensional multimodal function featuring nearly flat outer regions and a steep central valley. (2) **Rosenbrock:** A non-convex d-dimensional function with narrow parabolic valley structure; (3) **Hartmann:** A highly multimodal function with numerous local minima separated by high energy barriers, exhibiting strong parameter coupling effects. To ensure that LLMs cannot identify the optimal solutions by recognizing the benchmark names, we replaced the function names in the experiment compass with generic terms such as "mathematical function". See Appendix E for complete benchmark specifications.

**Implementation Details of Reasoning BO**. We implement our Reasoning BO algorithm using QWQ-Plus (Team, 2025), a medium-sized reasoning model from the Qwen series that achieves competitive performance against state-of-the-art reasoning models like DeepSeek-R1. All main experimental results, including those in Figure 3, are obtained using QWQ-Plus without any task-specific fine-tuning or reinforcement learning. For simplicity, the main pipeline does not incorporate the knowledge graph and vector database retrieval mechanisms, relying solely on the model's pre-trained reasoning capabilities. For less complex tasks, such as note extraction and verification, we employ the computationally efficient Qwen2.5-Instruct-14B (Yang et al., 2025) to reduce overhead. As detailed in Appendix A.1, the capabilities of reasoning models can vary significantly. For BO acquisition function, we adopt qLogEI (Ament et al., 2023) (see Appendix Section F.1 for details).

**Baselines**. (1) **Random Search**: Unbiased exploration baseline. (2) **Vanilla BO**: Example of vanilla BO using qLogEI acquisition function (Ament et al., 2023). (3) **Analytic EI**: BO algorithm using Analytic LogEI acquisition function (Frazier et al., 2011; 2009). (4) **CMA-ES**: Covariance Matrix Adaptation Evolution Strategy, a derivative-free evolutionary optimization method (Hansen, 2016). Detailed parameter configurations are provided in Appendix B.2.

**Experimental Protocol**. Optimization performance is evaluated based on the best observed objective value. Following the protocol established in BORA (Cissé et al., 2025), we initialize each experiment and evaluate 3 candidate points in each subsequent round. To balance real-world budget constraints, the maximum number of experiments is set to 30. Each experiment is repeated 10 times with fixed random seeds, and the average results are reported. Unlike conventional BO plots that display strictly monotonic best-so-far curves, our optimization trajectory plots report the best objective value among the three candidates evaluated within each iteration, rather than the cumulative best-so-far across all past iterations. This per-iteration visualization is intentionally chosen to reveal the dynamic decision-making patterns of LLM-augmented BO, including how hypotheses evolve and how the exploration–exploitation balance differs from classical BO. While this design may exhibit mild fluctuations, these oscillations do not affect convergence or final optimization performance. The source code is publicly available at `https://anonymous.4open.science/r/Reasoning-BO/`.

**Evaluation Metrics.** We employ multiple quantitative metrics to comprehensively assess optimization performance, as summarized in Table 1. The Coefficient of Variation (CV) measures relative dispersion, while standard deviation (Std) quantifies absolute variability. Log AUC evaluates cumulative performance by integrating the optimization trajectory. Log Regret captures the cumulative deviation from optimal values. IMP@$\alpha$ assesses initial performance over the first $\alpha$ iterations, and CVaR@n evaluates tail-end performance at confidence level n. These metrics collectively provide insights into optimization stability, convergence speed, and robustness across different phases of the search process.

## 3.2 RESULTS

**Empirical Insights**

(1) **Performance:** As shown in Figure 3, Reasoning BO robustly and efficiently outperforms baselines across diverse benchmarks, including both real-world chemical optimization tasks and syn-

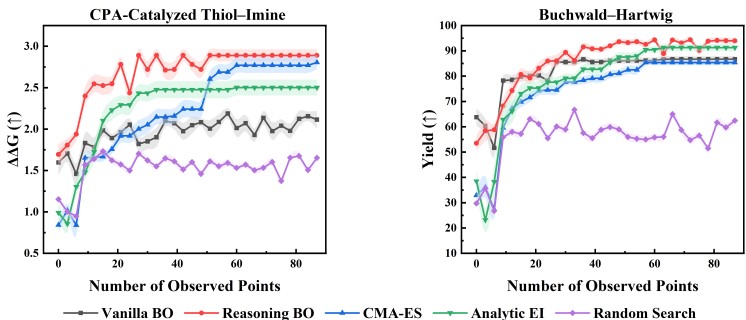

Figure 3: Reasoning BO vs Baselines on six experiments. The y-axis indicates the best objective value among candidates evaluated within each iteration (not cumulative best-so-far). Solid lines and shaded areas represent mean ± standard error.

| Benchmark | Method | CV ↓ | Std ↓ | Log Regret ↓ | Log AUC ↑ | CVaR@0.1 ↑ | CVaR@0.3 ↑ | CVaR@0.5 ↑ | IMP@1 ↑ | IMP@3 ↑ | IMP@5 ↑ |
|---|---|---|---|---|---|---|---|---|---|---|---|
| Direct Arylation | Random Search | 0.67 ± 0.09 | 26.49 ± 6.51 | 7.14 ± 0.19 | 7.01 ± 0.34 | 2.67 ± 2.67 | 9.53 ± 4.83 | 17.22 ± 5.36 | 28.74 ± 37.14 | 16.93 ± 16.76 | 27.57 ± 14.26 |
| | Vanilla BO | 0.35 ± 0.11 | 22.75 ± 5.81 | 6.64 ± 0.46 | 7.53 ± 0.50 | 19.76 ± 10.34 | 40.63 ± 16.47 | 52.67 ± 17.81 | 43.62 ± 32.68 | 45.17 ± 19.42 | 55.91 ± 19.95 |
| | CMA-ES | 0.28 ± 0.17 | 15.29 ± 8.02 | 6.41 ± 0.91 | 7.41 ± 0.50 | 32.73 ± 18.32 | 46.34 ± 20.68 | 52.08 ± 21.67 | 25.20 ± 21.16 | 32.73 ± 18.32 | 39.32 ± 19.02 |
| | Analytic EI | **0.16 ± 0.20** | **8.69 ± 9.10** | 6.11 ± 0.94 | 7.50 ± 0.53 | **53.35 ± 28.30** | 58.03 ± 25.11 | 62.40 ± 23.73 | 51.23 ± 30.65 | 53.35 ± 28.30 | 54.22 ± 28.08 |
| | Reasoning BO | 0.24 ± 0.18 | 13.77 ± 6.88 | **6.03 ± 1.13** | **7.56 ± 0.57** | 44.18 ± 31.01 | **60.76 ± 29.64** | **67.71 ± 28.08** | **60.07 ± 34.97** | **66.64 ± 31.58** | **71.22 ± 28.63** |
| Suzuki | Random Search | 0.34 ± 0.11 | 21.45 ± 5.16 | 6.65 ± 0.42 | 7.50 ± 0.49 | 22.98 ± 9.63 | 39.32 ± 14.44 | 50.27 ± 15.91 | 52.43 ± 33.42 | 48.43 ± 19.97 | 59.43 ± 21.34 |
| | Vanilla BO | 0.20 ± 0.15 | 12.21 ± 4.04 | 5.42 ± 2.13 | 7.59 ± 0.57 | 49.97 ± 23.49 | 67.26 ± 25.94 | 71.02 ± 26.09 | 62.32 ± 36.41 | 49.97 ± 23.49 | 60.33 ± 24.94 |
| | CMA-ES | 0.23 ± 0.14 | 15.41 ± 5.24 | 6.00 ± 0.84 | 7.67 ± 0.54 | 43.79 ± 23.87 | 66.20 ± 23.32 | 73.13 ± 23.41 | 38.16 ± 31.74 | 43.79 ± 23.87 | 56.52 ± 23.16 |
| | Analytic EI | 0.29 ± 0.13 | 19.84 ± 6.46 | 5.93 ± 0.76 | 7.63 ± 0.53 | 33.98 ± 18.21 | 53.30 ± 21.64 | 65.35 ± 21.36 | 41.91 ± 34.20 | 40.94 ± 18.67 | 42.57 ± 20.22 |
| | Reasoning BO | **0.14 ± 0.17** | **7.71 ± 4.96** | **4.91 ± 1.88** | **7.69 ± 0.58** | **64.76 ± 27.47** | **74.66 ± 26.00** | **78.02 ± 25.49** | **67.85 ± 33.57** | **69.52 ± 30.17** | **73.39 ± 28.14** |
| CPA-Catalyzed Thiol-Imine | Random Search | 0.45 ± 0.07 | 1.40 ± 2.46 | 5.54 ± 1.45 | 4.00 ± 0.68 | 0.51 ± 0.27 | 1.15 ± 0.96 | 1.60 ± 1.80 | 0.86 ± 0.66 | 0.85 ± 0.39 | 1.27 ± 0.37 |
| | Vanilla BO | 0.27 ± 0.15 | **1.21 ± 2.52** | 5.58 ± 1.23 | 4.21 ± 0.64 | 1.06 ± 0.27 | 1.72 ± 0.84 | 2.16 ± 1.66 | 1.45 ± 1.03 | 1.53 ± 0.40 | 1.70 ± 0.28 |
| | CMA-ES | 0.33 ± 0.12 | 1.43 ± 2.46 | **5.46 ± 1.52** | 4.32 ± 0.60 | 0.91 ± 0.19 | 1.70 ± 0.80 | 2.21 ± 1.62 | 0.77 ± 0.61 | 0.91 ± 0.19 | 1.27 ± 0.30 |
| | Analytic EI | 0.27 ± 0.15 | 1.29 ± 2.51 | 5.49 ± 1.40 | 4.33 ± 0.62 | 1.02 ± 0.48 | 1.90 ± 0.83 | 2.46 ± 1.58 | 0.90 ± 0.66 | 1.05 ± 0.47 | 1.34 ± 0.50 |
| | Reasoning BO | **0.21 ± 0.15** | 1.22 ± 2.52 | 5.52 ± 1.21 | **4.50 ± 0.53** | **1.57 ± 0.43** | **2.35 ± 0.67** | **2.88 ± 1.41** | **1.54 ± 0.92** | **1.74 ± 0.57** | **2.07 ± 0.43** |

Table 1: Quantitative performance comparison of optimization methods across three benchmarks. Bold values indicate the best quantitative result in each column.

thetic mathematical functions. This highlights the algorithm's effectiveness in navigating intricate search spaces.

(2) **Initialization Strategy:** Reasoning BO employs an advanced initialization strategy that leverages expert priors and domain-specific knowledge to systematically select critical starting points. For example, it initializes near the known optimum of the Levy function ([1, ..., 1]) to promote rapid convergence and identifies favorable starting points in the Rosenbrock function's narrow valley to accelerate progress. For functions with asymmetric bounds like Ackley, its adaptive learning mechanism effectively mitigates the impact of suboptimal initial points.

## 3.3 EXTENDED EXPERIMENTAL ANALYSIS

Figure 4: Chemical reaction benchmarks: CPA-Catalyzed ThiolImine and Buchwald-Hartwig amination.

This subsection analyzes our model's performance on additional chemical datasets and high-dimensional benchmarks. Figure 4 presents optimization trajectories across two chemical reactions. In the CPA-Catalyzed ThiolImine reaction, our framework achieves 0.20 kcal/mol improvement during initialization and 0.67 kcal/mol improvement during convergence compared to vanilla BO. For Buchwald-Hartwig amination, we achieve 10.37% yield improvement (95.06% vs. 84.68%).

Our approach also demonstrates notable improvements on high-dimensional benchmarks (Figure 5). On Lunar Lander (12D), we achieve 71.02 higher initial reward compared to vanilla BO. For Ackley

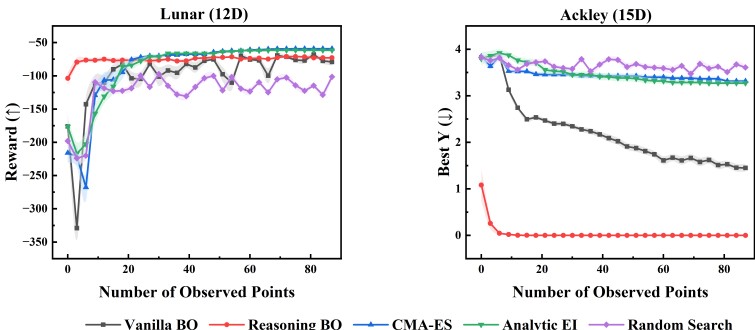

Figure 5: High-dimensional benchmarks: Lunar Lander (12D) and Ackley (15D).

(15D), our algorithm shows -2.34 difference during initialization and -1.46 final difference at 90 observations.

## 3.4 ABLATION STUDIES

We conduct ablation studies to investigate the impact of different components in our framework. These studies include initialization vs optimization analysis, structured notes analysis, and reinforcement learning model comparisons.

**Initialization vs Optimization Analysis.** To systematically analyze the contributions of our algorithm's initialization strategy and continuous optimization capability, we conduct comprehensive ablation studies comparing different initialization-optimization combinations. Figure 6 presents the comparative results across multiple benchmarks. *Empirical Insights:* The analysis reveals that Reasoning BO initialization consistently identifies better starting points compared to Sobol sampling in Vanilla BO, achieving 44.6% higher initial performance in Direct Arylation (66.08% vs 21.62%). The RBO optimization phase consistently outperforms VBO regardless of initialization method, validating our algorithm's superior optimization capability. The RBO→VBO configuration demonstrates that good initialization alone can help Vanilla BO surpass its full version, though with less efficiency than full RBO.

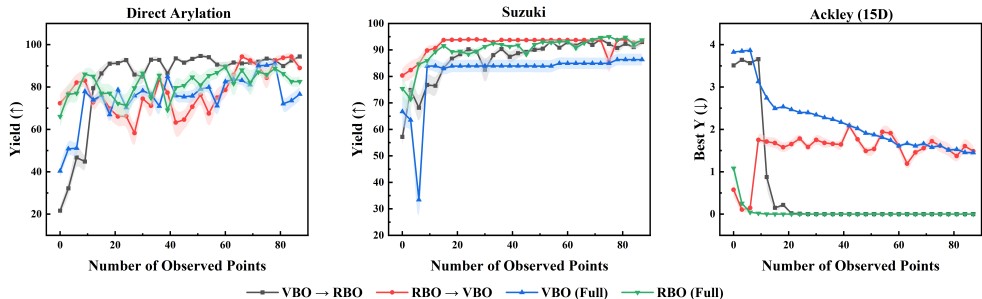

Figure 6: Ablation study comparing different initialization and optimization strategies. Four configurations are evaluated: full Reasoning BO (RBO), full Vanilla BO (VBO), RBO initialization followed by VBO optimization (RBO→VBO), and VBO initialization followed by RBO optimization (VBO→RBO).

**Structured Notes Analysis.** We investigate the impact of incorporating structured notes in our framework. Here we introduce structured notes as an ablation study to evaluate the contribution of knowledge-guided reasoning. The results are visualized in Figure 7. *Empirical Insights:* The ablation study reveals domain-dependent effects of incorporating notes. Chemical applications show consistent improvements: Suzuki coupling achieves 2% better initial performance, while Direct Arylation demonstrates significantly better optimization outcomes during mid-to-late stages. For the mathematical Hartmann function, where prior knowledge is limited, notes constrain exploration and slow optimization progress. These findings suggest that structured notes provide substantial

benefits in knowledge-rich domains but may hinder performance in purely synthetic optimization scenarios.

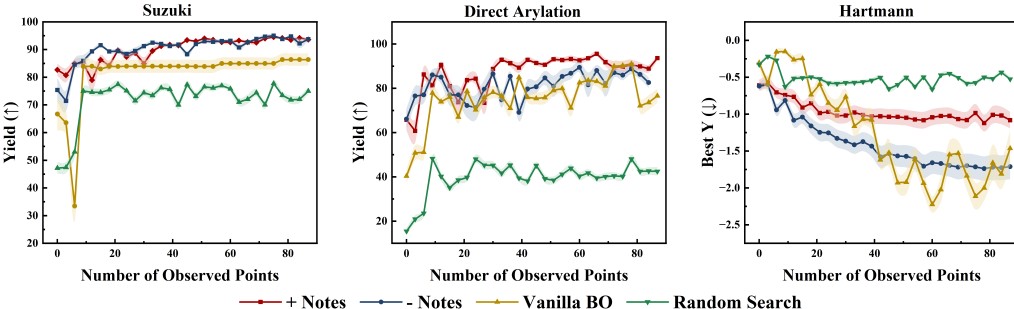

Figure 7: Ablation study comparing Reasoning BO with and without structured notes on three optimization benchmarks: (a) Suzuki coupling, (b) Direct Arylation reactions, and (c) 6D Hartmann function, alongside random search and vanilla BO baselines.

**Reinforcement Learning Enhanced Models.** We compare the capabilities of our RL-enhanced models against baseline. Figure 8 shows the performance comparison between our RL-14b model, Instruct-14b, and Distilled-14b variants, along with larger models for scalability assessment. ***Empirical Insights:*** The RL-enhanced models demonstrate significant improvements over baseline approaches, with the RL-14b model showing superior performance across multiple benchmarks. The comparison with larger models validates the scalability of our approach, while the distilled variants maintain competitive performance with reduced computational requirements.

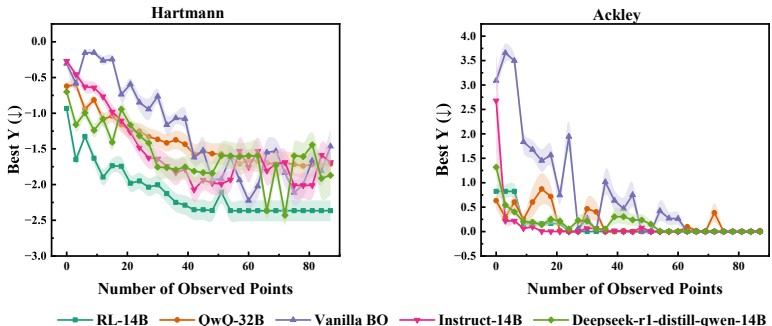

Figure 8: Performance comparison of RL-enhanced models versus baseline approaches with similar parameter counts.

## 4 CONCLUSIONS

This paper has presented Reasoning BO, a novel framework that systematically integrates knowledge graphs, multi-agent systems, and reinforcement learning into the Bayesian optimization paradigm, establishing an intelligent optimization system for scientific discovery with superior cost-effectiveness. The framework innovatively combines the reasoning capabilities of LLMs with the knowledge storage properties of knowledge graphs and vector databases, creating a dynamic knowledge fusion mechanism that effectively injects domain knowledge into the optimization process. The multi-agent architecture ensures flexibility and adaptability, providing a unified and customizable solution for user-friendly interactions. Future research directions include developing global attention mechanisms to overcome context length constraints, creating novel knowledge distillation paradigms to enhance knowledge extraction efficiency, and exploring synergistic optimization paths between reinforcement learning algorithms and generated data, further unlocking the reasoning potential of LLMs in Bayesian optimization.

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

**Appendix**

This appendix provides extended technical and experimental details that support the findings in the main paper. It is organized as follows:

1. **Appendix A: Extended Results** presents additional experimental analyses, including performance comparisons across different API models (DeepSeek, qwq-plus, and deepseek-distilled-qwen-7b) and comprehensive error bar analyses for chemical and high-dimensional benchmarks.

2. **Appendix B: Technical Implementation Details** provides technical specifications of the Low-Rank Adaptation (LoRA) implementation used in our post-training strategies, along with detailed parameter configurations for baseline methods including CMA-ES.

3. **Appendix C: Related Work** discusses the theoretical foundations and prior research in Bayesian Optimization, Language-Guided Bayesian Optimization, and Reinforcement Learning-Enhanced Reasoning in LLMs, positioning our contributions within the broader research landscape.

4. **Appendix D: Case Study** presents a complete end-to-end walkthrough of the Buchwald-Hartwig amination optimization task, including Experiment Compass input, overview generation, candidate initialization, optimization loop iterations, and final summary and conclusion.

5. **Appendix E: Benchmarks** provides comprehensive specifications for all optimization benchmarks used in our evaluation, including detailed descriptions of chemical reaction tasks (Suzuki, Direct Arylation, Buchwald-Hartwig, CPA-Catalyzed Thiol-Imine), control problems (Lunar Lander), and synthetic mathematical functions (Levy, Hartmann, Ackley, Rosenbrock).

6. **Appendix F: Baselines** details the implementation and theoretical foundations of baseline methods, including the LogEI family (Analytic LogEI and Monte Carlo Parallel LogEI) and CMA-ES, explaining their numerical stability properties and optimization mechanisms.

7. **Appendix G: System Messages and Prompts Templates** documents all prompt templates and system messages used throughout our framework, including Experiment Compass objects, Overview generation, Insights objects, optimization prompts, summary and conclusion templates, and notes extraction schemas.

8. **Appendix H: Usage of Large Language Models in This Manuscript** provides a disclosure statement regarding the use of LLMs for editorial purposes in manuscript preparation.

9. **Appendix I: Limitations and Future Work** discusses current limitations of our framework, including context window constraints, instruction-following capability dependencies, and potential extensions to multi-objective optimization scenarios.

## A    EXTENDED RESULTS

### A.1    IMPACT OF DIFFERENT API MODELS

We evaluate the performance of three API models (DeepSeek, qwq-plus, and deepseek-distilled-qwen-7b) across three distinct datasets to assess their reasoning capabilities within our framework. Figure 9 presents the comparative results on Suzuki coupling, Direct Arylation, and 6D Hartmann function benchmarks.

***Empirical Insights:*** On chemical datasets, both qwq-plus and DeepSeek substantially outperform the baseline BO, achieving comparable results that highlight their strong reasoning capabilities. For the Hartmann function benchmark, DeepSeek initially leads but shows limited exploration in later stages, while qwq-plus maintains more consistent performance. The distilled model demonstrates slight advantages over the baseline, suggesting that model capacity directly influences optimization effectiveness.

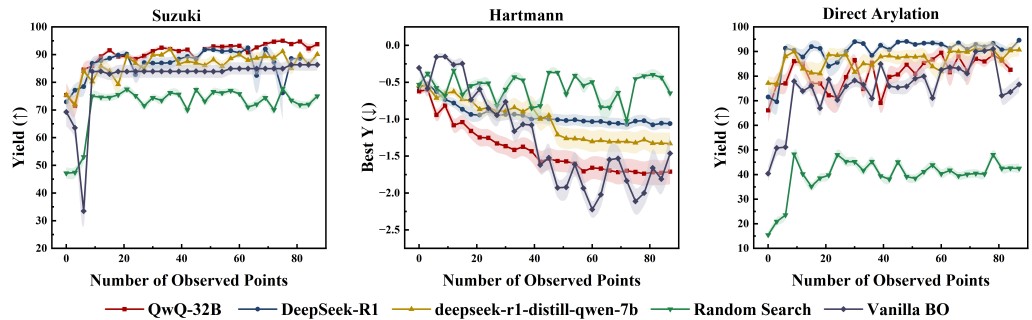

Figure 9: Performance comparison of different API models on three optimization tasks.

### A.2    ERROR BAR ANALYSIS

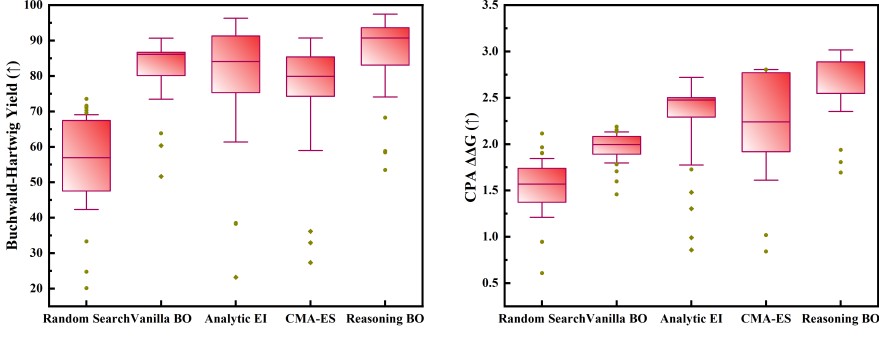

Figure 10: Error bar analysis of optimization performance on two chemical benchmarks: CPA-Catalyzed Thiol-Imine and Buchwald-Hartwig amination. The figure depicts the mean and standard deviation of the optimization results across multiple trials, highlighting the consistency and robustness of our framework compared to baseline methods.

For comprehensive error bar analyses of chemical and high-dimensional benchmarks, refer to Figures 10 and 11 respectively.

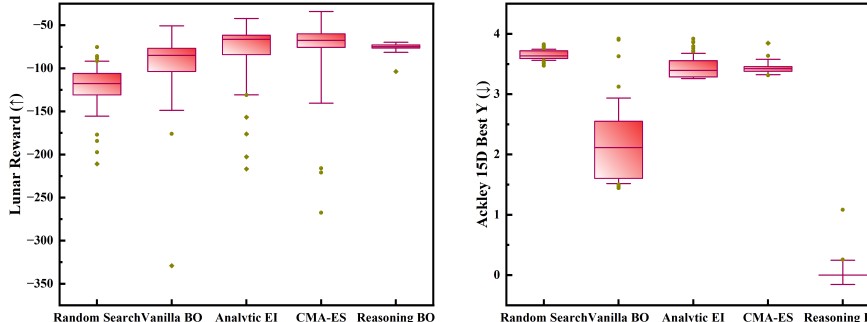

Figure 11: Error bar analysis of optimization performance on high-dimensional benchmarks: Lunar Lander (12D) and Ackley (15D). The figure illustrates the mean and standard deviation across multiple trials, demonstrating the stability and effectiveness of our framework in handling high-dimensional problems.

## B    TECHNICAL IMPLEMENTATION DETAILS

### B.1    LOW-RANK ADAPTATION (LoRA) IMPLEMENTATION

LoRA enhances fine-tuning efficiency by freezing the pre-trained model weights to prevent catastrophic forgetting, while introducing trainable rank decomposition matrices $B$ and $A$ into the multi-head self-attention layers of the transformer architecture. These matrices are sensitive to task-specific adaptations and allow the model to learn domain-specific terminology and relationships effectively. During forward propagation, the input $x$ is processed as follows:

$$\Delta W = BA, \quad h = W_0 x + \Delta W x = W_0 x + BA x, \tag{1}$$

where $W_0$ represents the original weight matrix of the pre-trained model, and $\Delta W$ denotes the incremental adjustment introduced by LoRA. This approach ensures that the model's general linguistic capabilities remain intact while adapting to the target domain. The rank decomposition significantly reduces the number of trainable parameters while maintaining effective adaptation to domain-specific optimization tasks.

### B.2    BASELINE CONFIGURATION DETAILS

This section provides detailed parameter configurations for CMA-ES baseline method used in our experiments.

**CMA-ES Parameters:** The Covariance Matrix Adaptation Evolution Strategy is configured with the following parameters based on our implementation: population size (popsize) = 5, maximum iterations (maxiter) = 6, initial step size $\sigma_0 = 0.1$, with variable bounds set to [0, 1] for normalized search space. The algorithm is initialized with 3 Sobol-generated points and uses Upper Confidence Bound (UCB) acquisition function with $\beta = 0.1$ for Gaussian Process-guided exploration. Each experiment is repeated 10 times for statistical robustness, with 30 total evaluation trials per run.

## C RELATED WORK

### C.1 BAYESIAN OPTIMIZATION

Bayesian Optimization (BO) is a widely used framework for black-box function optimization, which relies on probabilistic surrogate models that are iteratively refined using historical observations (Rasmussen, 2004). Typically, BO consists of two main components: a surrogate modelmost commonly a Gaussian Process (GP) that models the objective as $f(\mathbf{x}) \sim \mathcal{N}(\mu(\mathbf{x}), \sigma^2(\mathbf{x}))$ (Rasmussen, 2004)and an acquisition function that guides the search by balancing exploration and exploitation. Among various acquisition strategies, Expected Improvement (EI) is particularly popular due to its solid theoretical foundation (Mockus et al., 1978; Zhan & Xing, 2020). Despite its theoretical sample efficiency, BO faces notable challenges in practical, high-cost evaluation scenarios. First, the computational complexity of surrogate model training increases rapidly with the number of evaluations (Lan et al., 2022). Second, standard acquisition functions often struggle to dynamically adjust the exploration-exploitation trade-off, which can result in convergence to sub-optimal local optima (Eriksson et al., 2019). To address these limitations, our approach leverages the global heuristic perspective and reasoning capabilities of large language models (LLMs), which can partially mitigate the bottlenecks and enhance the overall efficiency.

### C.2 LANGUAGE-GUIDED BAYESIAN OPTIMIZATION

By leveraging LLMs' contextual understanding(Wei et al., 2022; Dong et al., 2024) and few-shot generation capabilities (Brown et al., 2020), recent advances demonstrate their potential to effectively address black-box optimization challenges when integrated with BO (Liu et al., 2024). Current mainstream work in this area can be primarily categorized into two types: (1) LLM-assisted experimental design, which incorporates reasoning capabilities into the development of candidates sampling strategies, and (2) knowledge transfer techniques, which leverage domain-specific insights or user belief to guide the optimization process. In the first category, SLLMBO (Mahammadli, 2024) employs a hybrid sampling strategy combining LLMs with Tree-structured Parzen Estimator (TPE) (Shakya et al., 2024) for efficient hyperparameter optimization. HypBO (Cissé et al., 2024) and BORA (Cissé et al., 2025) further accelerate Bayesian searches by leveraging expert hypotheses to rapidly identify promising regions. In the second category, MI-SMBO (Feurer et al., 2015) initializes Bayesian Optimization using configurations suggested by meta-learning, while BOChemian (Ranković & Schwaller, 2023) transforms textual chemical procedures into an informative feature space suitable for optimization.

While current approaches have made significant progress, they still encounter challenges such as potential hallucinations in LLM-generated suggestions that could affect optimization accuracy (Xu et al., 2024b; Ji et al., 2023), over-reliance on manual prompt templates limiting adaptability, and insufficient mechanisms for persisting knowledge. Additionally, these methods may face issues related to data leakage during benchmark testing. To address these issues, our framework integrates an evolvable domain knowledge graph for continuous learning (Zhao et al.,

2023), a multi-agent system to enhance adaptability, and RAG mechanisms to improve knowledge reuse efficiency and correct factual errors(Shuster et al., 2021).

### C.3 REINFORCEMENT LEARNING-ENHANCED REASONING IN LLMS

Recent advances in reinforcement learning have significantly enhanced the reasoning capabilities of large language models through policy optimization techniques. Policy gradient methods (Xu et al., 2024a; Patil et al., 2021) and Proximal Policy Optimization (PPO) (Schulman et al., 2017) form the theoretical foundation for training LLMs to generate high-quality reasoning trajectories. Building upon these approaches, Group Relative Policy Optimization (GRPO) (Shao et al., 2024) extends PPO through group relative advantage estimation, enabling more stable policy updates in complex reasoning scenarios. The application of RL to reasoning has shown promise across diverse domains, including rule-based reasoning (Xie et al., 2025; Pan et al., 2025; Zeng et al., 2025), visual reasoning with out-of-distribution generalization (Chen et al., 2025), and medical reasoning through verified question answering (Zhang et al., 2025). These works demonstrate that Reinforcement Learning from Human Feedback (RLHF) (Ouyang et al., 2022) can effectively optimize models to produce more reliable and structured reasoning outputs, providing the foundation for integrating reasoning capabilities into optimization frameworks.

## D CASE STUDY

### D.1 1. EXPERIMENT COMPASS INPUT

The algorithm begins with a user-prepared Experiment Compass, which serves as the structured input containing all necessary experimental parameters and constraints. This JSON-formatted document specifies the reaction components, optimization objectives, and parameter bounds for the Buchwald-Hartwig amination case study. The complete Experiment Compass configuration used in our evaluation is detailed below.

---

**Experiment Compass Configuration for Buchwald-Hartwig Amination Optimization**

- **name**: Buchwald-Hartwig Coupling Reaction Optimization

- **application_context**: Organic Chemistry / Reaction Condition Optimization

- **description**: This experiment aims to optimize the reaction conditions for the Buchwald-Hartwig amination, a key C-N cross-coupling reaction used in the synthesis of pharmaceuticals and functional materials. The optimization focuses on selecting optimal ligands, additives, bases, and aryl halides to maximize reaction yield. High-throughput experimentation and AI-driven decision-making are employed to explore a large combinatorial chemical space.

- **constraint**: Only one choice per category (Ligand, Additive, Base, Aryl halide) can be selected per reaction run. All inputs are categorical.

---

- **parameter_definitions**:
  - * **display_name**: `Ligand`
    * **description**: Phosphine-based or bulky ligands that influence catalyst reactivity and selectivity
    * **data_type**: categorical
    * **bounds**:
      · `CC(C)C(C=C(C(C)C)C=C1C(C)C)=C1C2=`
        `C(P(C(C)(C)C)C(C)(C)C)C=CC=C2`
      · `CC(C)C(C=C(C(C)C)C=C1C(C)C)=C1C2=`
        `C(P(C3CCCC3)C4CCCC4)C=CC=C2`
      · ...
  - * **display_name**: `Additive`
    * **description**: Nitrogen or oxygen heterocycles influencing reactivity and solubility
    * **data_type**: categorical
    * **bounds**:
      · `C1(C2=CC=CC=C2)=CC=NO1`
      · `C1(C2=CC=CC=C2)=CON=C1`
      · `O=C(OC)C1=NOC(C2=CC=CS2)=C1`
      · ...
  - * **display_name**: `Base`
    * **description**: Nitrogen-based or phosphine-derived bases used for deprotonation and catalytic cycle support
    * **data_type**: categorical
    * **bounds**:
      · `CC(C)(C)/N=C(N(C)C)/N(C)C`
      · `CN(C)P(N(C)C)(N(C)C)=NP(N(C)C)(N(C)C)=NCC`
      · `CN1CCCN2C1=NCCC2`
      · ...
- **target**:
  - **name**: Yield
  - **description**: Isolated yield percentage of the desired amine product from the B-H coupling reaction
  - **direction**: maximize

## D.2   2. OVERVIEW GENERATION

The framework processes the Experiment Compass through a large language model (LLM) to generate a comprehensive overview of the Buchwald-Hartwig amination reaction. This overview synthesizes key experimental parameters, chemical constraints, and optimization objectives into a structured knowledge representation that guides subsequent algorithmic decisions. The generated overview is presented below.

**Automatically generated overview for Buchwald-Hartwig amination optimization**

1. **Experimental Goal and Significance**
   The experiment aims to optimize reaction conditions for the Buchwald-Hartwig amination, a critical C-N cross-coupling reaction widely used in pharmaceutical and material synthesis. This optimization seeks to maximize the isolated yield of the desired amine product by systematically exploring ligands, additives, bases, and aryl halides. Successful optimization will enhance reaction efficiency, reduce experimental trial-and-error, and provide a scalable framework for designing high-yielding protocols, which is pivotal for accelerating drug discovery and industrial applications.

2. **Parameter Space**
   The parameter space includes four categorical variables:

   - **Ligand**: Four phosphine-based or bulky ligands (e.g., triarylphosphines with varying steric/electronic properties).
   - **Additive**: Twenty nitrogen or oxygen heterocycles (e.g., pyridines, oxazoles) that modulate reactivity, solubility, or catalyst stabilization.
   - **Base**: Three nitrogen/phosphine-derived bases (e.g., alkylamines, phosphazenes) for deprotonation and catalytic cycle support.
   - **Aryl halide**: Fifteen substrates with varied halogens (Br, Cl, I) and substituents (e.g., electron-withdrawing/-donating groups).

   Each parameter's bounds define discrete choices, reflecting structural diversity to probe reactivity trends.

3. **Constraints**
   Only one option per category is selected per experiment, ensuring orthogonal exploration of the combinatorial space. This constraint avoids overparameterization and maintains interpretability of results.

4. **Bayesian Optimization (BO) Strategy**
   BO will iteratively balance exploration and exploitation using a probabilistic surrogate model (e.g., Gaussian process) to predict yields. Initial trials will randomly sample the space to build prior distributions. Subsequent iterations will use acquisition functions (e.g., expected improvement) to prioritize high-yield regions while exploring under-sampled areas. Key features include:

   - **Surrogate modeling**: Captures interactions between ligand electronics, additive solubility, base strength, and substrate reactivity.
   - **Active learning**: Sequentially selects parameter combinations likely to improve yield, leveraging gradients in the response surface.
   - **Parallelization**: High-throughput experimentation enables rapid iteration across multiple candidates.

5. **Handling Plateaus**

If BO plateaus, domain knowledge will guide new combinations:

- **Ligand-additive**: Pair electron-rich ligands with electron-deficient additives to stabilize intermediates.
- **Base-substrate**: Match bulky bases with sterically hindered substrates to mitigate side reactions.
- **Halogen effects**: Test less common halides (e.g., I vs. Br) paired with ligands optimized for oxidative addition.

These suggestions exploit mechanistic insights (e.g., catalyst activation, steric control) to escape local optima.

6. **Expected Outcomes and Impact**

A high-yield protocol will reduce waste and time in pharmaceutical synthesis. Identified trends (e.g., optimal ligand-additive pairs for specific substrates) could generalize to other cross-coupling systems. The framework's scalability may enable AI-driven design of novel catalyst systems, advancing sustainable and efficient chemical manufacturing.

This structured approach bridges computational efficiency with chemical intuition, offering a robust path toward reaction optimization.

## D.3 3. CANDIDATE INITIALIZATION

The initialization phase leverages the reasoning capabilities of LLMs to generate promising candidate points within the constrained parameter space defined in the Experiment Compass. The LLM receives structured prompts containing domain knowledge and optimization objectives, as detailed below.

---

**Initialization prompts received by the LLM**

**Task**
Generate n_hypotheses diverse and well-justified initial hypotheses for exploring the optimization target based on:

**Domain** Organic Chemistry / Reaction Condition Optimization

**Experiment Goal** This experiment aims to optimize the reaction conditions for the Buchwald-Hartwig amination, a key C-N cross-coupling reaction used in the synthesis of pharmaceuticals and functional materials. The optimization focuses on selecting optimal ligands, additives, bases, and aryl halides to maximize reaction yield. High-throughput experimentation and AI-driven decision-making are employed to explore a large combinatorial chemical space.

**Target** {name: Yield, description: Isolated yield percentage of the desired amine product from the B-H coupling reaction, direction: maximize}

**Constraints** Only one choice per category (Ligand, Additive, Base, Aryl halide) can be selected per reaction run. All inputs are categorical.

**Instructions**

- Do NOT refer to memorized reaction outcomes, prior experimental data, or specific literature results (even if they match the current experiment).
- Base all reasoning strictly on the given input, parameter definitions, constraints, and optimization objective.
- For each hypothesis, provide at least one point that satisfies the constraint.

---

**Format Requirement**
Your response must be a valid JSON object in the exact format shown below. No additional commentary, explanations, or syntax should be included.
**Example Output (for format only, unrelated to this experiment):**

**comment**  A brief comment on the initial sampling strategy and rationale (~100 words)

**keywords**  keyword1, keyword2, keyword3

**hypotheses**  1. **strategy**: Descriptive name of exploration approach
**rationale**: Scientific justification for this approach
**confidence**: high/medium/low
**parameter_sets**:
- **ParameterA:** 0.1
  **ParameterB:** 0.9
- **ParameterA:** 0.4
  **ParameterB:** 0.6

2. **strategy**: Descriptive name of exploration approach
**rationale**: ...
**confidence**: high/medium/low
**parameter_sets**:
- **ParameterA:** 0.3
  **ParameterB:** 0.8
- **ParameterA:** 0.1
  **ParameterB:** 0.3

**Important Notes**

- Points must include all required parameters.
- Hypothesis "names" and "parameter" keys must match exactly with the experiment definition.(If they are discrete values, they must be returned strictly according to the format).
- All parameter names and values must exactly match predefined options in spelling, casing, and formatting - no substitutions, abbreviations, or modifications permitted.(if they are discrete values, they must be returned strictly according to the format).
- Any deviation from predefined parameters will result in automatic rejection, regardless of semantic similarity or logical equivalence.

**Parameter specs for this experiment:**

**Ligand**  {display_name: Ligand,
description: Phosphine-based or bulky ligands that influence catalyst reactivity and selectivity,
data_type: categorical,
bounds: [CC(C)(C)C(C=C(C(C)C)C=C1C(C)C)=C1C2=C(P([C@@](3[C@H]4C5)C[C@@H](C4)C[C@H]5C3)[C@]6(C7)C[C@@H](C[C@H]7C8)C[C@H]8C6)C(OC)=CC=C2OC,
CC(C)(C)C(C=C(C(C)C)C=C1C(C)C)=C1C2=C(P(C(C)(C)C)C(C)(C)C)C(OC)=CC=C2OC,
CC(C)(C)C(C=C(C(C)C)C=C1C(C)C)=C1C2=C(P(C(C)(C)C)C(C)(C)C)C=CC=C2,
CC(C)(C)C(C=C(C(C)C)C=C1C(C)C)=C1C2=C(P(C3CCCCC3)C4CCCCC4)C=CC=C2]}

**Additive**  {display_name: Additive,
description: Nitrogen or oxygen heterocycles influencing reactivity and solubility,
data_type: categorical,
bounds: [C1(C2=CC=CC=C2)=CON=C1, C1(C2=CC=CC=C2)=NOC=C1,
C1(N(CC2=CC=CC=C2)CC3=CC=CC=C3)=CC=NO1,
C1(N(CC2=CC=CC=C2)CC3=CC=CC=C3)=NOC=C1,
C12=C(C=CC=C2)ON=C1, C12=CON=C1C=CC=C2, CC1=C(C(OCC)=O)C=NO1,
CC1=CC(C(OCC)=O)=NO1, CC1=NOC(C(OCC)=O)=C1, CC1=CC(N2C=CC=C2)=NO1,
CC1=NOC=C1, CCOC(C1=CON=C1)=O, CCOC(C1=NOC=C1)=O,
O=C(OC)C1=NOC(C(OCC)=O)=C1, FC(C=CC=C1F)=C1C2=CC=NO2,
O=C(OC)C1=CC=NO1, O=C(OC)C1=NOC(C2=CC=CS2)=C1]}

**Base**  {display_name: Base,
description: Nitrogen-based or phosphine-derived bases used for deprotonation and catalytic cycle support,
data_type: categorical,
bounds: [CC(C)(C)/N=C(N(C)C)/N(C)C, CN(C)P(N(C)C)(N(C)C)=NP(N(C)C)(N(C)C)=NCC,
CN1CCCN2C1=NCCC2]}

**Aryl halide**  {display_name: Aryl halide,
       description: Aryl halide substrate used for oxidative addition in the catalytic cycle,
       data_type': categorical,
       bounds: [BrC1=CC=C(C(F)(F)F)C=C1, BrC1=CN=CC=C1, BrC1=NC=CC=C1,
       BrC1=CC=C(OC)C=C1, ClC1=CC=C(C(F)(F)F)C=C1, ClC1=CC=C(CC)C=C1,
       ClC1=CC=C(OC)C=C1, ClC1=CN=CC=C1, ClC1=NC=CC=C1,
       IC1=CC=C(C(F)(F)F)C=C1, IC1=CC=C(OC)C=C1, IC1=CN=CC=C1, IC1=NC=CC=C1]}

The model then synthesizes these inputs to produce initialization insights, recommending candidate points that are strategically positioned in regions likely to yield high performance. An example of the generated insights is shown below.

**LLM-generated initialization insights**

**comment**  The initial sampling strategy focuses on diverse exploration across ligand steric/electronic effects, additive solubility/reactivity, base deprotonation efficacy, and aryl halide substituent effects. Hypotheses are designed to cover contrasting chemical properties (e.g., bulky vs. electron-rich ligands, polar vs. nonpolar additives) to maximize coverage of the combinatorial space and identify high-yield regions.

**keywords**  ligand sterics, additive solubility, base reactivity, halide substituents

**hypotheses**  1.  **strategy**: Bulky Ligand with Electron-Deficient Additive
       **rationale**: A bulky ligand reduces side reactions, while an electron-deficient additive enhances transition state stabilization. A phosphine-derived base balances nucleophilicity, and a trifluoromethyl-substituted halide improves leaving group ability.
       **confidence**: medium
       **parameter_sets**:
       **Ligand:** `CC(C)(C)C(C=C(C(C)C)C=C1C(C)C)=C1C2=C(P([C@@](3[C@H]`
                `4C5)C[C@@H](C4)C[C@H]5C3)[C@]6(C7)C[C@@H](C[C@H]7C8)`
                `C[C@H]8C6)C(OC)=CC=C2OC`
       **Additive:** `C1(C2=CC=CC=C2)=CON=C1`
       **Base:** `CN(C)P(N(C)C)(N(C)C)=NP(N(C)C)(N(C)C)=NCC`
       **Aryl halide:** `BrC1=CC=C(C(F)(F)F)C=C1`
       2.  **strategy**: Solubility-Enhanced Additive with Moderate Base
       **rationale**: A polar additive (with methoxy substituent) improves solubility, paired with a less bulky ligand to avoid steric hindrance. A tertiary amine base minimizes side reactions, while a methoxy-substituted halide stabilizes the transition state.
       **confidence**: medium
       **parameter_sets**:
       **Ligand:** `CC(C)(C)C(C=C(C(C)C)C=C1C(C)C)=C1C2`
                `=C(P(C(C)(C)C)C(C)(C)C)C=CC=C2`
       **Additive:** `CC1=CC(N2C=CC=C2)=NO1`
       **Base:** `CC(C)(C)/N=C(N(C)C)/N(C)C`
       **Aryl halide:** `BrC1=CC=C(OC)C=C1`
       3.  **strategy**: Phosphine-Derived Base with Chloride Halide
       **rationale**: A phosphine-containing base enhances catalytic cycle efficiency. A chloride halide (better leaving group) paired with a moderately bulky ligand balances reactivity, while a simple nitro-containing additive avoids steric interference.
       **confidence**: medium
       **parameter_sets**:
       **Ligand:** `CC(C)(C)C(C=C(C(C)C)C=C1C(C)C)=C1C2`
                `=C(P(C(C)(C)C)C(C)(C)C)C(OC)=CC=C2OC`
       **Additive:** `CC1=CC=NO1`
       **Base:** `CN1CCCN2C1=NCCC2`
       **Aryl halide:** `ClC1=CC=C(C(F)(F)F)C=C1`

These intelligent initializations serve as the starting points for subsequent Bayesian optimization. Following the execution of initial experiments based on these recommendations, the framework incorporates both the newly acquired experimental data

and the initialization points to generate refined suggestions for the next iteration, which are then fed into the optimization loop for further refinement.

## 4. Optimization Loop

The framework enters an iterative optimization cycle. In each loop, the LLM receives a detailed prompt containing all historical experimental data, previously generated insights, new candidate points from the optimizer, and specific instructions for the current task. We provide two examples of such prompts, for the Buchwald-Hartwig and Suzuki coupling reactions, respectively:

---

**Optimization loop prompts received by the LLM**

**Objective**

- **name**: Yield

- **description**: Isolated yield percentage of the desired amine product from the B-H coupling reaction

- **direction**: maximize

**Historical Summary**

You generated the following comments and hypotheses in previous iterations:

**trial_index: 1**

- **insight(comment)**: The optimization shows that reducing ligand bulkiness, using a polar additive (`CC1=CC(N2C=C(C=C2)=N01)`), tertiary amine base (`CC(C)(C)/N=C(N(C)/C)/N(C)C`), and methoxy-substituted aryl halides (`BrC1=CC=C(OC)C=C1`) significantly boost yield. The first trial's bulky ligand strategy underperformed, while the second trial's moderate ligand and additive solubility improvements drove success. Emerging trends suggest that halide leaving group effects (Cl vs Br) and ligand structure (cyclohexyl vs. non-cyclohexyl) could further refine performance. Hypotheses now focus on optimizing additive solubility, ligand cyclohexyl substitution, and halide leaving group properties combinations.

- **keywords**: Ligand substitution, additive solubility, halide leaving group, base efficiency

- **hypotheses**:

  - * **strategy**: Cyclohexyl-Substituted Ligand with Proven Base/Additive
    * **rationale**: Testing a cyclohexyl-substituted ligand (`CC(C)(C)C(C=C1...C=C2)`) to enhance catalyst reactivity while retaining the successful polar additive (`CC1=CC(N2C=C(C=C2)=N01)`), tertiary amine base, and methoxy-substituted bromide. Cyclohexyl groups may improve steric balance.
    * **confidence**: high
    * **parameter_sets**:

---

- · **Ligand**: `CC(C)(C)C(C=C1C(C)=C(C)C=C1C2=C(P(C3CCCC3)C4CCCC4)C=CC=C2`
- · **Additive**: `CC1=CC(N2C=C(C=C2)=N01)`
- · **Base**: `CC(C)(C)/N=C(N(C)/C)/N(C)C`
- · **Aryl halide**: `BrC1=CC=C(OC)C=C1`
  - – ∗ **strategy**: Alternative Base with Cyclohexyl Ligand
    - ∗ **rationale**: Evaluating the alternative base (`CN1CCN2C1=NCCC2`) paired with a cyclohexyl-substituted ligand to potentially enhance deprotonation efficiency, while maintaining the proven polar additive and methoxy-substituted bromide.
    - ∗ **confidence**: medium
    - ∗ **parameter_sets**:
      - · **Ligand**: `CC(C)(C)C(C=C1C(C)=C(C)C=C1C2=C(P(C3CCCC3)C4CCCC4)C=CC=C2`
      - · **Additive**: `CC1=CC(N2C=C(C=C2)=N01)`
      - · **Base**: `CN1CCN2C1=NCCC2`
      - · **Aryl halide**: `BrC1=CC=C(OC)C=C1`

**trial_index: 2**

- **insight(comment)**: The optimization highlights that reducing ligand bulkiness, using a methoxy-substituted bromide, and the tertiary amine base paired with additive (`CC1=CC(N2C=C(C=C2)=N01)`) yield the highest results (e.g., trial 1_1 at 49.69). The cyclohexyl-substituted ligand and iodide halide are promising untested avenues. Hypotheses now focus on evaluating cyclohexyl ligand reactivity, iodide leaving group effects, and additive simplification, aryl substituent effects.

- **keywords**: ligand cyclohexyl substitution, iodide leaving group, additive simplification, aryl substituent effects

- **hypotheses**:
  - – ∗ **strategy**: Cyclohexyl Ligand with Proven Setup
    - ∗ **rationale**: Testing the cyclohexyl-substituted ligand to enhance catalyst reactivity while retaining proven additive, base, and methoxy-bromide halide. This may improve steric balance and electronic effects.
    - ∗ **confidence**: medium
    - ∗ **parameter_sets**:
      - · **Ligand**: `CC(C)(C)C(C=C1C(C)=C(C)C=C1C2=C(P(C3CCCC3)C4CCCC4)C=CC=C2`
      - · **Additive**: `CC1=CC(N2C=C(C=C2)=N01)`
      - · **Base**: `CN1CCN2C1=NCCC2`
      - · **Aryl halide**: `BrC1=CC=C(OC)C=C1`
  - – ∗ **strategy**: Iodide Halide with Optimal Parameters

          * **rationale**: Exploring iodide as the leaving group paired with the most effective ligand, additive, and base to leverage its superior leaving group ability and evaluate reactivity.
          * **confidence**: medium
          * **parameter_sets**:
            · **Ligand**: `CC(C)(C)C(C=C1C(C)=C(C)C=C1C2=C` `(P(C(C)(C)C)C(C)(C)C)C=CC=C2`
            · **Additive**: `CC1=CC(N2C=C(C=C2)=N01)`
            · **Base**: `CN1CCN2C1=NCCC2`
            · **Aryl halide**: `IC1=CC=C(OC)C=C1`
        − * **strategy**: Alternative Aryl Substituent with Bromide
          * **rationale**: Evaluating a bromide aryl halide with an ethyl substituent (`BrC1=CC=C(CC)C=C1`) to assess substituent effects on electronic properties and reactivity, paired with proven components.
          * **confidence**: medium
          * **parameter_sets**:
            · **Ligand**: `CC(C)(C)C(C=C1C(C)=C(C)C=C1C2=C` `(P(C(C)(C)C)C(C)(C)C)C=CC=C2`
            · **Additive**: `CC1=CC(N2C=C(C=C2)=N01)`
            · **Base**: `CN1CCN2C1=NCCC2`
            · **Aryl halide**: `BrC1=CC=C(CC)C=C1`

**New Experimental Data** Your suggested points were evaluated, and the results have been added to the dataset. We now have data from 11 completed experiments.

```
 trial_index,Ligand,Additive,
Base,Aryl_halide,buchwald_mean_value
0_1,
CC(C)(C)C(C=C(C(C)C)C=C1C(C)C)=C1C2=C(P(C(C)(C)C)
C(C)(C)C)C=CC=C2,
CC1=CC(N2C=CC=C2)=N01,
CC(C)(C)/N=C(N(C)C)/N(C)C,
BrC1=CC=C(OC)C=C1,39.214
0_2,
CC(C)(C)C(C=C(C(C)C)C=C1C(C)C)=C1C2=C(P(C(C)(C)C)
C(C)(C)C)C=CC=C2,
CC1=CC(N2C=CC=C2)=N01,
CC(C)(C)/N=C(N(C)C)/N(C)C,BrC1=CC=C(OC)C=C1,39.214
1_0,
CC(C)(C)C(C=C(C(C)C)C=C1C(C)C)=C1C2=C(P(C(C)(C)C)
C(C)(C)C)C=CC=C2,
CCOC(C1=CON=C1)=O,
CC(C)(C)/N=C(N(C)C)/N(C)C,BrC1=CC=C(OC)C=C1,0.748
1_1,
```

```
CC(C)(C)C(C=C(C(C)C)C=C1C(C)C)=C1C2=C(P(C(C)(C)C)
C(C)(C)C)C=CC=C2,
C1(N(CC2=CC=CC=C2)CC3=CC=CC=C3)=NOC=C1,
CN1CCN2C1=NCCC2,
BrC1=CC=C(OC)C=C1,49.691
1_2,
CC(C)(C)C(C=C(C(C)C)C=C1C(C)C)=C1C2=C(P(C(C)(C)C)
C(C)(C)C)C=CC=C2,
CC1=CC(N2C=CC=C2)=N01,
CC(C)(C)/N=C(N(C)C)/N(C)C,
ClC1=CC=C(OC)C=C1,1.03
2_0,
CC(C)(C)C(C=C(C(C)C)C=C1C(C)C)=C1C2=C(P(C3CCCCC3)
C4CCCCC4)C=CC=C2,
CC1=CC(N2C=CC=C2)=N01,
CC(C)(C)/N=C(N(C)C)/N(C)C,
BrC1=CC=C(OC)C=C1,14.423
2_1,
CC(C)(C)C(C=C(C(C)C)C=C1C(C)C)=C1C2=C(P(C(C)(C)C)
C(C)(C)C)C=CC=C2,
CCOC(C1=CON=C1)=O,
CC(C)(C)/N=C(N(C)C)/N(C)C,
ClC1=CC=C(OC)C=C1,0.0
2_2,
CC(C)(C)C(C=C(C(C)C)C=C1C(C)C)=C1C2=C(P(C3CCCCC3)
C4CCCCC4)C=CC=C2,
C1(N(CC2=CC=CC=C2)CC3=CC=CC=C3)=NOC=C1,
CN1CCN2C1=NCCC2,
BrC1=CC=C(OC)C=C1,9.509
2_3,
CC(C)(C)C(C=C(C(C)C)C=C1C(C)C)=C1C2=C(P(C3CCCCC3)
C4CCCCC4)C=CC=C2,
CC1=CC(N2C=CC=C2)=N01,
CN1CCN2C1=NCCC2,
BrC1=CC=C(OC)C=C1,9.509
3_0,
CC(C)(C)C(C=C(C(C)C)C=C1C(C)C)=C1C2=C(P(C(C)(C)C)
C(C)(C)C)C=CC=C2,
CC1=CC(N2C=CC=C2)=N01,
CN1CCN2C1=NCCC2,
IC1=CC=C(OC)C=C1,41.808
3_1,
CC(C)(C)C(C=C(C(C)C)C=C1C(C)C)=C1C2=C(P(C(C)(C)C)
C(C)(C)C)C=CC=C2,
```

```
CC1=CC=N01,CN1CCN2C1=NCCC2,
BrC1=CC=C(OC)C=C1,44.394
```

**Bayesian Optimizer Suggestions** Here are the candidate points recommended by the optimizer in this round:

- {Ligand:`CC(C)(C)C(C=C(C(C)C)C=C1C(C)C)=C1C2` `=C(P(C(C)(C)C)C(C)(C)C)(OC)=CC=C2OC,` `Additive:  CC1=CC(N2C=CC=C2)=N01,` `Base:  CN1CCN2C1=NCCC2,` `Aryl halide:  BrC1=CC=C(CC)C=C1`}

- {Ligand:  `CC(C)(C)C(C=C(C(C)C)C=C1C(C)C)=C1C2` `=C(P(C(C)(C)C)C(C)(C)C)C(OC)=CC=C2OC,` `Additive:  CC1=NOC(C(OCC)=O)=C1,` `Base:  CC(C)(C)/N=C(N(C)C)/N(C)C,` `Aryl halide:  BrC1=CN=CC=C1`}

**Your Task** Reflect on the current state of the optimization. Do not rely on prior assumptions-reason, only based on the data provided.

- What trends or patterns emerge from the experimental data?

- Which regions in the parameter space seem most effective in achieving the target?

- Which areas consistently underperform?

- Improve or revise your previous hypotheses based on observed outcomes.

- You may discard low-performing hypotheses and propose new ones.

- For each hypothesis, you must provide one point that satisfies the constraint: Only one choice per category (Ligand, Additive, Base, Aryl halide) can be selected per reaction run. All inputs are categorical.

**Important**

- Your output must be valid pure JSON without ```json or any other formatting, start with { and end with }. Never put comments or text outside the JSON structure.

- Do NOT wrap it in markdown, text, or explanations.

- Do NOT return fewer than 3 nor more than 5 hypotheses.

- All parameter names and values must exactly match predefined options in spelling, casing, and formatting - no substitutions, abbreviations, or modifications permitted.(if they are discrete values, they must be returned strictly according to the format).

- Any deviation from predefined parameters will result in automatic rejection, regardless of semantic similarity or logical equivalence.

**Parameter specs for this experiment:**

- – **display_name**: `Ligand`

– **description**: Phosphine-based or bulky ligands that influence catalyst reactivity and selectivity

– **data_type**: categorical

– **bounds**:

* `CC(C)C(C=C(C(C)C)C=C1C(C)C)=C1C2=C(P([C@@]3 (C[C@@H]4C5)C[C@H](C4)C[C@H]5C3)[C@]6(C7) C[C@@H](C[C@@H]7C8)C[C@@H]8C6)C(OC)=CC=C2OC`

* `CC(C)C(C=C(C(C)C)C=C1C(C)C)=C1C2=C(P(C(C)(C) C)(C(C)(C)C))C(OC)=CC=C2OC`

* `CC(C)C(C=C(C(C)C)C=C1C(C)C)=C1C2=C(P(C(C)(C) C)(C(C)(C)C))C=CC=C2`

* ...

**Optimization loop prompts received by the LLM**

- **comment**: Pyridine solvent (`O=CN(C)C`) demonstrates strong performance, achieving 91.32% yield with the first ligand. THF underperforms, suggesting solvent choice is critical. The first ligand paired with pyridine shows promise, but alternative ligands may synergize better. Bromine electrophiles could balance reactivity-stability better than iodine. Testing fluorinated nucleophiles under optimal conditions explores electronic effects. KOH base compatibility with pyridine is untested but could improve yields. Underperforming trials involved THF and non-optimal ligand combinations.

- **keywords**: solvent synergy, bromine evaluation, nucleophile substituent effects, ligand optimization, base compatibility

- **hypotheses**:
  - * **strategy**: Bromine Electrophile Optimization
    * **rationale**: Bromine electrophiles may offer better stability-reactivity balance than iodine. Testing under high-yield conditions (pyridine solvent, first ligand) could validate this.
    * **confidence**: medium
    * **parameter_sets**:
      · **Electrophile_SMILES**: `BrC1=CC=C(N=CC=C2)C2=C1`
      · **Nucleophile_SMILES**: `CC1=CC=C(N(C2CCCCO2)N=C3) C3=C1B(O)O`
      · **Ligand_SMILES**: `[c-]1(P(C2=CC=CC=C2)C3=CC =CC̄3)cccc1.[c]4(P(C5=CC=CC=C5)C6=CC=CC =C6)cccc4.[Fe+2]`
      · **Base_SMILES**: `[Cs+].[F-]`
      · **Solvent_SMILES**: `O=CN(C)C`
  - * **strategy**: Alternative Ligand-Pyridine Synergy

* **rationale**: Testing third ligand option
(`CC(C)(P(C(C)(C)C)[c-]1cccc1)C...`) with pyridine solvent to identify optimal catalytic activity.
* **confidence**: medium
* **parameter_sets**:
  · **Electrophile_SMILES**: `IC1=CC=C(N=CC=C2)C2=C1`
  · **Nucleophile_SMILES**: `CC1=CC=C(N(C2CCCCO2)N=C3)C3=C1B(O)O`
  · **Ligand_SMILES**: `CC(C)(P(C(C)(C)C)[c]1cccc1)C.CC(C)(P(C(C)(C)C)[c-]2cccc2)C.[Fe+2]`
  · **Base_SMILES**: `[Cs+].[F-]`
  · **Solvent_SMILES**: `O=CN(C)C`

- **Bayesian Optimizer Suggestions** Here are the candidate points recommended by the optimizer in this round:

```
[{'Electrophile_SMILES': 'BrC1=CC=C(N=CC=C2)C2=C1', '
    ↪ Nucleophile_SMILES': 'CC1=CC=C(N(C2CCCCO2)N=C3)C3=C1[B
    ↪ -](F)(F)F', 'Ligand_SMILES': 'CCCCP(C12C[C@@H]3C[C@@H](C
    ↪ [C@H](C2)C3)C1)C4SC[C@H]5C[C@@H](C[C@H]5C4)C6', '
    ↪ Base_SMILES': 'CC([O-])C.[Li+]', 'Solvent_SMILES': 'CO
    ↪ '}, {'Electrophile_SMILES': 'ClC1=CC=C(N=CC=C2)C2=C1', '
    ↪ Nucleophile_SMILES': 'CC1=CC=C(N(C2CCCCO2)N=C3)C3=C1[B
    ↪ -](F)(F)F', 'Ligand_SMILES': 'P(C1CCCCC1)(C2CCCCC2)
    ↪ C3CCCCC3', 'Base_SMILES': '[Na+].[OH-]', 'Solvent_SMILES
    ↪ ': 'N#CC'}, {'Electrophile_SMILES': 'ClC1=CC=C(N=CC=C2)
    ↪ C2=C1', 'Nucleophile_SMILES': 'CC1=CC=C(N(C2CCCCO2)N=C3)
    ↪ C3=C1B4OC(C)(C)C(C)(C)O4', 'Ligand_SMILES': 'COC1=CC=C(
    ↪ OC)=C1C2=C(P(C3CCCCC3)C4CCCC4)C=CC=C2', 'Base_SMILES': '
    ↪ CCN(CC)CC', 'Solvent_SMILES': 'N#CC'}, {'
    ↪ Electrophile_SMILES': 'ClC1=CC=C(N=CC=C2)C2=C1', '
    ↪ Nucleophile_SMILES': 'CC1=CC=C(N(C2CCCCO2)N=C3)C3=C1[B
    ↪ -](F)(F)F', 'Ligand_SMILES': 'P(C1CCCCC1)(C2CCCCC2)
    ↪ C3CCCCC3', 'Base_SMILES': '[Na+].[OH-]', 'Solvent_SMILES
    ↪ ': 'CO'}, {'Electrophile_SMILES': 'ClC1=CC=C(N=CC=C2)C2=
    ↪ C1', 'Nucleophile_SMILES': 'CC1=CC=C(N(C2CCCCO2)N=C3)C3=
    ↪ C1B4OC(C)(C)C(C)(C)O4', 'Ligand_SMILES': 'COC1=CC=C(OC)=
    ↪ C1C2=C(P(C3CCCCC3)C4CCCC4)C=CC=C2', 'Base_SMILES': 'CCN(
    ↪ CC)CC', 'Solvent_SMILES': 'CO'}]
```

- **Your Task** Reflect on the current state of the optimization. Do not rely on prior assumptions-reason, only based on the data provided.

  – What trends or patterns emerge from the experimental data?
  – Which regions in the parameter space seem most effective in achieving the target?
  – Which areas consistently underperform?
  – Improve or revise your previous hypotheses based on observed outcomes.
  – You may discard low-performing hypotheses and propose new ones.
  – For each hypothesis, **at least provide one point** that satisfies the constraint: Reagents must be chemically compatible and reaction

conditions must maintain intermediate stability. All combinations must form stable palladium complexes during catalytic cycle.

- **Important**
  - Your output must be valid pure JSON without ```` ```json ```` or any other formatting, start with { and end with }. Never put comments or text outside the JSON structure.
  - Do NOT wrap it in markdown, text, or explanations.
  - Do NOT return fewer than 3 nor more than 5 hypotheses.
  - All parameter names and values must exactly match predefined options in spelling, casing, and formatting - no substitutions, abbreviations, or modifications permitted.(if they are discrete values, they must be returned strictly according to the format).
  - Any deviation from predefined parameters will result in automatic rejection, regardless of semantic similarity or logical equivalence.
  - Parameter specs for this experiment:

```
[{'display_name': 'Electrophile_SMILES', 'description': '
    ↪ Aryl halide substrate for coupling (SMILES
    ↪ representation)', 'data_type': 'categorical', '
    ↪ bounds': ['BrC1=CC=C(N=CC=C2)C2=C1', 'ClC1=CC=C(N=CC
    ↪ =C2)C2=C1', 'IC1=CC=C(N=CC=C2)C2=C1', 'O=S(OC1=CC=C(
    ↪ N=CC=C2)C2=C1)(C(F)(F)F)=O']}, {'display_name': '
    ↪ Nucleophile_SMILES', 'description': 'Boronic acid
    ↪ nucleophile (SMILES representation)', 'data_type': '
    ↪ categorical', 'bounds': ['CC1=CC=C(N(C2CCCCO2)N=C3)
    ↪ C3=C1[B-](F)(F)F', 'CC1=CC=C(N(C2CCCCO2)N=C3)C3=C1B(
    ↪ O)O', 'CC1=CC=C(N(C2CCCCO2)N=C3)C3=C1B4OC(C)(C)C(C)(
    ↪ C)O4']}, {'display_name': 'Ligand_SMILES', '
    ↪ description': 'Phosphine ligand for palladium
    ↪ catalyst (SMILES)', 'data_type': 'categorical', '
    ↪ bounds': ['[c-]1(P(C2=CC=CC=C2)C3=CC=CC=C3)cccc1.[c
    ↪ -]4(P(C5=CC=CC=C5)C6=CC=CC=C6)cccc4.[Fe+2]']......
```

**New Experimental Data** Your suggested points were evaluated, and the results have been added to the dataset. We now have data from 6 completed experiments:

```
trial_index,Electrophile_SMILES,Nucleophile_SMILES,Ligand_SMILES,
    ↪ Base_SMILES,Solvent_SMILES,Objective_Mean_value
0_0,IC1=CC=C(N=CC=C2)C2=C1,CC1=CC=C(N(C2CCCCO2)N=C3)C3=C1B(O)O,[c-]1(
    ↪ P(C2=CC=CC=C2)C3=CC=CC=C3)cccc1.[c-]4(P(C5=CC=CC=C5)C6=CC=CC=C6
    ↪ )cccc4.[Fe+2],[Cs+].[F-],C1COCC1,54.13
0_1,IC1=CC=C(N=CC=C2)C2=C1,CC1=CC=C(N(C2CCCCO2)N=C3)C3=C1B(O)O,CC(C)(
    ↪ P(C(C)(C)C)[c-]1cccc1)C.CC(C)(P(C(C)(C)C)[c-]2cccc2)C.[Fe+2],[
    ↪ Cs+].[F-],C1COCC1,43.13
0_2,IC1=CC=C(N=CC=C2)C2=C1,CC1=CC=C(N(C2CCCCO2)N=C3)C3=C1B(O)O,[c-]1(
    ↪ P(C2=CC=CC=C2)C3=CC=CC=C3)cccc1.[c-]4(P(C5=CC=CC=C5)C6=CC=CC=C6
    ↪ )cccc4.[Fe+2],[Cs+].[F-],O=CN(C)C,91.32
1_0,IC1=CC=C(N=CC=C2)C2=C1,CC1=CC=C(N(C2CCCCO2)N=C3)C3=C1B(O)O,CC(C)(
    ↪ P(C(C)(C)C)[c-]1cccc1)C.CC(C)(P(C(C)(C)C)[c-]2cccc2)C.[Fe+2],[
    ↪ Cs+].[F-],O=CN(C)C,76.61
1_1,IC1=CC=C(N=CC=C2)C2=C1,CC1=CC=C(N(C2CCCCO2)N=C3)C3=C1[B-](F)(F)F
    ↪ ,[c-]1(P(C2=CC=CC=C2)C3=CC=CC=C3)cccc1.[c-]4(P(C5=CC=CC=C5)C6=
    ↪ CC=CC=C6)cccc4.[Fe+2],[K+].[OH-],O=CN(C)C,70.99
```

```
1_2,IC1=CC=C(N=CC=C2)C2=C1,CC1=CC=C(N(C2CCCCO2)N=C3)C3=C1[B-](F)(F)F
   ↪ ,[c-]1(P(C2=CC=CC=C2)C3=CC=CC=C3)cccc1.[c-]4(P(C5=CC=CC=C5)C6=
   ↪ CC=CC=C6)cccc4.[Fe+2],[Cs+].[F-],O=CN(C)C,19.51
2_0,BrC1=CC=C(N=CC=C2)C2=C1,CC1=CC=C(N(C2CCCCO2)N=C3)C3=C1B(O)O,[c
   ↪ -]1(P(C2=CC=CC=C2)C3=CC=CC=C3)cccc1.[c-]4(P(C5=CC=CC=C5)C6=CC=
   ↪ CC=C6)cccc4.[Fe+2],[Cs+].[F-],O=CN(C)C,14.38
3_0,IC1=CC=C(N=CC=C2)C2=C1,CC1=CC=C(N(C2CCCCO2)N=C3)C3=C1B4OC(C)(C)C(
   ↪ C)(C)O4,[c-]1(P(C2=CC=CC=C2)C3=CC=CC=C3)cccc1.[c-]4(P(C5=CC=CC=
   ↪ C5)C6=CC=CC=C6)cccc4.[Fe+2],[Cs+].[F-],O=CN(C)C,80.02
3_1,IC1=CC=C(N=CC=C2)C2=C1,CC1=CC=C(N(C2CCCCO2)N=C3)C3=C1B4OC(C)(C)C(
   ↪ C)(C)O4,CC(C1=C(C2=CC=CC=C2P(C3CCCC3)C4CCCC4)C(C(C)C)=C(C(C)C)
   ↪ C1)C,[Cs+].[F-],O=CN(C)C,85.48
4_0,S(OC1=CC=C(N=CC=C2)C2=C1)(C(F)(F)F)=O,CC1=CC=C(N(C2CCCCO2)N=C3)C3
   ↪ =C1B(O)O,[c-]1(P(C2=CC=CC=C2)C3=CC=CC=C3)cccc1.[c-]4(P(C5=CC=CC
   ↪ =C5)C6=CC=CC=C6)cccc4.[Fe+2],[Cs+].[F-],O=CN(C)C,97.32
4_1,ClC1=CC=C(N=CC=C2)C2=C1,CC1=CC=C(N(C2CCCCO2)N=C3)C3=C1B(O)O,[c
   ↪ -]1(P(C2=CC=CC=C2)C3=CC=CC=C3)cccc1.[c-]4(P(C5=CC=CC=C5)C6=CC=
   ↪ CC=C6)cccc4.[Fe+2],[Cs+].[F-],O=CN(C)C,8.67
5_0,IC1=CC=C(N=CC=C2)C2=C1,CC1=CC=C(N(C2CCCCO2)N=C3)C3=C1B4OC(C)(C)C(
   ↪ C)(C)O4,CC(C1=C(C2=CC=CC=C2P(C3CCCC3)C4CCCC4)C(C(C)C)=C(C(C)C)
   ↪ C1)C,[Cs+].[F-],O=CN(C)C,75.7
```

Based on such a prompt, the LLM analyzes the provided data and generates new hypotheses to guide the subsequent round of experiments. An example of the LLM's output, containing its reasoning and recommended parameter sets, is shown below:

**LLM-Generated Recommendations**

- **comment**: The highest yields consistently arise from the ligand
  `CC(C)(C)C...C=C2` paired with additive
  `CC1=CC(N2C=CC=C2)=N01` and base `CN1CCN2C1=NCCC2`.
  Methoxy-substituted bromides (`BrC1=CC=C(OC)C=C1`) perform
  well, while iodide (`IC1=CC=C(OC)C=C1`) shows promise but
  requires further optimization. Underperforming parameters include
  chloride halides and additives like `CCOC(C1=CON=C1)=O`. Emerging
  trends suggest substituent effects (e.g., ethyl groups) and simplified
  additives could enhance yields.

- **keywords**: iodide leaving group, substituent effects, additive simplification, aryl halide substituent

- **hypotheses**:

  - * **strategy**: Ethyl Substituent with Proven Components
    * **rationale**: Evaluating the ethyl-substituted aryl halide
      (`BrC1=CC=C(CC)C=C1`) paired with the top-performing
      ligand, additive, and base to assess substituent effects on
      electronic properties and reactivity.
    * **confidence**: medium
    * **parameter_sets**:
      · **Ligand**: `CC(C)(C)C(C=C1C(C)=C(C)C=C1C2=C`
                    `(P(C(C)(C)C)C(C)(C)C)C=CC=C2`
      · **Additive**: `CC1=CC(N2C=C(C=C2)=N01)`

- · **Base**: `CN1CCN2C1=NCCC2`
- · **Aryl halide**: `BrC1=CC=C(CC)C=C1`
  - – * **strategy**: Iodide Halide with Simplified Additive
    - * **rationale**: Combining iodide (`IC1=CC=C(OC)C=C1`) with the simplified additive (`CC1=CC=N01`) to leverage superior leaving group ability and reduced steric hindrance while retaining proven ligand and base.
    - * **confidence**: medium
    - * **parameter_sets**:
      - · **Ligand**: `CC(C)(C)C(C=C1C(C)=C(C)C=C1C2=C` `(P(C(C)(C)C)C(C)(C)C)C=CC=C2`
      - · **Additive**: `CC1=CC=N01`
      - · **Base**: `CN1CCN2C1=NCCC2`
      - · **Aryl halide**: `IC1=CC=C(OC)C=C1`
  - – * **strategy**: Alternative Additive with Methoxy Bromide
    - * **rationale**: Testing a novel additive (`CC1=NOC(C(OCC)=O)=C1`) with the top-performing ligand, base, and methoxy-substituted bromide to explore new reactivity pathways.
    - * **confidence**: medium
    - * **parameter_sets**:
      - · **Ligand**: `CC(C)(C)C(C=C1C(C)=C(C)C=C1C2=C` `(P(C(C)(C)C)C(C)(C)C)C=CC=C2`
      - · **Additive**: `CC1=NOC(C(OCC)=O)=C1`
      - · **Base**: `CN1CCN2C1=NCCC2`
      - · **Aryl halide**: `BrC1=CC=C(OC)C=C1`
  - – * **strategy**: Alternative Aryl Halide Structure with Optimal Setup
    - * **rationale**: Evaluating the aryl halide `BrC1=CN=CC=C1` to assess the impact of a nitrogen-containing substituent paired with proven components.
    - * **confidence**: medium
    - * **parameter_sets**:
      - · **Ligand**: `CC(C)(C)C(C=C1C(C)=C(C)C=C1C2=C` `(P(C(C)(C)C)C(C)(C)C)C=CC=C2`
      - · **Additive**: `CC1=CC(N2C=C(C=C2)=N01)`
      - · **Base**: `CN1CCN2C1=NCCC2`
      - · **Aryl halide**: `BrC1=CN=CC=C1`

These recommendations are then evaluated through Bayesian optimization, and the results are fed back into the system for the next iteration, continuing until predefined limits are reached.

## D.4    5. SUMMARY AND CONCLUSION

In the final stage of the optimization framework, the LLM synthesizes all accumulated insights and experimental data to generate comprehensive summaries and conclusions. The system's final summary of the Buchwald-Hartwig amination optimization is presented below, highlighting key findings and performance metrics. The concluding analysis provides actionable recommendations and broader implications for reaction optimization, demonstrating the framework's ability to extract meaningful chemical insights from complex experimental data.

---

**Summary of the Buchwald-Hartwig Amination Optimization**

**Hypothesis Evolution:**

The optimization began with diverse exploration of ligand sterics, additive solubility, and halide substituents. Early hypotheses prioritized bulky ligands and electron-deficient additives but underperformed (e.g., trial0_0 yielded 13.524).The second trial (0_1) introduced a less bulky ligand, polar additive (CC1=CC(N2C=C(C)=N01)=C1), and methoxy-substituted bromide, achieving 39.214, signaling a shift toward moderate steric effects and solubility-enhanced additives.

By trials 2-5, hypotheses focused on iodide leaving groups and substituent effects. Chloride halides consistently underperformed, while iodide-based trials (e.g., trial 8_0:84.896) and nitrogen-containing aryl halides (e.g., trial 6_1:80.074) showed promise. A breakthrough occurred in trial 6_2, where the ethyl-substituted bromide with additive CC1=NOC(C(OCC)=O)=C1 achieved 86.598, highlighting substituent-electronic effects.

Later trials (8-12) refined iodide nitrogen aryl halides (e.g., IC1=NC=C(C)=C1), achieving yields exceeding 99% (trial 10_1:99.619). Hypotheses increasingly prioritized iodide's superior leaving group ability paired with nitrogen substituents, while additives like CCOC(C1=CON=C1)=O were consistently discarded. Final trials confirmed top combination (non-bulky ligand, additive CC1=NOC(C(OCC)=O)=C1, base CN1CCON2C1=NCCC2, and iodide nitrogen aryl halides) as the high-yield standard.

**Supported vs. Refuted Hypotheses:**

- **Supported:**
    - Reduced ligand bulkiness and polar additives (e.g., CC1=C(N2C=C(C)=NO1) consistently boosted yields.
    - Iodide halides with nitrogen substituents (e.g., IC1=NC=C(C)=C1) outperformed bromine and chloride variants.
    - Additive CC1=NOC(C(OCC)=O)=C1 emerged as optimal, stabilizing yields above 98% when paired with top components.

- **ReFuted:**
    - Bulky ligands (e.g., cyclohexyl-substituted) and electron-withdrawing substituents (e.g., trifluoromethyl underperformed.

---

– Chloride halides and additives like `CCOC(C1=CON=C1)=O` consistently failed.

**Confidence Shifts:** Initial hypotheses had medium confidence but gained certainty as the top combination (ligand3, additive2, base3, `IC1=NC=C(C)=C1`) repeatedly achieved >99% yields. Confidence in iodide nitrogen aryl halides rose sharply after trial 9_0 (94.691), while cyclohexyl ligands and chloride halides were discarded after consistent underperformance.

**Conclusions:** The optimal conditions require a non-bulky ligand (`C(C)(C)C(C1=C(P(C(C)(C)C)C)C=C(C)C=C1)...`), additive `CC1=NOC(C(OCC)=O)=C1`, base `CN1CCON2C1=NCCC2`, and iodide aryl halides with nitrogen substituents (e.g., `IC1=NC=C(C)=C1`). Substituent electronic effects and additive-aryl halide synergy were critical, with nitrogen substituents enhancing reactivity and iodide's leaving group ability maximizing yields. Chloride and bulky ligands were unsuitable, while additives like `CCOC(C1=CON=C1)=O` were ineffective. The final dataset confirms the top combination's robustness, with minor variations (e.g., bromine or simplified additives) yielding <99.6%.

| Trial Range | Hypothesis Focus | Confidence Shift |
|---|---|---|
| Trials 0-3 | Diverse ligands, additives, and halide types | Low – Medium (early failures) |
| Trials 4-8 | Ethyl-substituted bromides, iodide exploration | Medium – High (ethyl iodide) |
| Trials 9-29 | Iodide nitrogen aryl halides and additive CC1 | High (stable top performers) |
| Final Trials | Confirmatory testing of top combination | High (no further improvements) |

The process converged on a high-yield region defined by specific substituent-additive interactions, with confidence peaking in the final hypotheses.

---

**Conclusion of the Buchwald-Hartwig Amination Optimization**

1. **Key Outcomes** The optimization achieved a maximum yield of 99.6% for the Buchwald-Hartwig amination reaction, representing a 658% improvement over the initial trial. The most influential parameters were the aryl halide substituent (iodide with nitrogen-containing ring), additive (`CC1=NOC(C(OCC)=O)=C1`), and base (`CN1CCON2C1=NCCC2`). The process demonstrated high efficiency, converging to near-optimal yields within 30 trials through systematic exploration of substituent effects and catalyst-component synergies.

2. **Experimental Retrospective**

   • **Objectives:** Maximize isolated yield in a palladium-catalyzed B-H coupling reaction by optimizing ligand sterics, additive solubility/reactivity, base efficacy, and aryl halide substituent effects.

- **Initial Approach:** The first hypotheses (Trial #0) prioritized diverse exploration, testing bulky ligands, polar/nonpolar additives, and halides with varied leaving group abilities. Trial #0_0 (bulky ligand/electron-deficient additive) achieved only 13.5% yield, while Trial #0_1 (moderate ligand/polar additive) reached 39.2%, establishing additive solubility and ligand bulk as critical factors.

3. **Optimization Journey** Between trials 0-5: Initial validation revealed ligand bulk reduction and polar additives (e.g., Trial #0_1) significantly boost yield. Confidence in bulky ligands dropped from "medium" to "discarded" (Evidence: Trial #0_0 vs. #0_1). Between 6-15: Substituent effects emerged as key drivers. Ethyl-substituted bromide (Trial #4_0, 71.6%) and nitrogen-containing aryl halides (Trial #6_1, 80.0%) showed promise, shifting focus to electronic properties. Confidence in iodide halides rose to "high" after Trial #8_1 (iodide nitrogen aryl, 92.6%). Between 16-30: Refinement of substituent isomers and additive variations solidified the top configuration. Trial #10_1 (iodide nitrogen aryl with optimized components) achieved 99.6%, confirming substituent position and additive-electron interactions as decisive factors.

4. **Definitive Findings**

   - **Optimal Configuration:**
     Ligand3 (non-bulky, `CC(C)C(C=C(C(C)C)C=C1C(C)C)=C1 C2=C(P(C(C)(C)C)C(C)(C)C)C=CC=C2`),
     Additive2 (`CC1=NOC(C(OCC)=O)=C1`),
     Base3 (`CN1CCON2C1=NCCC2`),
     Aryl Halide14 (`IC1=NC=C(C)=C1`).
     This combination achieved >**99.6% yield** (Trial #10_1).

   - **Parameter Relationships:**
     - **Aryl Halides:**
       Iodide with nitrogen-containing substituents (e.g., `IC1=NC=C(C)=C1`) outperformed bromine/chloride due to superior leaving group ability and electronic tuning.
       Positional isomers (e.g., `IC1=CN=C(C)=C1`, Trial #9_0, 94.7%) showed yield sensitivity to substituent placement.
     - **Additives:** The oxymidazole-based additive (`CC1=NOC(C(OCC)=O)=C1`) enabled transition state stabilization, with phenyl-substituted variants (Trial #3_2, 69.9%) underperforming.
     - **Ligand-Base Pairing:** Non-bulky Ligand3 with base3 (Trial #10_1) maximized catalytic efficiency, while bulky ligands (Trial #2_2, 9.5%) disrupted coordination.

5. **Forward Guidance**

   - **Immediate Recommendations:**

> (a) Validate the top configuration (Trial #10_1) with additional replicates to confirm reproducibility.
>
> (b) Explore bromine-based nitrogen aryl halides (Trial #10_0, 97.3%) as scalable alternatives to iodide due to cost considerations.
>
> - **Long-term Considerations:**
>   - Investigate substituent effects on catalyst turnover numbers and side reaction suppression.
>   - Evaluate alternative additives (e.g., Trial #15_0's `COC1=NOC...`, 84.7%) for cost-optimized industrial setups.
>
> 6. **Scientific Impact** This work underscores the interplay between electronic effects and leaving group ability in transition metal-catalyzed couplings. The nitrogen-substituted iodide aryl halide (Trial #10_1) exemplifies how substituent design can synergistically enhance nucleophilicity and transition state stabilization. The additive's role in solubility and catalyst activation (Trial #8_2 vs. Trial #10_1) provides a framework for rational co-solvent selection. Practically, this configuration enables >99% yields in C-N cross-coupling, advancing scalable synthesis of medicinally relevant amines. The iterative strategy-prioritizing substituent screening after establishing core components-demonstrates an efficient pathway for complex reaction optimization.

## E    BENCHMARKS

### E.1    SUZUKI REACTION (5D)

The Suzuki-Miyaura cross-coupling reaction is a widely used strategy in organic synthesis for forming biaryl motifs by coupling an aryl halide (electrophile) with a boronic acid derivative (nucleophile), catalyzed by palladium complexes. This benchmark simulates a combinatorial exploration of substrate pairs and reaction conditions in a data-driven optimization task.

- **Objective**: Maximize the product yield (%) of the Suzuki reaction through selection of optimal electrophile-nucleophile pairs, ligand, base, and solvent under catalyst compatibility constraints.

- **Optimization Variables**: The input variables include the electrophile SMILES, nucleophile SMILES, ligand SMILES, base SMILES, and solvent SMILES, which are described in detail in Table 2.

- **Target**: Simulated or experimentally derived product yield (%) of the coupling reaction.

- **Constraints**:
  - All reagent combinations must form stable palladium complexes.

| Variable Name | Description |
|---|---|
| Electrophile_SMILES | Aryl halide substrate represented as SMILES. Four halide derivatives tested. |
| Nucleophile_SMILES | Boronic acid substrate in SMILES form. Three boronate variants included. |
| Ligand_SMILES | Phosphine ligands used to form Pd(0) complexes. Eleven ligands included. |
| Base_SMILES | Base required for the transmetallation step. Seven options provided. |
| Solvent_SMILES | Solvent medium (SMILES); affects solubility and catalyst stability. |

Table 2: Optimization variables for the Suzuki reaction benchmark

- – Conditions must support the full catalytic cycle including oxidative addition, transmetallation, and reductive elimination.

- **Challenges**:

  - – Discrete Combinatorial Landscape: The entire search space is composed of categorical variables with strong dependency interactions, making it highly multimodal and sparse.
  - – Chemical Compatibility Filtering: Some electrophile–ligand–base combinations are chemically incompatible and lead to decomposition or catalyst poisoning, creating discontinuities in the yield landscape.
  - – Ligand Diversity: The ligands span a broad range of steric and electronic profiles, affecting both oxidative addition rates and Pd complex stability.

This benchmark is designed to assess the optimizer's ability to navigate highly categorical chemical reaction spaces, identify synergistic effects, and avoid known failure modes due to chemical incompatibility—all essential in real-world medicinal chemistry and process development workflows.

E.2 DIRECT ARYLATION REACTION (5D)

Direct Arylation Reaction Optimization (5D) enables C–H activation without the need for pre-functionalized substrates, offering a highly atom-economical route for biaryl bond formation. However, this reaction is notoriously sensitive to reaction conditions, including ligand/base/solvent synergy and subtle effects from concentration and temperature. This benchmark simulates a complex reaction landscape composed entirely of categorical and discrete variables, designed to optimize reaction yield under experimentally feasible constraints.

- **Objective**: Maximize the isolated product yield (%) of a direct arylation reaction by selecting an optimal combination of ligand, base, solvent, concentration, and temperature.

- **Optimization Variables**: As detailed in Table 3, we optimize five key variables: Base_SMILES, Ligand_SMILES, Solvent_SMILES, Concentration, and Temp_C.

- **Target**: Reaction_Yield (%) - isolated or calculated product yield under each condition set.

- **Constraints**:

| Variable Name | Description |
|---|---|
| Base_SMILES | Base used in the reaction, including its counterion (e.g., cesium or potassium salts). |
| Ligand_SMILES | Phosphine ligand coordinating the Pd center. Spans bulky to $\pi$-extended ligands. |
| Solvent_SMILES | Solvent system influencing reactivity and selectivity. Four options included. |
| Concentration | Molar concentration (mol/L) of the reactants. Discrete experimental levels. |
| Temp_C | Reaction temperature (C). Three practical heating levels are considered. |

Table 3: Optimization variables for the Direct Arylation reaction benchmark

- All SMILES strings must correspond to chemically feasible components.
- Temperature and concentration values are restricted to predefined, experimentally validated levels.

- **Challenges**:

  - Categorical-Only Input Space: All variables are discrete, forcing the optimizer to navigate a fully combinatorial landscape.
  - Multi-Factor Interactions: Ligand–base–solvent compatibility plays a critical role; some combinations lead to zero or negligible reactivity.
  - Kinetic–Thermodynamic Balance: Subtle shifts in temperature or dilution can invert reaction selectivity or alter decomposition profiles.

This benchmark serves as a stringent test for discrete Bayesian optimization frameworks, requiring reasoning over rich chemical priors and non-obvious synergistic effects. It reflects a realistic synthetic design problem where trial numbers must be minimized, and each reaction carries real-world material and time cost.

E.3   BUCHWALD-HARTWIG REACTION (4D)

The Buchwald-Hartwig amination is a palladium-catalyzed cross-coupling reaction that forms C-N bonds, widely used in pharmaceutical synthesis and materials science (Shields et al., 2021). This benchmark evaluates optimization algorithms on a 4-dimensional categorical space representing key reaction components.

| Variable | Description |
|---|---|
| Ligand_SMILES | Phosphine-based ligands modulating catalyst activity. Includes monophosphines and bulky phosphines. |
| Additive_SMILES | Reaction modifiers affecting intermediate stability and side reactions. |
| Base_SMILES | Reagents supporting deprotonation and catalyst turnover. |
| ArylHalide_SMILES | Electrophilic component determining oxidative addition efficiency. |

Table 4: Optimization variables for the Buchwald-Hartwig reaction benchmark

- **Objective**: Maximize the isolated yield (%) of the desired amine product by optimizing ligand, additive, base, and aryl halide combinations while maintaining catalytic cycle stability.

- **Optimization Variables**: As detailed in Table 4, we optimize four categorical variables: Ligand_SMILES, Additive_SMILES, Base_SMILES, and ArylHalide_SMILES.

- **Target**: Reaction_Yield (%) - isolated product yield under each condition set.

- **Challenges**:
  - Fully Categorical Space: Requires navigation of discrete combinations with strong multi-modality.
  - Catalyst Deactivation: Some combinations may lead to inactive catalytic systems.
  - Complex Ligand Effects: Steric and electronic properties dramatically influence reactivity.

This benchmark provides a realistic test for discrete optimization in chemical reaction spaces, requiring algorithms to recognize synergistic patterns while avoiding invalid regions - a critical capability for pharmaceutical process development.

E.4 CHIRAL PHOSPHORIC ACIDCATALYZED THIOLIMINE ADDITION REACTIONS (3D)

The chiral phosphoric acid (CPA)catalyzed addition of thiols to N-acylimines is a powerful strategy for constructing chiral thioaminal motifs, widely used in asymmetric catalysis and pharmaceutical synthesis. This benchmark task focuses on optimizing reaction conditions to achieve high enantioselectivity by maximizing the free energy difference ($\Delta\Delta G$, kcal/mol) between competing transition states leading to each enantiomer.

| Variable | Description |
|---|---|
| Catalyst | Chiral phosphoric acids derived from the BINOL scaffold, with diverse steric and electronic profiles through variation at the 3,3'-substitution sites. |
| Imine | N-acyl imine substrates derived from various aryl aldehydes and amines, serving as the electrophilic partner. |
| Thiol | A range of nucleophilic thiol reagents whose structure impacts both reactivity and side reaction propensity. |

Table 5: Optimization variables for the Chiral Phosphoric AcidCatalyzed ThiolImine Addition Reaction benchmark

- **Objective**: Maximize enantioselectivity ($\Delta\Delta G$) while preserving mechanistic integrity during the asymmetric transformation.

- **Optimization Variables**: As detailed in Table 5, we optimize three categorical variables: Catalyst, Imine, and Thiol.

- **Target Metric**: Free energy difference ($\Delta\Delta G$, kcal/mol), quantifying enantioselectivity. Higher values indicate stronger stereoselective induction.

- **Experimental Constraints**:

  - Reactions must maintain CPA catalytic activity and stereocontrol under organic solvent conditions.
  - Avoid side reactions such as dehydration, polymerization, or oxidation by conducting the reaction under mild conditions.
  - Catalysts must be synthetically accessible and conformationally stable.

- **Optimization Challenges**:

  - High-dimensional categorical space: Purely discrete variables with non-linear and multimodal response surfaces.
  - Stereoselectivity sensitivity: Small structural changes in the CPA catalyst can significantly affect $\Delta\Delta G$ and product selectivity.
  - Catalystsubstrate synergy: Effective combinations depend on subtle non-covalent interactions and spatial complementarity.

This benchmark provides a realistic challenge for evaluating optimization algorithms' ability to navigate complex discrete spaces and identify synergistic combinations that enhance enantioselectivity while avoiding failed reactions.

E.5   LUNAR LANDER TASK (12D)

The Lunar Lander is a classic control problem where the objective is to design a policy that safely lands a spacecraft on the moon's surface while minimizing fuel consumption and distance from the target landing zone. This benchmark evaluates optimization algorithms on a 12-dimensional space combining continuous state variables and discrete actions.

| Variable | Description | Bound |
|---|---|---|
| horizontal_position | Normalized x-coordinate of lander | [-1.5, 1.5] |
| vertical_position | Normalized y-coordinate of lander | [-1.5, 1.5] |
| horizontal_velocity | Normalized x-velocity | [-5.0, 5.0] |
| vertical_velocity | Normalized y-velocity | [-5.0, 5.0] |
| angle | Orientation in radians | [-3.14, 3.14] |
| angular_velocity | Rotation rate | [-5.0, 5.0] |
| left_leg_contact | Boolean ground contact | [0, 1] |
| right_leg_contact | Boolean ground contact | [0, 1] |
| no_action | No engine firing | [0, 1] |
| fire_left_engine | Left rotation thrust | [0, 1] |
| fire_main_engine | Vertical deceleration | [0, 1] |
| fire_right_engine | Right rotation thrust | [0, 1] |

Table 6: Optimization variables for the Lunar Lander benchmark

- **Objective**: Maximize the average landing reward over 50 terrain variations by optimizing the control policy's response to the 8D state space and 4D action space.

- **Optimization Variables**: As shown in Table 6, we optimize 12 variables including position, velocity, orientation states, and discrete engine control actions.

- **Target**: The average landing reward is calculated based on continuous rewards during descent and final outcomes. Each step's reward depends on the lander's position, velocity, and orientation relative to the landing pad. Leg contacts add +10 points each, while engine firings deduct points (-0.3 per main engine frame, -0.03 per side engine frame). The episode concludes with +100 for safe landing or -100 for crashing, summed with all step rewards.

- **Constraints**:
  - State variables must remain within physical bounds
  - Only one action can be active at any time
  - Successful landing requires coming to rest within target zone

- **Challenges**:
  - High-dimensional mixed space with continuous and discrete variables
  - Delayed reward signals requiring long-term planning
  - Precise control needed during final descent phase
  - Trade-off between fuel efficiency and landing accuracy

This benchmark tests an optimizer's ability to handle hybrid spaces and delayed rewards in a physics-based environment. The problem originates from OpenAI's Gymnasium implementation (Lunar Lander environment).

### E.6 LEVY FUNCTION (5D)

The Levy function (Laguna & Mart, 2005) is a challenging synthetic benchmark for global optimization algorithms, particularly in high-dimensional spaces. The 5D implementation has the form:

$$f(\mathbf{x}) = sin^2(\pi w_1) + \sum_{i=1}^{4}(w_i - 1)^2[1 + 10\sin^2(\pi w_i + 1)] + (w_5 - 1)^2[1 + \sin^2(2\pi w_5)]$$

where $w_i = (1 + \frac{x_i - 1}{4})$ and $\mathbf{x} = [x_1, x_2, x_3, x_4, x_5]$ is the input vector. The function has a global minimum of 0 at $x^* = [1, 1, 1, 1, 1]$ within the search space $x_i \in [-10, 10]$. Its highly multimodal, non-convex landscape features numerous local minima, complex variable interactions, and a rugged oscillatory surface that becomes exponentially more challenging with increasing dimensionality, making it an excellent test for an optimizer's ability to escape local optima and navigate complex high-dimensional spaces with multiple basins of attraction.

### E.7 HARTMANN FUNCTION (6D)

The 6-dimensional Hartmann function (Picheny et al., 2012) is a challenging synthetic benchmark for global optimization algorithms, particularly in high-dimensional spaces. The function takes the form:

$$f(\mathbf{x}) = -\sum_{i=1}^{4}\alpha_i \exp\left(-\sum_{j=1}^{6}A_{ij}(x_j - P_{ij})^2\right)$$

where $\mathbf{x} = [x_1, x_2, x_3, x_4, x_5, x_6]$ is the input vector within the hypercube $x_i \in (0, 1)$. The 6D version contains approximately $10^6$ local minima, with the global minimum surrounded by multiple high-barrier local optima. The function exhibits strong nonlinear coupling effects between parameters, creating a complex energy landscape that tests an optimizer's ability to navigate multimodal spaces and escape local traps. The global minimum value is approximately $-3.32237$ in the 6D case, with numerous suboptimal solutions separated by energy barriers $\Delta f \geq 0.5$, making it particularly challenging for optimization algorithms to locate the true optimum.

### E.8 ACKLEY FUNCTION (2D)

The 2-dimensional Ackley function and 15-dimensional (Adorio & January, 2005) is a widely used benchmark for testing optimization algorithms' ability to balance global exploration and local exploitation. The function takes the form:

$$f(\mathbf{x}) = -20 \exp\left(-0.2\sqrt{\frac{1}{2}\sum_{i=1}^{2} x_i^2}\right) - \exp\left(\frac{1}{2}\sum_{i=1}^{2}\cos(2\pi x_i)\right) + 20 + \exp(1)$$

where $\mathbf{x} = [x_1, x_2]$ is the input vector within the search space $x_i \in [-32.768, 32.768]$. The function has a global minimum of 0 at $x^* = [0, 0]$. Its landscape features an almost flat outer region that can trap optimization algorithms in local optima, combined with a narrow funnel-shaped global optimum region that requires precise local search. The deceptive nature of the function comes from the interaction between the exponential and cosine terms, creating a complex multimodal surface that tests an optimizer's ability to escape flat regions while maintaining precision during final convergence. The standard 2D version contains numerous shallow local minima surroundingx the global optimum, making it particularly challenging for algorithms to navigate between coarse and fine-grained search patterns.

### E.9 ROSENBROCK FUNCTION (3D)

The 3-dimensional Rosenbrock function (Zimmermann, 1979) is a classic benchmark for evaluating optimization algorithms in non-convex spaces with challenging geometric properties. The function takes the form:

$$f(\mathbf{x}) = \sum_{i=1}^{2}\left[100(x_{i+1} - x_i^2)^2 + (1 - x_i)^2\right]$$

where $\mathbf{x} = [x_1, x_2, x_3]$ is the input vector within the search space $x_i \in [-5, 10]$. The function has a global minimum of 0 at $x^* = [1, 1, 1]$. Its landscape features a parabolic-shaped valley with a flat bottom containing the global minimum, surrounded by steep walls that create challenging optimization dynamics. The 3D version exhibits the characteristic difficulties of higher-dimensional Rosenbrock functions, including gradient direction oscillations in the valley region and deceptive local optima that can trap optimization algorithms. The function tests an optimizer's ability to navigate long, narrow, and flat-bottomed valleys while maintaining convergence precision, with the difficulty increasing exponentially in higher dimensions due to the emergence of $2^{n-1}$ local minima for $n \geq 4$.

# F  BASELINES

In our Reasoning BO approach, we primarily focus on the acquisition function design by leveraging prior knowledge and LLMs' reasoning capabilities. Therefore, our baseline selection mainly considers variations of acquisition functions, particularly the LogEI family. Additionally, we include Covariance Matrix Adaptation Evolution Strategy (CMA-ES) as a special numerical optimization baseline. CMA-ES is a stochastic, derivative-free optimization method for nonlinear or non-convex continuous problems, belonging to the class of evolutionary algorithms that mimic biological evolution principles through mutation and selection mechanisms.

## F.1  LOGEI FAMILY

The LogEI family addresses numerical stability issues in traditional Expected Improvement (EI) by operating in log-space. While standard EI implementations can suffer from numerical underflow when improvement probabilities are small, LogEI transformations enable stable computation across the full range of possible inputs (Ament et al., 2023). The key transformation is given by:

$$\text{LogEI}_{y^*}(\mathbf{x}) = \texttt{log\_h}((\mu(\mathbf{x}) - y^*)/\sigma(\mathbf{x})) + \texttt{log}(\sigma(\mathbf{x}))$$

where $y^*$ is the current best observation value, $\mu(\mathbf{x})$ is the predicted mean, and $\sigma(\mathbf{x})$ is the predicted deviation.

### F.1.1  ANALYTIC LOGEI

Analytic LogEI provides a numerically robust implementation through piecewise decomposition:

$$\texttt{log\_h}(z) = \begin{cases} \texttt{log}(\phi(z) + z\Phi(z)) & z > -1 \\ -z^2/2 - c_1 + \texttt{log1mexp}(\texttt{log}(\texttt{erfcx}(-z/\sqrt{2})|z|) + c_2) & -1/\sqrt{\epsilon} < z \le -1 \\ -z^2/2 - c_1 - 2\texttt{log}(|z|) & z \le -1/\sqrt{\epsilon} \end{cases}$$

where $c_1 = \log(2\pi)/2$, $c_2 = \log(\pi/2)/2$, $\epsilon$ is the numerical precision, $\phi$ is the standard normal PDF, $\Phi$ is the standard normal CDF, $\texttt{log1mexp}$ is a numerically stable implementation of $\log(1 - \exp(z))$, and $\texttt{erfcx}$ is a numerically stable implementation of $\exp(z^2)\text{erfc}(z)$.

### F.1.2  MONTE CARLO PARALLEL LOGEI

For parallel batch optimization, Monte Carlo LogEI extends the stability benefits through:

$$q\text{LogEI}_{y^*}(\mathbf{X}) \approx \log \text{sumexp}_i \left( \tau_{\max} \log \text{sumexp}_j \left( \log \text{softplus}_{\tau_0}(\xi^i(\mathbf{x}_j) - y^*) \right) / \tau_{\max} \right)$$

where $i$ indexes Monte Carlo draws from the GP posterior, $j = 1, \ldots, q$ indexes candidates in the batch, $\tau_0$ and $\tau_{\max}$ are temperature parameters controlling the approximation quality, and $\texttt{logsoftplus}$ is a numerically stable implementation of $\log(\log(1 + \exp(z)))$.

### F.2 CMA-ES

CMA-ES (Covariance Matrix Adaptation Evolution Strategy) is an evolutionary algorithm for continuous nonlinear optimization that adapts both the mean and covariance matrix of its search distribution. The method combines maximum-likelihood principles with evolution path tracking, making it particularly effective for complex, non-convex optimization problems (Hansen, 2016).

The algorithm updates its parameters through likelihood maximization similar to expectation-maximization. The mean vector update maximizes:

$$m_{k+1} = \arg\max_m \sum_{i=1}^{\mu} w_i \log p_{\mathcal{N}}(x_{i:\lambda} \mid m)$$

where $\log p_{\mathcal{N}}(x)$ is the log-likelihood from a multivariate normal distribution:

$$\log p_{\mathcal{N}}(x) = -\frac{1}{2}\log\det(2\pi C) - \frac{1}{2}(x-m)^T C^{-1}(x-m)$$

The covariance matrix update follows:

$$\sum_{i=1}^{\mu} w_i \frac{x_{i:\lambda}-m_k}{\sigma_k}\left(\frac{x_{i:\lambda}-m_k}{\sigma_k}\right)^T = \arg\max_C \sum_{i=1}^{\mu} w_i \log p_{\mathcal{N}}\left(\frac{x_{i:\lambda}-m_k}{\sigma_k} \mid C\right)$$

These updates enable CMA-ES to learn a second-order model of the objective function while requiring only solution rankings rather than derivatives. The method has demonstrated superior performance to Bayesian optimization in certain hyperparameter tuning tasks (Loshchilov & Hutter, 2016). The algorithm's effectiveness stems from several distinctive characteristics: its adaptive step-size control mechanism prevents premature convergence, while the updates can be interpreted as natural gradient descent. Furthermore, CMA-ES performs principal component analysis on successful search steps to guide the optimization trajectory, and exhibits notable robustness when handling ill-conditioned problems.

## G  SYSTEM MESSAGES AND PROMPTS TEMPLATES

### G.1  EXPERIMENT COMPASS OBJECT

The Experiment Compass serves as the standardized input template for our algorithm, enabling the description of complex objectives in natural language. It incorporates prior knowledge via the "description" field, aiding LLMs in generating overviews and initial experimental points. Additionally, it standardizes variable names for insights generation, ensuring consistency across multiple rounds of experiments. The template below illustrates the structure of the Experiment Compass.

---

**Prompt Template for Generating the Compass Object**

- **name**: Experiment Name (e.g., Nanoparticle Synthesis Optimization)
- **application_context**: Domain/Use Case (e.g., Chemical Engineering/ML Hyperparameter Tuning)

---

- **description**: Any details or context about the experiment to optimize (approx. 100 words)

- **constraint**: Any constraints on the input space that must be satisfied during the optimization.

- **parameter_definitions**: A list of parameter objects, where each object has:

  - `display_name`: Name of the parameter (e.g., Temp)
  - `description`: Description of the parameter
  - `data_type`: continuous — discrete — categorical
  - `step`: Discretization step (if discrete)
  - `bounds`: Bounds of the parameters

- **target**: An object defining the optimization goal:

  - `name`: Name of the target (e.g., Yield)
  - `description`: Description of the target
  - `direction`: maximize/minimize

As an example, the complete Experiment Compass used for the Suzuki experiment is shown below.

---

**Example Experiment Compass of Suzuki Reaction**

- **name**: Suzuki Reaction Optimization

- **application_context**: Organic Chemistry/Cross Coupling Reaction Optimization

- **description**: Optimization of Suzuki-Miyaura cross-coupling reaction conditions to maximize product yield through systematic exploration of electrophile-nucleophile combinations, ligand selection, base types, and solvent effects. The reaction involves palladium-catalyzed coupling between aryl halides and arylboronic acids.

- **constraint**: Reagents must be chemically compatible and reaction conditions must maintain intermediate stability. All combinations must form stable palladium complexes during catalytic cycle.

- **parameter_definitions**:

  - * **display_name**: `Electrophile_SMILES`
    * **description**: Aryl halide substrate for coupling (SMILES representation)
    * **data_type**: categorical
    * **bounds**:
      · `BrC1=CC=C(N=CC=C2)C2=C1`
      · `ClC1=CC=C(N=CC=C2)C2=C1`
      · `IC1=CC=C(N=CC=C2)C2=C1`
      · `O=S(OC1=CC=C(N=CC=C2)C2=C1)(C(F)(F)F)=O`

- * **display_name**: `Nucleophile_SMILES`
  * **description**: Boronic acid nucleophile (SMILES representation)
  * **data_type**: categorical
  * **bounds**:
    · `CC1=CC=C(N(C2CCCCO2)N=C3)C3=C1[B-](F)(F)F`
    · `CC1=CC=C(N(C2CCCCO2)N=C3)C3=C1B(O)O`
    · `CC1=CC=C(N(C2CCCCO2)N=C3)C3=C1B4OC(C)(C)C(C)(C)O4`
- * **display_name**: `Ligand_SMILES`
  * **description**: Phosphine ligand for palladium catalyst (SMILES)
  * **data_type**: categorical
  * **bounds**:
    · `CC(C)(C)P(C(C)(C)C)C1=CC=C(N(C)C)C=C1`
    · `CC(C)(P(C(C)(C)C)[c-]1cccc1)C.CC(C)(P(C(C)(C)C)[c-]2cccc2)C.[Fe+2]`
    · `CC(C1=C(C2=CC=CC=C2P(C3CCCCC3)C4CCCCC4)C(C(C)C)=CC(C(C)C)=C1)C`
    · `CC(P(C(C)(C)C)C(C)(C)C)(C)C`
    · `COC1=CC=CC(OC)=C1C2=C(P(C3CCCCC3)C4CCCCC4)C=CC=C2`
    · `P(C1=CC=CC=C1)(C2=CC=CC=C2)C3=CC=CC=C3`
    · `P(C1CCCCC1)(C2CCCCC2)C3CCCCC3`
- * **display_name**: `Base_SMILES`
  * **description**: Reaction base for transmetallation step (SMILES)
  * **data_type**: categorical
  * **bounds**:
    · `[Cs+].[F-]`
    · `[K+].[OH-]`
    · `[Na+].[OH-]`
    · `CC([O-])C.[Li+]`
    · `CCN(CC)CC`
    · `O=P([O-])([O-])[O-].[K+].[K+].[K+]`
    · `OC([O-])=O.[Na+]`
- * **display_name**: `Solvent_SMILES`
  * **description**: Reaction medium (SMILES representation)
  * **data_type**: categorical
  * **bounds**:
    · `C1COCC1`
    · `CO`
    · `N#CC`
    · `O=CN(C)C`
- **target**:

> – **name**: Yield
> – **description**: Percentage yield of cross-coupled product
> – **direction**: maximize

## G.2 OVERVIEW

The Overview Object is a key component that bridges the Experiment Compass and the initial experimental setup. It uses the "description" field of the Experiment Compass to help LLMs understand the experiment's objectives, constraints, and outcomes.

---

**Prompt Template for Generating the Overview Object**

**Important Notes**

- Do NOT refer to memorized reaction outcomes, prior experimental data, or specific literature results (even if they match the current experiment).
- You MAY use general domain knowledge from {application_context} (e.g., chemical reactivity principles, mechanism patterns) to inform reasoning.

**Experiment Domain**
{application_context}

**Experiment Overview**
{description}

**Optimization Parameters**
The experiment design space is defined by the parameters below, including their bounds and any relevant constraints:

- **Parameters and Bounds**:
  {parameter_definitions}
- **Constraint**:
  {constraint}
- **Target**:
  {target}

**Task**
In approximately 300 words, provide a clear and concise overview of this experiment. Your overview should:

1. Clearly state the experimental goal and its significance in the {application_context} domain
2. Describe the parameter space including data types and bounds of each parameter
3. Explain any constraints that must be maintained during optimization

---

4. Outline how Bayesian Optimization will systematically explore the parameter space to target.

5. Mention how you will assist when BO plateaus by suggesting promising parameter combinations

6. Conclude with the expected outcomes and potential impact of successful optimization

Focus on creating a coherent narrative that connects these elements while maintaining scientific rigor and clarity.

The prompt template shown above includes the experiment's goals, variables, and constraints. This Overview enhances initial sampling quality and improves the surrogate model's accuracy in Bayesian Optimization.

### G.3 INSIGHTS OBJECT

The Insights Object plays a crucial role during the experimental optimization process. Generated by LLMs, it encapsulates key elements such as comments, keywords, hypotheses, and candidates. As the experiment progresses, LLMs dynamically adjust the confidence levels of hypotheses based on accumulating evidence. These adjustments guide the Bayesian Optimization process by refining sampling strategies according to the insights provided.

> **Template for Generating the Insights Object**
>
> - **comment**: A brief comment on the initial sampling strategy and rationale
> - **keywords**: keyword1, keyword2, keyword3
> - **hypotheses**:
>   - * **strategy**: Descriptive name of exploration approach
>     * **rationale**:Scientific justification for this approach
>     * **confidence**:high/medium/low
>     * **parameter_sets**:
>       · { **ParameterA**: 0.1, **ParameterB**: 0.9, ... }
>       · { **ParameterA**: 0.4, **ParameterB**: 0.6, ... }

The template above illustrates the structure used for generating the Insights Object. It provides a structured format for LLMs to produce actionable guidance.

> **Example Insights from the Suzuki Experiment**
>
> - **comment**: From the experimental data, it's evident that lower concentrations and ambient temperatures are more effective in achieving higher yields. The highest yielding trial was at a concentration of 0.057 mol/L

and temperature of 90C, indicating that lower concentrations and ambient temperatures might be beneficial.

- **keywords**:concentration, temperature, ligand, yield, optimization

- **hypotheses**:
  - * **strategy**: Optimization of Concentration and Temperature
    * **rationale**:Based on the highest yielding trials, we hypothesize that an optimal concentration of 0.057 mol/L and temperature of 90C could be achieved by fine-tuning these parameters within their bounds.
    * **confidence**: high
    * **parameter_sets**:
      · { **Concentration**: 0.057, **Temp_C**: 90,
      **Base_SMILES**: O=C([O-])C(C)(C)C.[Cs+]
      **Ligand_SMILES**: C[C@]1(O2)O[C@](C[C@]2(C)P3C4
      =CC=CC=C4)(C)O[C@]3(C)C1
      **Solvent_SMILES**: CC(N(C)C)=O }
  - * **strategy**: Exploration of Diverse Ligands
    * **rationale**: To further enhance the reaction yield, we will explore different ligands to identify the effective one.
    * **confidence**: high
    * **parameter_sets**:
      · { **Concentration**: 0.057, **Temp_C**: 90,
      **Base_SMILES**: O=C([O-])C(C)(C)C.[Cs+]
      **Ligand_SMILES**: CN(C)C1=CC=CC(N(C)C)=C1C2=CC
      =CC=C2P(C(C)(C)C)C3=CC=CC=C3
      **Solvent_SMILES**: CC1=CC=C(C)C=C1 }
      · { **Concentration**: 0.057, **Temp_C**: 90,
      **Base_SMILES**: O=C([O-])C(C)(C)C.[Cs+]
      **Ligand_SMILES**: CP(C)C1=CC=CC=C1
      **Solvent_SMILES**: CC(N(C)C)=O }
  - * **strategy**: Refinement of high Concentration and Elevated Temperature
    * **rationale**: While the highest yield was seen at high concentration and elevated temperature, we hypothesize that there might be an optimal point beyond which yields decrease due to side reactions or instability.
    * **confidence**: medium
    * **parameter_sets**:
      · { **Concentration**: 0.153, **Temp_C**: 110,
      **Base_SMILES**: O=C([O-])C(C)(C)C.[K+]
      **Ligand_SMILES**: C[C@]1(O2)O[C@](C[C@]2(C)P3C4
      =CC=CC=C4)(C)O[C@]3(C)C1
      **Solvent_SMILES**: CC(N(C)C)=O }

The example above displays Insights generated during the Suzuki reaction experiment, showcasing how hypotheses and candidates evolve over iterations.

### G.4 OPTIMIZATION

The optimization process in Reasoning BO involves two distinct phases: initialization and iterative refinement. These phases are guided by specific prompts designed to leverage the capabilities of LLMs effectively.

#### G.4.1 INITIALIZATION PROMPTS

During the initialization phase, the system generates an initial Insights Object using the Overview and Experiment Compass. This object guides the first round of experiments. The specific prompt template for this phase is shown below.

---

**Prompt Template for Initialization Phase**

You are assisting with the critical initial sampling phase of Bayesian Optimization. The quality of these initial hypotheses significantly impacts the optimization efficiency.

**Task**

Generate `n_hypotheses` diverse and well-justified initial hypotheses for exploring the optimization target based on:

- Domain: {`application_context`}
- Experiment Goal: {`description`}
- Target: {`target`}
- Constraints: {`constraint`}

**Instructions**

- Do NOT refer to memorized reaction outcomes, prior experimental data, or specific literature results (even if they match the current experiment).
- Base all reasoning strictly on the given input, parameter definitions, constraints, and optimization objective.
- For each hypothesis, provide at least one point that satisfies the constraint.

**Format Requirement**

Your response must be in the exact format shown below. No additional commentary, explanations, or syntax should be included.

**Example Output (for format only, unrelated to this experiment):**

- **comment**: A brief comment on the initial sampling strategy and rationale ( 100 words).
- **keywords**: keyword1, keyword2, keyword3
- **hypotheses**:
    - ∗ **strategy**: Descriptive name of exploration approach

---

      ∗ **rationale**: Scientific justification for this approach

      ∗ **confidence**: high/medium/low

      ∗ **parameter_sets**:

       · { **ParameterA**: 0.1, **ParameterB**: 0.9, ... }

       · { **ParameterA**: 0.4, **ParameterB**: 0.6, ... }

    – ∗ **strategy**: Descriptive name of exploration approach

      ∗ **rationale**: ...

      ∗ **confidence**: high/medium/low

      ∗ **parameter_sets**:

       · { **ParameterA**: 0.3, **ParameterB**: 0.8, ... }

       · { **ParameterA**: 0.1, **ParameterB**: 0.3, ... }

**Important Notes**

- Points must include all required parameters.

- Hypothesis "names" and "parameter" keys must match exactly with the experiment definition.(if they are discrete values, they must be returned strictly according to the format).

- All parameter names and values must **exactly match** predefined options in spelling, casing, and formatting - no substitutions, abbreviations, or modifications permitted.(if they are discrete values, they must be returned strictly according to the format).

- Any deviation from predefined parameters will result in automatic rejection, regardless of semantic similarity or logical equivalence.

- Parameter specs for this experiment: {`parameter_definitions`}

### G.4.2 OPTIMIZATION LOOP PROMPTS

In the optimization loop, the LLM-based agent receives the Experiment Compass and the most up-to-date version of the Insights Object. Based on this information, the LLM generates new hypotheses and updates the Insights Object. The specific prompt template for this phase is shown below.

**Prompt Template for Optimization Loop Phase**

**Optimization Progress**
Bayesian Optimization is in progress at iteration {`iteration`}.
**Objective**
{`target`}
**Retrieved message**
{`retrieved_context`}
**Historical Summary**
You generated the following comments and hypotheses in previous iterations:
{`insight_history`}

Your suggested points were evaluated, and the results have been added to the dataset. We now have data from {iteration} completed experiments: {trial_data}

**Bayesian Optimizer Suggestions**

Here are the candidate points recommended by the optimizer in this round: {bo_recommendations}

**Your Task**

Reflect on the current state of the optimization. Do not rely on prior assumptionsreason, only based on the data provided.

- What trends or patterns emerge from the experimental data?
- Which regions in the parameter space seem most effective in achieving the target?
- Which areas consistently underperform?
- Improve or revise your previous hypotheses based on observed outcomes.
- You may discard low-performing hypotheses and propose new ones.
- For each hypothesis, **at least provide one point** that satisfies the constraint: {constraint}.

**Example Output Format for two hypotheses (template only, unrelated to current experiment):**

- **comment**: "A 200-word summary of optimization progress, insights, and directions.",
- **keywords**: "keyword1, keyword2, keyword3",
- **hypotheses**: [
  - * **strategy**: "Descriptive name of exploration approach",
    * **rationale**: "Scientific justification for this approach",
    * **confidence**: "high/medium/low",
    * **parameter_sets**: [
      · { **ParameterA**: 0.1, **ParameterB**: 0.9, ... },
      · { **ParameterA**: 0.4, **ParameterB**: 0.6, ... }
  - * **strategy**: "Descriptive name of exploration approach",
    * **rationale**: "...",
    * **confidence**: "high/medium/low",
    * **parameter_sets**: [
      · { **ParameterA**: 0.3, **ParameterB**: 0.8, ... },
      · { **ParameterA**: 0.1, **ParameterB**: 0.3, ... }

**Important**

- Your output must be valid pure JSON without "'json or any other formatting, start with {{ and end with }}.
- Never put comments or text outside the JSON structure.

- Do NOT wrap it in markdown, text, or explanations.

- Do NOT return fewer than 3 nor more than 5 hypotheses.

- All parameter names and values must **exactly match** predefined options in spelling, casing, and formatting - no substitutions, abbreviations, or modifications permitted.(if they are discrete values, they must be returned strictly according to the format).

- Any deviation from predefined parameters will result in automatic rejection, regardless of semantic similarity or logical equivalence.

- Parameter specs for this experiment: {`parameter_definitions`}

### G.5 EXPERIMENT SUMMARY AND CONCLUSION

The Experiment Summary and Conclusion phases in Reasoning BO synthesize insights from the entire optimization process, leveraging accumulated Insight History and experimental data.

**Experiment Summary**

The Experiment Summary evaluates hypothesis validity strictly based on experimental data, providing a concise interim conclusion. This phase focuses on how hypotheses evolved, which were supported or refuted, significant shifts in confidence levels, and key conclusions. A table summarizing the evolution of confidence in hypotheses concludes this section. The prompt template used to generate this summary is shown below, followed by a concrete example of the summary generated for the Suzuki reaction experiment.

---

**Prompt Template for Generating the Experiment Summary**

**Summary of Optimization Process**
{`insight_history`}
We have now completed the optimization process. Below is the final dataset from {`iteration`} experiments: {`trial_data`}
**Your Task**
Based strictly on the experimental data and the hypotheses generated throughout the optimization (do not rely on prior knowledge or assumptions), write a concise summary ( 300 words) addressing the following:

- How did your hypotheses evolve over time?

- Which hypotheses were consistently supported or refuted by data?

- What were the most significant shifts in your confidence levels?

- What conclusions can be drawn from the experiment?

End it with a table summary showing how your confidence in the hypotheses evoved during the optimization.

---

**Example Experiment Summary for Suzuki Reaction**

**Summary of Hypothesis Evolution**

The optimization began with orthogonal screening of electrophile leaving groups, nucleophile activation modes, ligand steric, base strength, and solvent polarity. Initial data (trials 1-2) revealed the critical superiority of the triflate electrophile (52.36% vs. bromide/chloride<45%), shifting focus to electrophile centric probes. Hypotheses evolved to exploit triflate's leaving group advantage through synergistic pairing with activated nucleophiles (trifluoroborate, 74.65% yield), ferrocene ligands, and phosphate bases (K3PO4).

**Supported Hypotheses**

1. **Triflate electrophile superiority**: Consistently outperformed halides (3-10x higher yields), validated across all trials.

2. **Trifluoroborate activation**: Achieved peak yield (74.65%) with K3PO4, outperforming boronic acids/pinacol esters (<44.08%).

3. **Ferrocene ligand efficacy**: Outperformed bulky phosphines (e.g., 74.65% vs. 15.15-30.48%), balancing Pd stabilization and transmetallation accessibility.

**Refuted Hypotheses**

1. **Bulky phosphine ligands**: Consistently underperformed (<30.48%), indicating steric hindrance outweighs stabilization benefits.

2. **Polar solvents (DMF/NMP)**: Reduced yields by>50% vs. dioxane, contradicting initial assumptions about transmetallation stabilization.

3. **Chloride/bromide systems**: Persistently low yields (<45%) led to their exclusion in later stages.

**Confidence Shifts**

- **Increased confidence** in triflate/trifluoroborate/K3PO4/ferrocene synergy after repeated validation (trial 1: 74.65%, trial 14: sustained dominance).

- **Decreased confidence** in ligand steric bulk (high-low) and solvent polarity effects (medium-low) due to consistent underperformance.

**Conclusions**

The optimal system combines triflate's leaving group ability, trifluoroborate's transmetallation efficiency, K3PO4's activation, ferrocene's balanced coordination, and dioxane's solvent compatibility. Key factors include avoiding steric hindrance (ligands), leveraging phosphate activation (base), and prioritizing oxidative addition kinetics (electrophile).

**Confidence Evolution Table**

| Hypothesis | Initial Confidence | Final Confidence | Trend |
|---|---|---|---|
| Triflate electrophile | High | High | - |
| Trifluoroborate activation | Medium | High | ↑ |
| Ferrocene ligand efficacy | High | High | - |
| Bulky phosphine ligands | High | Low | ↓ |
| Polar solvent benefits | High | Low | ↓ |
| K3PO4 base superiority | Medium | High | ↑ |
| CsF base utility | Medium | Medium | - |

Key: ↑ = Increased confidence, ↓ = Decreased confidence, - = Stable

## Conclusion

The Conclusion phase synthesizes the Experiment Summary and the entire optimization process to provide a comprehensive, professional report. It includes key outcomes, an experimental retrospective, milestones achieved, definitive findings, forward guidance, and scientific impact. This report adheres to a structured format with numbered sections and maintains an academic tone throughout. The prompt template used to generate the conclusion, and a concrete example for the Suzuki reaction, are shown below.

---

**Prompt Template for Generating the Conclusion Report**

### Final Report on Optimization

YOUR PREVIOUS INSIGHTS

`{insight_history}`

EXPERIMENTS DATA

`{trial_data}`

EXPERIMENT TARGET

`{target}`

Generate a polished, professional report summarizing the complete optimization process using only the experimental data and documented hypotheses.

1. **Key Outcomes**

    2-4 sentence executive summary highlighting:

    - Best achieved result for target
    - Most influential parameters
    - Optimization efficiency

2. **Experimental Retrospective**

    - Objective: target in `{application_context}`
    - Initial Approach: Summary of starting hypotheses from insight_history

3. **Optimization Journey**

    Present key milestones in this format:

---

"Between iterations X-Y: [development description]. Confidence changed from [level] to [level] (Evidence: Trials #A,B,C)."

4. **Definitive Findings**

   - Optimal Configuration: best_parameters (achieved target of best_value)
   - Parameter Relationships:

Parameter1 : [trend description]
Parameter2 : [response pattern]

5. **Forward Guidance**

   - Immediate Recommendations:
     (a) Specific actionable suggestion
     (b) Alternative approach
   - Long-term Considerations:
     (a) Constraint modification
     (b) New parameter suggestions

6. **Scientific Impact**
   Concise discussion connecting findings to:

   - {application_context} principles
   - Theoretical implications
   - Practical applications

**Format Requirements**

- Use numbered sections
- Reference specific trials (e.g., "Trial #X")
- Maintain academic tone
- 500-600 words total
- Plain text only (no markdown formatting)

---

**Example Conclusion Report for Suzuki Reaction**

**Conclusion**

1. **Key Outcomes**
   The Suzuki cross-coupling optimization achieved a maximum yield of 74.65% (Trial #1_2) through systematic parameter refinement. The triflate electrophile reactivity and K3PO4-driven trifluoroborate nucleophile activation proved most influential, with ferrocene ligands providing optimal Pd stabilization. The elimination of underperforming parameters (bulky phosphine ligands, polar solvents, chloride/bromide electrophiles) improved optimization efficiency, enabling focused exploration of the high-yield design space within 15 iterations.

2. **Experimental Retrospective**

- **Objective**: Maximize yield in a Suzuki-Miyaura coupling between heteroaromatic electrophiles and boron-containing nucleophiles.
- **Initial Approach**: Orthogonal screening of leaving groups (Br-/Cl/OTf), boron species (acid/trifluoroborate/pinacol ester), ligand sterics (ferrocene/trialkylphosphines), base strength (CsF/K3PO4/KOH), and solvent polarity (dioxane/DMF/NMP) (Trials #0_0-0_2). Early hypotheses prioritized oxidative addition kinetics and transmetallation compatibility.

3. **Optimization Journey**

Between iterations 1-3: Triflate electrophile (OTf) demonstrated 2.3x higher yield vs bromide (52.32% vs 23.53%, Trial #0_1 vs #0_0), shifting focus to triflate systems. Confidence in leaving group strategy rose from medium to high.

Between iterations 4-7: K3PO4 base with trifluoroborate nucleophile yielded 74.65% (Trial #1_2), revealing critical base-nucleophile synergy. Bulky phosphine ligand confidence dropped from high to low as 15-30.48% yields. (Trials #1_1 vs #1_0).

Between iterations 8-12: DMF solvent underperformance (43.66%, Trial #3_1) solidified dioxane preference despite polarity hypotheses. Ferrocene ligand confidence remained high throughout (62.71-74.65%, Trials #0_0, #1_2).

4. **Definitive Findings**

- **Optimal Configuration**:
  - **Electrophile**: `B-S(C1=CC(=CC=C2C1=C(C(F)(F)F)=O) C2=O)NC3=CC=NC=C3`
  - **Nucleophile**: `CC1=CC=C(N(C)C2=CC=C(B(O)(O))C=C2) C=C1`
  - **Ligand**: `Ferrocene diphosphine`
  - **Base**: `K3PO4`
  - **Solvent**: `Dioxane`
  
  (Yield: 74.65%, Trial #1_2)
- **Parameter Relationships**:
  - Electrophile: Triflate > Bromide > Chloride (52.32-74.65% vs 23.53-44.98%, Trials #0_1 vs #0_0/#0_2)
  - Ligands: Ferrocene outscored bulky phosphines by >2x yield margin (74.65% vs 15.15-30.48%, Trial #1_2 vs #1_1/#2_0)
  - Solvent: Inverse correlation between polarity and yield (dioxane 74.65% vs DMF 43.66%, Trial #1_2 vs #3_1)

5. **Forward Guidance**

- **Immediate Recommendations**:
  (a) Test tert-butyl-modified ferrocene ligands (Proposed Trial #11_0) to balance electron donation/Pd accessibility
  (b) Explore dioxane/DMF solvent gradients (90:10 50:50 v/v) to probe transmetallation stabilization

- **Long-term Considerations**:
  - Evaluate microwave-assisted conditions to accelerate oxidative addition
  - Screen N-heterocyclic carbene ligands for enhanced Pd(0)/Pd(II) cycling

6. **Scientific Impact**

The results validate three Suzuki reaction design principles: 1) Triflates' superior leaving group ability accelerates oxidative addition despite steric bulk; 2) Phosphate bases enable simultaneous trifluoroborate activation and Pd intermediate stabilization, and 3) Ferrocene ligands provide optimal accepting capacity without transmetallation steric hindrance. Practically, this work establishes a template for coupling electron-deficient heteroaromatics a persistent challenge in pharmaceutical synthesis. The systematic exclusion of underperforming parameters (Trials #0_0, #13_0) demonstrates efficient design space navigation, suggesting Bayesian optimization effectively replaces traditional one-variable-at-a-time approaches in complex catalytic systems.

G.6 EXTRACT NOTES PROMPTS

The following prompts are used to extract structured notes from both the Experiment Compass and reasoning data.

**Extracting Notes from Experiment Compass**

**Prompt Template for Extracting Notes from Experiment Compass**

Extract and structure scientific notes from the following experimental setup.
**Guidelines:**

- Use bullet-point style strings in each list.
- Base your output only on the given input.
- Do not fabricate information not present in the experiment.

**Focus on the following aspects:**

1. **Theoretical background** – Scientific principles, hypotheses, or mechanisms implied or explicitly stated.

2. **Variable properties** – Roles, attributes, or behaviors of each variable/component involved.

3. **Variable relationships** – Any cause-effect, dependency, or correlation observed or described.

4. **General notes** – Additional relevant scientific information or implicit assumptions.

**Experimental setup:** {Experiment Compass}

**Extracting Notes from Reasoning Data**

---
**Prompt Template for Extracting Notes from Reasoning Data**

Analyze the following reasoning data and extract structured scientific notes.
**Guidelines:**

- Each item should be a clear and self-contained bullet point.
- Only use information present in the reasoning. Do not speculate.
- Keep the content factual, concise, and helpful.

**Focus on the following aspects:**

1. **Key findings** – Verifiable results, conclusions, or observations grounded in reasoning.

2. **Parameter relationships** – How different variables or conditions affect each other (cause-effect, correlation).

3. **Optimization principles** – Rules or strategies suggested by the reasoning that could improve experimental results.

4. **General notes** – Summarized insights that don't fall into the above categories.

**Reasoning data:** {Reasoning data}

---

**Example: Notes Extraction and Storage for Suzuki Reaction**

---
**Example: Notes Extraction and Storage for Suzuki Reaction**

- **Key findings**:
  - The highest yields were obtained using the sulfone electrophile with CsF base, DMF solvent, and specific ligands such as cyclohexyl and biaryl.

- **Parameter relationships**:
  - Sulfone and iodine electrophiles perform best with CsF base and DMF solvent.
  - Bulky ligands like cyclohexyl and biaryl enhance yields.

- **Optimization principles**:
  - Focus on refining the cyclohexyl ligand variants for sulfone,
  - testing biaryl ligand with sulfone and acetone,
  - and exploring modified biaryl ligand structures.

- **General notes**:
  - Avoid underperforming electrophiles (bromine, chlorine) and bases (KOH, NaOH).
  - Ensure all parameters are from predefined lists and use CsF base for all hypotheses.

- **Created knowledge triples**:

---

```
        – (Sulfone Electrophile, PerformsBestWith, CsF
          Base)
        – (Sulfone Electrophile, PerformsBestWith, DMF
          Solvent)
        – (Iodine Electrophile, PerformsBestWith, CsF
          Base)
        – (Iodine Electrophile, PerformsBestWith, DMF
          Solvent)
        – (Cyclohexyl Ligand, EnhancesYieldsWith,
          Sulfone Electrophile)
        – (Biaryl Ligand, EnhancesYieldsWith, Sulfone
          Electrophile)
        – (Biaryl Ligand, ShouldBeTestedWith, Acetone)
```

G.7 CUSTOM NOTES SCHEMA DESIGN

To accommodate the diverse forms of knowledge across different downstream domains, such as SMILES expressions in chemistry, we provide a customizable interface for defining the schema of notes. By specifying the schema, users can constrain and structure the format of the extracted notes to fit their specific needs.

In our framework, the schema definition is based on two primary classes: ReasoningNotesResponse and CompassNotesResponse. These classes are designed to capture structured information from reasoning data and experimental descriptions, respectively. The definitions of these classes are shown below.

---

**Schema Definition for Reasoning Notes Response**

```
class ReasoningNotesResponse(BaseModel):
```
   *Structured notes extracted from reasoning data*

- **notes**: `List[str]`
  - **description**: May include summaries of findings, relationships, or principles.
- **key_findings**: `List[str]`
  - **description**: Important factual discoveries or observations drawn from the reasoning process.
- **parameter_relationships**: `List[str]`
  - **description**: Descriptive cause-effect or correlation relationships between different parameters or variables.
- **optimization_principles**: `List[str]`
  - **description**: Actionable principles or rules that can help optimize the experiment. Should be verifiable and clearly beneficial.

---

**Schema Definition for Compass Notes Response**

```
class CompassNotesResponse(BaseModel):
```

> *General-purpose response format for structured scientific notes. Suitable for notes extracted from any experimental description or setup.*

- **notes**: `List[str]`
  - **description**: Free-form bullet-point notes. General scientific facts, summaries, or highlights.
- **theoretical_background**: `List[str]`
  - **description**: Scientific principles, mechanisms, or theories underlying the experiment.
- **variable_properties**: `List[str]`
  - **description**: Descriptions of individual variables' attributes, roles, or behaviors.
- **variable_relationships**: `List[str]`
  - **description**: Cause-effect or correlated relationships between different variables or parameters.

---

To better illustrate our algorithm's workflow, we select Buchwald-Hartwig amination as a case study. This palladium-catalyzed C-N cross-coupling reaction between aryl halides and amines presents a challenging optimization problem with categorical parameters (ligands, bases, additives, and aryl halides) while being widely applied in pharmaceutical synthesis. The constrained combinatorial space makes it an ideal testbed for evaluating optimization frameworks.

## H    USAGE OF LARGE LANGUAGE MODELS IN THIS MANUSCRIPT

In preparing this manuscript, we used a large language model (LLM) solely for editorial purposes. Its functions were limited to proofreading for typographical errors, correcting grammatical mistakes, and enhancing the clarity and readability of the text.

## I    LIMITATIONS AND FUTURE WORK

### I.1    CONTEXT WINDOW CONSTRAINTS

The context window limitation has been a fundamental constraint since the initial adoption of LLMs, remaining a critical factor in assessing model capabilities. While our algorithm employs a single-turn interaction paradigm that avoids the memory burden of traditional multi-turn dialogues, the accumulation of historical insights and experimental data across optimization iterations may eventually approach context window limits. Although our current experiments have not encountered this boundary, more complex or prolonged optimization campaigns could face increased computational costs due to context expansion. Future developments should investigate more efficient memory-sharing architectures and attention mech-

anisms to enable sustainable long-horizon optimization workflows. Additionally, multi-agent framework designs could offer more flexible sampling strategies, while early stopping mechanisms may help mitigate context inflation.

### I.2 INSTRUCTION-FOLLOWING CAPABILITY DEPENDENCIES

While fine-tuning and reinforcement learning can enhance a model's zero-shot warmstartting and in-context learning (ICL) capabilities, single-task optimization may inadvertently reduce the model's generalization and instruction-following abilities (Yang et al., 2024). Excessive task-specific adaptation risks overfitting the model to particular formats, potentially compromising its adaptability to novel tasks. In our framework, the Insight Object serves as the critical interface between LLMs and Bayesian optimization, with strict formatting requirements enforced through prompt engineering and regular expression parsing. A valid Insight Object must satisfy three criteria:

- Strict adherence to the predefined Insight format specification
- Generation of valid hypotheses and candidate points that comply with all defined constraints
- Absence of extraneous annotations or markers

Empirical observations reveal that smaller models (3B-7B parameters) frequently fail to maintain consistent format compliance. This limitation persists during reinforcement learning, where we observe significant degradation in foundational instruction comprehension even when employing anti-catastrophic forgetting techniques like LoRA. For instance, our experiments with Qwen-2.5 (7B)(Yang et al., 2025) demonstrate substantial deterioration in instruction adherence post-adaptation, necessitating the use of larger 14B models. Although real-world applications may tolerate minor output deviations compared to benchmark conditions, the observed declines in base capabilities and generalization remain concerning. Developing adaptation paradigms that preserve model versatility represents a crucial research direction for automated Bayesian experimentation, with potential to significantly lower the adoption barrier for LLM-driven optimization in science discovery.

### I.3 MULTI-OBJECTIVE OPTIMIZATION POTENTIAL

Our framework currently handles single-objective optimization, but LLMs show promise for multi-objective scenarios. Their natural language capabilities could help navigate trade-offs between competing objectives like yield and cost (Shields et al., 2021). Future work should explore how LLMs can model these relationships and explain optimization decisions.

