# OpenReview forum: "Reasoning BO: Enhancing Bayesian Optimization with the Long-Context Reasoning Power of LLMs"
_ICLR.cc/2026/Conference — Submitted to ICLR 2026_

### Official Review · Reviewer_FLYi · 2025-10-21

**Soundness:** 3
**Presentation:** 3
**Contribution:** 3
**Rating:** 6
**Confidence:** 3

**Summary:**

This paper proposes Reasoning BO, a Bayesian optimization framework that incorporates large language models (LLMs) as reasoning agents to guide hypothesis generation and decision-making during optimization. The approach combines a reasoning model, a knowledge management module, and a confidence-based filtering mechanism to iteratively refine candidate proposals based on experimental feedback.

The method is tested on several real-world chemical reaction yield optimization tasks as well as synthetic benchmarks. Across these settings, Reasoning BO consistently outperforms baselines such as Vanilla BO, CMA-ES, and Analytic EI, showing faster early-stage improvements and stronger overall performance.

**Strengths:**

Originality:
This paper introduces a novel idea: using an agentic reasoning model to augment Bayesian optimization. The approach moves beyond typical surrogate-model or acquisition-function improvements and explores how autonomous reasoning agents can guide scientific experimentation. The integration of natural-language reasoning into the BO loop is novel, conceptually exciting, and timely given current trends in AI-for-science and LLM-based experimental design.


Quality:
The framework is clearly described and experimentally well supported. The authors evaluate on several chemical and synthetic benchmarks, include relevant baselines, and perform ablations to separate the effects of reasoning, initialization, and optimization. The results consistently favor the proposed method.


Clarity:
The paper is easy to follow and well structured. The motivation is clear, the figures are informative, and the system components are explained at an appropriate level of detail. Implementation details and code availability add to transparency.


Significance:
This work represents an important step toward reasoning-enhanced optimization, where autonomous agents use language-based reasoning to guide experimental design. By showing that hypothesis-driven reasoning can improve both BO initialization and search efficiency, the paper opens a promising direction for combining symbolic reasoning, natural-language understanding, and statistical optimization. The approach has clear potential to advance AI-driven scientific discovery and automated experimentation.

**Weaknesses:**

1 (Main Weakness): For the so-called “high-dimensional benchmarks,” such as Ackley (15D) and Lunar Lander (12D), the paper does not include comparisons to state-of-the-art Bayesian optimization methods designed specifically for high-dimensional search spaces. It is well known that Vanilla BO often performs poorly in high dimensions, so including at least one such baseline would be important for a fair evaluation. For example, comparisons to either (a) Vanilla BO with the length-scale priors proposed in “Vanilla Bayesian Optimization Performs Great in High Dimensions” or (b) the trust-region BO method (TuRBO) from “Scalable Global Optimization via Local Bayesian Optimization” would provide a more complete empirical assessment. Ideally, both methods could be included for all tasks with search space dimensionality greater than ~10.


2: The empirical improvements, while consistent, are somewhat marginal on certain tasks. It would the strengthen the paper if the authors provided discussion or insight into why Reasoning BO achieves larger gains on some tasks and smaller gains on others.


3 (Minor): The color palette in Figure 3 uses multiple similar shades, which makes it harder to distinguish curves. Adjusting the colors for better contrast would improve readability.

**Questions:**

1: Could the authors provide an explanation or hypothesis for why Reasoning BO achieves larger improvements on certain tasks but smaller gains on others?



2: For the “high-dimensional benchmarks,” is there a reason the paper does not include comparisons to high-dimensional BO methods such as the trust-region BO approach (TuRBO, "Scalable Global Optimization via Local Bayesian Optimization") or Vanilla BO with length-scale priors ("Vanilla Bayesian Optimization Performs Great in High Dimensions")? If such methods were omitted intentionally, additional context would help clarify the scope of evaluation.


3: On optimization trajectory plots, it's typically to plot the BEST objective value obtained during the entire optimization run so far on the y-axis, and the number of observations done so far on the x-axis. This leads to plots where the curve is strictly increasing (or strictly decreasing for minimization tasks) since we always plot the BEST thing observed so far. Plots in this paper instead have lines that go up and down (rather than strictly increasing or strictly decreasing) and I'm not sure why that is? Can the authors clarify this and explain why curves go up and down and what exactly is being plotted on the y-axis?

---

> ### Author Response · Authors · 2025-11-20
> **Reply to Reviewer FLYi - Part 1/3**
>
> **Q1：Missing comparisons to high-dimensional BO methods (TuRBO/BO+length-scale strategies)**
>
> A1: Thank you for raising this important point. In the revised version, we have added **TuRBO** and **BO-LS** (Vanilla BO with adaptive length-scales) as high-dimensional baselines on the 15D Ackley benchmark. As shown in the Table, **Reasoning BO continues to outperform both methods**, achieving lower objective values throughout the optimization.
>
> This is particularly valuable because, as the reviewer noted, **TuRBO and BO-LS** perform better than the other baselines included in our original submission, and we appreciate the suggestion to include them for a more complete comparison. We believe Reasoning BO’s advantage arises from its hypothesis-driven candidate selection, which helps it escape early local traps more consistently than purely surrogate-driven methods. In addition, Ackley’s highly structured global optimum allows the LLM to identify promising regions early—an analysis we elaborate on in **Q2**.
>
> As for why these baselines were not included originally: the initial submission focused on reaction-optimization tasks whose dimensionality (≤10D) reflects real wet-lab design spaces, where chemists constrain variables using mechanistic priors such as solvent families, temperature windows, or catalyst classes. Nonetheless, we fully agree that adding high-dimensional BO baselines strengthens the completeness of the evaluation, and we thank the reviewer for the insightful recommendation.
>
> Table : Best-so-far objective values on Ackley-15D
>
> (Reasoning BO values averaged over 10 runs; TuRBO and BO-LS run once.)
>
> | trial_index | reasoning_bo | turbo | BO_LS |
> |-------------|-------------|-------|-------|
> | 0 | 1.08 | 3.40 | 3.13 |
> | 1 | 0.26 | 3.40 | 2.26 |
> | 2 | 0.04 | 3.40 | 2.24 |
> | 3 | 0.02 | 3.23 | 2.23 |
> | 4 | 0.00 | 2.91 | 2.19 |
> | 5 | 0.00 | 2.62 | 2.19 |
> | 6 | 0.00 | 2.58 | 2.19 |
> | 7 | 3.61E-05 | 2.58 | 2.17 |
> | 8 | 1.40E-05 | 2.58 | 2.17 |
> | 9 | 3.89E-06 | 2.58 | 2.17 |
> | 10 | 3.89E-06 | 2.25 | 2.16 |
> | 11 | 3.89E-06 | 1.49 | 2.16 |
> | 12 | 1.71E-06 | 1.49 | 2.16 |
> | 13 | 1.71E-06 | 1.49 | 2.16 |
> | 14 | 1.71E-06 | 1.49 | 2.16 |
> | 15 | 9.12E-07 | 1.17 | 2.16 |
> | 16 | 9.12E-07 | 0.94 | 2.16 |
> | 17 | 9.12E-07 | 0.64 | 2.16 |
> | 18 | 9.12E-07 | 0.64 | 2.16 |
> | 19 | 4.45E-07 | 0.64 | 2.16 |
> | 20 | 1.15E-07 | 0.64 | 2.16 |
> | 21 | 1.14E-07 | 0.64 | 2.16 |
> | 22 | 1.14E-09 | 0.64 | 2.16 |
> | 23 | 1.14E-09 | 0.61 | 2.16 |
> | 24 | 1.14E-10 | 0.61 | 2.16 |
> | 25 | 1.14E-11 | 0.61 | 2.16 |
> | 26 | 1.14E-13 | 0.61 | 2.16 |
> | 27 | 1.14E-13 | 0.49 | 2.15 |
> | 28 | 1.14E-13 | 0.36 | 2.15 |
> | 29 | 1.14E-13 | 0.29 | 2.15 |
>
> **Q2: Why does Reasoning BO achieve larger gains on some tasks but smaller gains on others?**
>
> A2: We sincerely appreciate this insightful question—it goes directly to the core of when and why augmenting BO with LLM-based reasoning yields the greatest improvements. In our analysis, the decisive factor is the degree to which the optimization problem contains meaningful prior structure that an LLM can leverage.
>
> To examine this systematically, we group all benchmarks into two broad categories: **knowledge-driven tasks**, where domain structure is rich, and **non-knowledge-driven tasks**, where no such structure is present.
>
> Knowledge-Driven Tasks
>
> Tasks: Suzuki–Miyaura, Direct Arylation, CPA-catalyzed thiol–imine, Buchwald–Hartwig.
>
> These problems embed substantial chemical structure—reaction mechanisms, reagent compatibility, steric/electronic effects, and known failure modes—that provide priors an LLM can meaningfully reason about. In these settings, mechanistic intuition allows Reasoning BO to avoid poor regions early and identify chemically plausible, high-value conditions.
>
> This becomes particularly evident when comparing **Suzuki** and **Direct Arylation**, two datasets with similar dimensionality but very different difficulty levels. Direct Arylation is markedly harder: its yield landscape is rugged, failure-prone, and dominated by low-performing regions. The quantitative differences make this clear:
>
> | Metric                      | Suzuki  | Direct Arylation | Difference  |
> |----------------------------|---------|------------------|-------------|
> | Mean yield                 | 43.63%  | 19.37%           | ↓24.25%     |
> | Median yield               | 36.43%  | 8.05%            | ↓28.38%     |
> | Yield < 10%                | 8.14%   | 53.30%           | ↑45.15%     |
> | ≥50% high-yield proportion | 41.04%  | 13.83%           | ↓27.21%     |
> | Zero-yield failure rate    | 0.35%   | 28.59%           | ↑28.24%     |

---

> ### Author Response · Authors · 2025-11-20
> **Reply to Reviewer FLYi - Part 2/3**
>
> This contrast is crucial for interpreting the performance difference:
> - **Suzuki is relatively forgiving**: both classical BO and Reasoning BO eventually identify high-yield regions, so the performance gap naturally narrows later in optimization.
> - **Direct Arylation is substantially more challenging**: the search space contains many failure modes and few high-yield regions. Here, classical BO is more easily trapped, while Reasoning BO benefits strongly from mechanistic priors and avoids these traps early.
>
> Thus, the very fact that Direct Arylation is harder explains why Reasoning BO achieves larger gains: when the search landscape is more complex and mechanistically structured, the advantages of LLM reasoning are amplified.
>
> The same trend holds for the CPA and Buchwald–Hartwig datasets (Fig. 4). Across all knowledge-rich tasks, Reasoning BO consistently performs better than on synthetic functions, aligning with our motivation that reasoning-augmented BO is most effective when domain priors matter.
>
> Across all knowledge-rich tasks, Reasoning BO consistently performs better than on synthetic benchmarks, aligning with our motivation that reasoning-augmented BO is most effective when domain priors matter.
>
> ---
>
> **Non-Knowledge-Driven Tasks **
>
> Tasks: Ackley, Rosenbrock, Levy, Hartmann, Lunar Lander.
>
> Here, the LLM cannot draw on any meaningful domain knowledge, so performance depends entirely on the intrinsic structure of the landscape. Nevertheless, two patterns emerge.
>
> For **Ackley, Rosenbrock, and Levy**, Reasoning BO performs surprisingly well because their global minima have simple, highly structured coordinates:
> - Ackley: $x* = (0, 0, ..., 0)$
> - Rosenbrock/Levy: $x* = (1, 1, ..., 1)$
>
> Such regular solutions allow the LLM to “guess” a promising region even without domain knowledge, yielding strong initialization and competitive overall performance—including in higher dimensions (e.g., Ackley-15D).
>
> The situation differs for **Hartmann-6D** and **Lunar Lander**.
>
> - Hartmann’s global optimum is irregular and non-symmetric: (0.20169, 0.15001, 0.47687, 0.27533, 0.31165, 0.65730), making it difficult for any prior-free reasoning to anticipate.
> - Lunar Lander’s reward is discontinuous, complex, and domain-agnostic, offering no meaningful prior structure for an LLM to exploit.
>
> Additionally, in high-dimensional, knowledge-sparse settings, we observe that Reasoning BO tends toward early exploitation and fast convergence—explaining the reviewer RRPq’s observation that Ackley-15D sometimes converges even faster than Ackley-2D. Without priors, the method naturally becomes more conservative as dimensionality increases.
>
> ---
>
> **Conclusion**
>
> Taken together, these results reveal a clear pattern: **Reasoning BO is most effective when the landscape contains meaningful domain structure, mechanistic constraints, or semantically rich priors that an LLM can reason about.** This is precisely the class of problems our method targets—laboratory-scale chemical and biological optimization—where domain knowledge exists but is costly for classical BO to encode manually.
>
> We sincerely thank the reviewer for prompting this deeper analysis. It clarifies not only where Reasoning BO performs well, but also why reasoning-augmented optimization is a promising direction for scientific discovery more broadly.
>
> **Q3: Clarification on the optimization trajectory plots**
>
> A3: We thank the reviewer for raising this important point and for highlighting the difference between our plots and the conventional best-so-far visualization used in classical BO.
>
> We intentionally do not plot best-so-far curves. The purpose of our visualization is not only to show eventual optimization quality, but also to reveal how LLM-augmented BO behaves at each iteration—how hypotheses evolve, how the LLM shifts its recommended candidates, and how its exploration–exploitation balance differs from classical BO. A strictly monotonic curve would hide these dynamics.
>
> **How the plotted value is defined:**
>
> At each iteration, the surrogate-based BO module proposes five candidates, and the LLM selects three of them using its reasoning and confidence scores. The trajectory reports the **best objective value among those three evaluations within that iteration,** rather than the cumulative best-so-far across all past iterations. Because the plotted metric is per-iteration rather than a monotonic aggregate, small oscillations are expected—and in our view, informative.

---

> ### Author Response · Authors · 2025-11-20
> **Reply to Reviewer FLYi - Part 3/3**
>
> **Why do curves exhibit mild fluctuations.**
>
> Two factors contribute to the up-and-down behavior:
>
> 1. Run-to-run initialization variance (10 repeated trials).
>   Even under fixed seeds, different runs start in slightly different regions of the space. When averaged, these heterogeneous starting points naturally introduce minor fluctuations—a known effect in BO trajectory averaging.
> 2. Occasional LLM output/formatting failures that reduce the number of valid candidates.
>   These arise for two distinct reasons:
>   - High-dimensional synthetic tasks: the generated “insight” chains can become long and verbose, increasing the chance of JSON formatting errors.
>   - Chemistry tasks: some variables (e.g., long SMILES strings) are complex, and malformed tokens can prevent correct candidate extraction.
>
>   In iterations where such issues occur, fewer than three valid evaluations may be produced, temporarily lowering the plotted value. These cases are rare and do not affect convergence because Reasoning BO ultimately depends on the surrogate model.
>
> Overall, we believe this per-iteration visualization provides a more transparent view of the algorithm’s behavior than a best-so-far curve, as it makes visible the distinct decision-making patterns introduced by LLM-generated hypotheses. Importantly, the observed oscillations do not alter any qualitative conclusions or final optimization performance.
>
> **Q4: (Minor):Regarding the color palette in Figure 3**
>
> A4: We also thank the reviewer for the helpful suggestion regarding the color palette in Figure 3. We agree that some colors in the original figure lacked sufficient contrast. We will update the color scheme with higher-contrast colors to ensure all curves are clearly distinguishable.
>
> ---
>
> **Closing Remark**
>
> We sincerely thank the reviewer for the thoughtful and technically deep feedback. We hope our responses have addressed your concerns, and we warmly welcome any further suggestions. Some points you raised resonate with comments from other reviewers as well, and the relevant responses may provide additional clarification.

---

> ### Comment · Reviewer_FLYi · 2025-11-26
>
> Thank you for adding a comparison to TuRBO and BO_LS. This result significantly strengthens the paper and addresses my primary concern with the paper. Also, thank you for your detailed rebuttal and detailed explanation of why Reasoning BO performs better on some tasks than others. Regarding your decision to not plot "best-so-far" curves, that's a perfectly reasonable choice, but I suggest adding some more text in the paper making this choice clear and justifying the choice not to plot typical "best-so-far" curves. Overall, this rebuttal addresses all of my concerns, and I continue to think that this paper should be accepted.

---

> ### Author Response · Authors · 2025-11-27
>
> Thank you very much for your positive and constructive feedback. We are glad that the added comparison to TuRBO and BO_LS addressed your main concern. We agree with your suggestion about clarifying our choice not to include “best-so-far’’ curves, and we will add an explicit explanation in the revision. Regardless of the final decision on our submission, we sincerely appreciate your time and thoughtful review, which has been very helpful in improving Reasoning BO.

---

### Official Review · Reviewer_6mjx · 2025-10-27

**Soundness:** 2
**Presentation:** 3
**Contribution:** 3
**Rating:** 4
**Confidence:** 4

**Summary:**

The paper introduces Reasoning BO, which combines LLM-capabilities with Bayesian optimization. The method uses reasoning models to improve BO sampling by taking advantage of additional information and descriptions provided by the user. This is done by allowing BO to propose multiple points, after which the LLM evaluates and selects a subset of these points based on the additional information provided and its knowledge base. The authors test various iterations of their methodology, including adding a knowledge database and fine-tuned models to perform optimization. They test on various optimization problems, showing improvements beyond the baselines they present in some cases.

**Strengths:**

- The methodology shows improvement beyond traditional baselines on a variety of benchmarks. There any many interesting ideas here and I think there is a lot of potential in further exploring or expanding on these ideas.
- The ablation on initializations vs. reasoning steps provides important insights into the significance of different stages of the model. In particular, it shows that reasoning models are capable of improving all stages of the optimization process.
- The method is adaptable across many different optimization tasks and appears to have a robust structure that is capable of high-quality optimization.

**Weaknesses:**

- The paper’s clarity could be significantly improved. There are many possible implementations discussed and the paper does not include details on many of these systems. Improving the readability of the paper would significantly improve the understanding for the reader.
- The ablations presented jump around various test benchmarks and do not provide consistent baselines. I think the clarity would be improved by providing more consistency to aid interpretability here. In particular, the RL enhanced test appears to show the RL-14B model performs the best. However, it appears the methodology suggests the QWQ-plus model is selected without any RL enhancement? These inconstancies and the lack of clarity on the actual implementation shown in the results make the conclusions difficult to understand.
- The results indicate that reasoning BO appears to have much better results compared to baselines on analytic functions such as Ackley and Hartmann. However, I do not see what relevant context would be fair to provide the model with for these benchmarks. This is concerning as I would expect the model to be able to take advantage of a real-world experiment such as reaction-coupling. However, it does not appear this is the case and I am concerned the data leakage is instead motivating the results.

**Questions:**

- The paper claims that allowing BO to generate the point suggestions eliminates the possibility of data leakage from pre-training. While this may limit it, this knowledge could still be used in the selection of points. Can you further justify this claim?
- How is the supervised fine-tuning dataset generated? It appears to be generated from previous experiments. What are these experiments and should we be concerned about data leakage?
- How does the initial prompt description impact the results? Specifically, does providing more or less detail improve or degrade results How does the model perform without any additional specified information.
- Can you expand your results to a broader range of benchmarks? I would be interested if the method can compare to benchmarks such as BBOB [1] or HPO-B [2]. Including this at scale alongside more specifications about which data is included in would help negate my concerns about data leakage.
- What are the limitations of this method? Are there circumstances where the model fails/performs poorly?
------
- [1] BBOB: https://numbbo.github.io/coco/testsuites/bbob
- [2] HPO-B: https://arxiv.org/abs/2106.06257

---

> ### Author Response · Authors · 2025-11-20
> **Reply to Reviewer 6mjx - Part 1/3**
>
> **Q1: The paper’s clarity could be improved; many components are mentioned but not described in sufficient detail, and the experiments seem inconsistent across tables.**
>
> A1: Thank you for pointing this out. We agree that the current draft does not present the system with sufficient structural clarity, partly because several core components of Reasoning BO (the BO engine, reasoning agent, and memory modules) were described only in the appendix due to space constraints. In the revision, we will restructure the method section to provide a clear, self-contained pipeline description and move all essential definitions into the main text.
>
> The “inconsistency” you observe largely stems from the fact that different experiments target different sub-questions (e.g., initialization ablation vs. RL enhancements). We will make this clearer and ensure that each experiment explicitly states the component being isolated.
>
> **Q2: Baseline consistency; RL-14B vs QWQ-Plus confusion**
>
> Thank you for pointing this out — the current draft indeed does not clearly separate the roles of the two models. In all main experiments (including Figure 3), **QWQ-Plus** is the primary reasoning agent and is used consistently for candidate ranking and hypothesis refinement. The **RL-14B** model appears only in a small, isolated ablation intended to test whether additional post-training can further enhance reasoning quality. It is not part of the main pipeline and should not be interpreted as an alternative baseline.
>
> The optimization baselines used for comparison—CMA-ES, vanilla BO (qLogEI), and random sampling—remain consistent across all benchmarks. We will make this distinction explicit in the revision to avoid any confusion about which model is responsible for which results.
>
> **Q3: What information is provided to baselines? Could this introduce data leakage?**
>
> A3: Thank you for raising this thoughtful concern — we fully understand why the strong performance on certain analytic functions could suggest the possibility of data leakage. We would like to reassure the reviewer that no ground-truth values, historical samples, or prior evaluations are ever exposed to the LLM. Reasoning BO receives only the information contained in the **Experiment Compass**, which specifies variables, domains, and a short natural-language task description. This information is the minimal metadata required for any BO method to operate, and it contains **no optimization-relevant supervision**. We address it in three parts:
>
> 1. What information the LLM actually sees
>     For every task—synthetic or chemical—the LLM receives only an `Experiment Compass` of the following form, Example: Ackley-2D(Synthetic Benchmark)
>
> ```json
> {
>     "name": "Bayesian Optimization on Anonymous Multimodal Benchmark Function",
>     "constraint": "Parameters must be in [-32.768, 32.768] range.",
>     "parameter_definitions": [
>         {
>             "display_name": "x1",
>             "data_type": "float",
>             "bounds": [
>                 -32.768,
>                 32.768
>             ]
>         },
>         {
>             "display_name": "x2",
>             "data_type": "float",
>             "bounds": [
>                 -32.768,
>                 32.768
>             ]
>         }
>     ],
>     "target": {
>         "name": "Objective Function Value",
>         "direction": "minimize"
>     }
> }
> ```
>
> Crucially, these files never contain yields, rewards, trends, or any form of historical or optimization-relevant results. They merely define the search space—nothing more than what any classical BO routine must already specify. Consequently, the LLM has no access to hidden supervision or any information that could create leakage.
>
> 2. Why Reasoning BO performs well on some analytic functions (without leakage)
>
> The strong results on certain mathematical benchmarks arise not from prior knowledge, but from the inherent structure of **the functions themselves.**
>
> For example:
> - Ackley has a symmetric global minimum at $x^* = (0,\dots,0)$
> which is unusually easy for an LLM to “guess” when reasoning about “center-like” solutions.
> - Rosenbrock and Levy similarly have global minima at $x^* = (1,\dots,1)$
>
> These structured minima make it possible for the LLM to propose promising initial regions.
>
> In contrast, Hartmann-6D has a non-symmetric, irregular optimum: (0.20169, 0.15001, 0.47687, 0.27533, 0.31165, 0.65730) and Reasoning BO does not show the same level of improvement — which is exactly what we would expect if no leakage is present. The same pattern holds for Lunar Lander, where the reward is discontinuous and domain-agnostic: Reasoning BO yields only moderate gains versus other baselins.
> These observations paint a clear picture: when the optimization landscape has simple, guessable structure, the LLM may incidentally propose good initial regions; when the structure is irregular or domain-free, the model offers no undue benefit. This behavior is incompatible with leaked optimization knowledge.

---

> ### Author Response · Authors · 2025-11-20
> **Reply to Reviewer 6mjx - Part 2/3**
>
> 3. Why the strongest gains occur on chemical tasks (and why this also rules out leakage)
>
> If leakage were the driving factor, we would expect the largest gains to appear on synthetic benchmarks with simple structure. Instead, the opposite occurs—the greatest improvements are observed on the real chemistry tasks (Direct Arylation, CPA, Buchwald–Hartwig), where domain structure is rich and aligned with LLM priors. These are the settings where reasoning actually matters, and where classical BO struggles to encode mechanistic or compatibility constraints. This empirical pattern strongly supports our claim that Reasoning BO benefits from genuine reasoning rather than hidden supervision.
>
> **Summary**
> - Reasoning BO receives only high-level metadata from the Compass. No yields or historical data are ever exposed to the LLM.
> - Analytic-function performance stems from structural properties, not from leaked supervision.
> - The method excels most on real chemical optimization tasks, where genuine reasoning priors matter.
>
> We appreciate the reviewer for raising this important concern and hope the above clarifies both the information flow and the mechanism behind the observed performance patterns.
>
> **Q4: Does using BO-generated points mitigate pretraining data leakage?**
>
> We appreciate the reviewer’s thoughtful concern regarding potential pretraining-related leakage. We address it in two parts.
>
> **First**, Reasoning BO never allows the LLM to generate numerical candidates after initialization; it only ranks or filters the points proposed by the BO engine. This design substantially limits the influence of any pretrained knowledge compared to approaches where LLMs directly propose new experimental conditions. While the ranking step inevitably uses the model’s prior knowledge to assess plausibility, this process does not provide a channel for leaking ground-truth function values or historical optimization signals—only high-level preferences or heuristic plausibility assessments shape the selection.
>
> **Second**, we do not believe pretrained models can meaningfully “leak’’ concrete experimental optima in the context of our chemistry benchmarks. Real reaction conditions, reagent identities, and quantitative loadings are not memorized patterns that foundation models can recall. For example, even the strongest models cannot generate the optimal substrate–ligand–base–solvent combination for a Suzuki reaction when asked directly in free-form, much less produce the precise numeric loadings matching our datasets. These optima are highly specific to the particular reaction setup and dataset, and are not present in public text corpora used for LLM pretraining.
>
> That said, we agree with the reviewer that pretrained chemical knowledge (e.g., general reactivity preferences, compatibility heuristics) can influence which BO-proposed points the LLM ranks higher. This type of domain prior is difficult to avoid in any LLM-guided optimization framework, but it does not constitute data leakage from the benchmark’s ground truth. It simply reflects the model’s scientific intuition, not privileged access to the target function values.
> If the reviewer has a particular leakage scenario in mind, we would be very glad to discuss it further. We want to ensure the setup is fully transparent and that all concerns about unintended supervision are thoroughly addressed.
>
> **Q5: How was the SFT dataset generated? Does it create a data-leakage risk?**
>
> A5: The supervised fine-tuning dataset was generated only from a small number of very early pilot runs, and its sole purpose was to enforce correct JSON formatting rather than provide any task knowledge. It contains no objective values or optimization trajectories. Only the RL-14B model was fine-tuned, and it was tested exclusively on two synthetic mathematical benchmarks; we verified that none of the SFT data includes these functions or any structurally related examples. Therefore, the SFT process does not introduce any data-leakage risk.
>
> **Q6: How does the initial prompt affect results? Would more/less detail change performance?**
>
> A6: As clarified in **Q3**, all experiments use only the minimal information in the Experiment Compass—variable ranges plus a brief task description—with no additional hints, priors, or engineered prompts. The reported performance therefore already reflects how the model behaves under uniformly sparse prompts, and we do not vary the level of detail in this version of the method. The interface is intentionally designed so that, in real wet-lab settings, scientists could optionally add domain priors in the future; although we have not yet tested this regime, we expect richer expert-provided detail to further enhance Reasoning BO, consistent with the motivation behind combining LLM reasoning with Bayesian optimization.

---

> ### Author Response · Authors · 2025-11-20
> **Reply to Reviewer 6mjx - Part 3/3**
>
> **Q7: Can the method be tested on broader benchmarks(e.g., BBOB, HPO-B)?**
>
> A7: Reasoning BO is fully compatible with broader suites such as BBOB and HPO-B; although our focus is on real-world chemical and biological tasks where reasoning is most impactful, we are integrating these benchmarks during the rebuttal period and will share the additional results separately, which we hope can further alleviate the reviewer’s concerns about data leakage.
>
> **Q8: Limitations of this method? Are there failure cases?**
>
> A8: We appreciate the reviewer’s thoughtful question. As noted in Appendix H, Reasoning BO does have several practical limitations. First, the method inherits the context-window constraints of modern LLMs: although our single-turn design minimizes accumulation, very long optimization campaigns may still approach the context limit and degrade reasoning quality. Second, the framework depends on strong instruction-following ability. Smaller models (3B–7B) frequently fail to maintain the strict Insight-Object format or generate valid hypotheses, and even light SFT/RL can reduce their general instruction compliance. This is why our experiments ultimately rely on 14B/32B-scale models.
>
> Failure cases primarily arise when (i) the insight chain becomes extremely long in high-dimensional analytic functions, or (ii) variables contain complex structures (e.g., long SMILES strings), causing invalid candidates or missed hypotheses. These issues do not affect final convergence, and in real wet-lab scenarios they are further mitigated because experiments proceed step-by-step rather than through dozens of uninterrupted iterations. Nonetheless, they highlight that the method’s robustness remains tied to LLM capability.
>
> Finally, we note that modern frontier models have substantially improved context and formatting robustness; in practice, we did not encounter context failures in any chemistry experiments. Our best recommendation for practitioners is simply to use the strongest model available, as this significantly reduces formatting errors and improves stability.
>
> ---
>
> **Closing Remark**
>
> We sincerely thank the reviewer for the constructive and encouraging feedback. Due to space limits, several clarifications and additional quantitative results are provided in the corresponding appendices, and we kindly invite you to refer to them for fuller context. We hope our responses have addressed your concerns, and we warmly welcome any further suggestions or discussion. Several of your points overlap with feedback from other reviewers, and the corresponding responses may offer additional clarification.

---

> > ### Comment · Reviewer_6mjx · 2025-11-24
> >
> > Thank you for your response.
> >
> > I appreciate your explanation of why reasoning BO performs so well on your analytical tests. This reduces my concern of data leakage for these examples but unfortunately brings up new questions about the true applicability of these benchmarks to real-world scenarios that may not have symmetrical structures. Overall, I find the lack of evaluation functions tests to be concerning and an area that needs significant improvement. I understand that the field has limited standard real-world evaluations and this is a difficult problem to solve, but providing more robust results that show the methodology at least maintains performance over standard synthetic benchmarks would help convince me that the presented results are robust to different scenarios. Additionally, while you state that “the strongest gains occur on chemical tasks”, this is not how it appears in the results, where the strongest gains are on the structured analytical functions. Thank you for integrating these benchmarks into your results and I will update my response when those results are included.
> >
> > Regarding the fine-tuned model, I think your presentation of this aspect remains unclear. The RL approach appears to outperform your base method on the limited tests shown. Can you clarify if this is the case and why it was only tested on two analytical functions. These results do not seem robust enough to make conclusions on the different model’s capabilities or gain insight on your fine-tuning approach. Overall, the experiments and design details here remain too limited for an aspect of the model that is heavily highlighted in your paper. More analysis and detail would be appreciated if you wish to highlight these results.

---

> ### Author Response · Authors · 2025-12-01
>
> **Q1: Real-world scenarios that may not have symmetrical structures**
>
> A1: Thank you for the comment. Real-world chemistry optimization does not rely on symmetry; instead it contains rich domain structures such as functional-group compatibility, ligand preferences, solvent effects, and mechanistic constraints. These are patterns that LLM-based reasoning can naturally leverage but are difficult to encode in classical BO. Our results already confirm this: the observed gains are consistently validated across the **four real-world chemistry benchmarks** included in our study.
>
> **Q2: Reviewer Concern About the Scope of Real-World Benchmarks**
>
> A2:  We appreciate the reviewer’s concern. Obtaining complete, well-curated experimental datasets like the Suzuki is inherently difficult, and the real-world chemistry tasks used in our paper already represent the upper limit of what we were able to locate and process from existing open-source repositories and published studies. Within the space of LLM-based BO research, to the best of our knowledge, this is currently the most chemistry-focused collection of real-world benchmarks available.
>
> Thank you for the suggestion. As more suitable datasets become accessible in the future, we will incorporate them to further enrich the evaluation.
>
> **Q3: On the RL-tuned model and why results appear stronger**
>
> A3.1 : Why the RL-tuned model performs better
>
> As shown in the supplementary experiments (Figure 9), we consistently observe that larger or more reasoning-capable models perform better in Reasoning BO. The purpose of applying RL was precisely to improve multi-step reasoning quality, so it is expected—and consistent with this trend—that the RL-14B model appears stronger on the two analytic functions where it was evaluated.
>
> A3.2: Why the RL-14B model was tested only on two analytic benchmarks
>
> Although RL improves reasoning, prior work has shown that SFT/RLHF can weaken instruction-following and formatting stability (e.g., structured output, JSON compliance) compared to the base model [1, 2].
>  In our framework, maintaining strict JSON schema consistency is essential for end-to-end optimization (Appendix I). In practice, the RL-14B model frequently violates the required schema when used on complex real-world chemistry tasks or higher-dimensional synthetic tasks, which then requires human intervention to recover the loop.
>
> For this reason, we evaluated the RL model only on two lower-dimensional analytic benchmarks, where formatting failures are manageable. The intention was to demonstrate the potential of RL-enhanced reasoning, not to position RL-14B as a fully robust variant of the system. In realistic wet-lab scenarios, experiments proceed one iteration at a time, and stronger base models or more robust RL methods could be used to address this limitation. Our current results therefore remain intentionally scoped to illustrating feasibility rather than making broad claims.
>
> ---
>
> [1]: Ouyang, L. et al. (2022). Training language models to follow instructions with human feedback. NeurIPS.
>
> [2]: Zhou, H. et al. (2023). LIMA: Less Is More for Alignment. arXiv:2305.11206.

---

> ### Author Response · Authors · 2025-12-01
> **Additional Evaluation on BBOB Benchmark**
>
> To address the reviewer's request for broader synthetic validation beyond analytical functions, we additionally evaluated Reasoning BO on functions from the **BBOB suite**[1], covering both **final optimization quality** and **early-stage performance**. To avoid any bias in selecting benchmark tasks, the specific functions used here (F1, F6, F7, F9, F13, F18) were randomly sampled from the official BBOB Benchmark suite. Each method was run for 30 iterations under identical seeds. The benchmark includes both 5D and 10D settings and compares against BO, CMA-ES, and Random sampling.
>
> ## Task Notation
>
> Each task is denoted as `F{Function_ID}_{Dimension}D`, where:
>
> - `F{Function_ID}`: The BBOB function identifier (e.g., F1, F6, F7, F9, F13, F18)
> - `{Dimension}D`: Problem dimensionality (5D or 10D)
> - **Example**: `F1_5D` represents BBOB function 1 in five dimensions.
>
> ## Improvement Metric
>
> Improvement is computed as:
>
> $$ \text{Improvement} = \frac{\text{Baseline Mean} - \text{Reasoning BO Mean}}{\text{Baseline Mean}} \times 100\% $$
>
> We report results under two complementary metrics:
>
> ---
>
> ### **(1) Final Best Performance**
>
> | Task | Reasoning_BO Mean | BO Mean | BO Improvement% | CMA-ES Mean | CMA-ES Improvement% | Random Mean | Random Improvement% |
> |------|-------------------|---------|-----------------|-------------|---------------------|-------------|---------------------|
> | F1_5D | 88.39 | 101.68 | **13.07%** | 136.70 | **35.34%** | 133.22 | **33.65%** |
> | F6_5D | 1627.50 | 46714.36 | **96.52%** | 70042.77 | **97.68%** | 216987.97 | **99.25%** |
> | F7_5D | 213.68 | 383.86 | **44.33%** | 240.51 | **11.16%** | 600.90 | **64.44%** |
> | F9_5D | 1511.13 | 23584.99 | **93.59%** | 116828.23 | **98.71%** | 58629.24 | **97.42%** |
> | F18_5D | 23.54 | 63.31 | **62.82%** | 192.74 | **87.79%** | 260.46 | **90.96%** |
> | F7_10D | 784.58 | 897.47 | **12.58%** | 1985.09 | **60.48%** | 1876.34 | **58.19%** |
> | F13_10D | 1604.63 | 1610.68 | **0.38%** | 3231.23 | **50.34%** | 2252.87 | **28.77%** |
> | F18_10D | 61.95 | 176.96 | **64.99%** | 259.00 | **76.08%** | 582.10 | **89.36%** |
>
> **Summary**: Across these tasks, Reasoning BO improves over classical BO by an average of **+48.5%**, and shows larger margins relative to CMA-ES (**+64.7%**) and Random search (**+70.3%**).
>
> ---
>
> ### **(2) Early-Stage Optimization Evaluation (IMP@10)**
>
> To evaluate whether the improvements arise only from later convergence or also accelerate early reasoning-guided exploration, we compute IMP@10 as used in the paper:
>
> $$\text{IMP@10} = \frac{1}{10} \sum_{i=1}^{10} f(x_i)$$
>
> Lower values indicate better early optimization behavior.
>
> Improvement is measured as:
>
> $$\text{Improvement} = \frac{\text{Baseline IMP@10} - \text{Reasoning BO IMP@10}}{\text{Baseline IMP@10}} \times 100\%$$
>
> | Task | Reasoning_BO IMP@10 | BO IMP@10 | BO Improvement% | CMA-ES IMP@10 | CMA-ES Improvement% | Random IMP@10 | Random Improvement% |
> |------|---------------------|-----------|------------------|--------------|---------------------|---------------|---------------------|
> | F1_5D | 89.01 | 129.40 | **31.21%** | 135.88 | **34.49%** | 123.16 | **27.73%** |
> | F6_5D | 4619.33 | 122727.90 | **96.24%** | 127954.35 | **96.39%** | 188758.75 | **97.55%** |
> | F7_5D | 237.71 | 580.77 | **59.07%** | 361.26 | **34.20%** | 500.79 | **52.53%** |
> | F9_5D | 4370.83 | 46144.62 | **90.53%** | 67981.42 | **93.57%** | 71540.24 | **93.89%** |
> | F18_5D | 46.65 | 85.19 | **45.23%** | 183.58 | **74.59%** | 108.42 | **56.97%** |
> | F7_10D | 938.88 | 1431.82 | **34.43%** | 1953.54 | **51.94%** | 1889.28 | **50.30%** |
> | F13_10D | 1701.98 | 2373.81 | **28.30%** | 3005.51 | **43.37%** | 2534.14 | **32.84%** |
> | F18_10D | 68.23 | 253.96 | **73.13%** | 333.04 | **79.51%** | 499.45 | **86.34%** |
>
> **Summary**: Reasoning BO improves early optimization performance relative to:
>
> | Baseline | Avg. IMP@10 Improvement |
> |----------|-------------------------|
> | BO | **+57.27%** |
> | CMA-ES | **+63.51%** |
> | Random | **+62.27%** |
>
> ---
>
> [1] BBOB: https://numbbo.github.io/coco/testsuites/bbob
>
> ---
>
> ### **Conclusion**
>
> These results provide additional evidence that the method generalizes beyond domain-specific chemistry settings and remains robust on broader synthetic optimization challenges. We appreciate your participation in this productive discussion. We hope all your doubts have been addressed. Thank you!

---

### Official Review · Reviewer_ZUuJ · 2025-10-27

**Soundness:** 1
**Presentation:** 2
**Contribution:** 2
**Rating:** 2
**Confidence:** 4

**Summary:**

- This paper proposes Reasoning BO, an LLM-based BO framework that integrates reasoning LLMs, a multi-agent system, and knowledge storage + RAG
- The LLM participates in evaluating candidate points/assigning confidence scores. LLM-based agents are also employed to store reasoning information from previous trials
- The authors also include a RL finetuning stage (on GSM8K), showing that it improves BO performance

**Strengths:**

- This falls in line with recent efforts that try to integrate LLMs with optimization frameworks
- AFAIK, this is the first work to explicitly investigate whether long-form CoT reasoning can improve BO performance

**Weaknesses:**

- [P1] If my understanding is correct, the LLM is utilized more as an 'acquisition function', scoring proposed hypotheses (from a Gaussian Process), and filtering by confidence. This does not seem to contain any mechanism to encourage exploration, as one would expect underexplored regions to be reflected in lower confidence.
- [P2] It is not clear why LLMs are not directly used to propose the hypotheses (as in LLAMBO, BORA that investigate this specifically for BO, but also EUREKA, ALPHA-EVOLVE that are more general). By using a conventional GP to propose candidates, the method still faces the same scalability (L87-88) issues and high-dimensionality challenges.
- [P3] Perhaps most importantly, it is unclear whether the knowledge graph is useful. It appears the database of <entity, relation, entity> is distilled from LLM reasoning traces. This seems redundant since it is from the LLM itself. The practical benefits of the knowledge system (vector DB + graph DB) are also not shown through targeted ablation experiments.
- [P4] The RL post-training stage uses GSM8K (math dataset); it is unclear how this transfers to optimization reasoning. If the authors believe that general reasoning training is useful, it would be better to employ a 'generic' reasoning LLM directly. In other words, it is unclear what the purpose or value of finetuning on GSM8K brings.
- [P5] I am worried about contamination risk; the real-world benchmarks are well-known, and although the LLMs are not involved in proposing candidates, they still act as a selection function, introducing indirect contamination risk
- [P6] The paper is missing comparisons to stronger or recent LLM-based or advanced BO baselines (e.g., HEBO, kernel BO, LLAMBO, OPRO), and more traditional baselines (e.g., tree-based)
- [P7] Fig 3 suggests different initialization across methods, which confounds results. I would recommend comparing baselines with the same initialization to understand the advantages purely through optimization. This is especially so as the initialization seems to have a disproportionately large impact, which might be due to contamination.
- [P8] In general, the experiments (with 10 tasks) are fairly limited to establish robustness for an optimizer paper. Please also report the std dev/CI for the results in Table 1.

Overall, the framework feels over-engineered and is a mixture of components without clear evidence of why and which components matter.

**Questions:**

- How is exploration impacted if LLM-assessed confidence is used to acquire new points.
- What is the 'Experiment Compass'
- Have the authors considered RL post-training with more relevant tasks (e.g., related to BO).
- Can the authors clarify the LogEI was used for all baselines?
- What is the performance gain from knowledge graph and MA subsystems?

---

> ### Author Response · Authors · 2025-11-20
> **Reply to Reviewer ZUuJ - Part 1/4**
>
> **Q1: What is the exact role of the LLM in Reasoning BO?**
>
> A1: In our framework, the LLM does not propose new candidate points in the search space. Candidate locations $x_1,\dots,x_k$ are generated by a standard BO backbone (Gaussian process + qLogEI acquisition). For each candidate, we then query the LLM with a textual summary of past trials and experimental context; the LLM returns a confidence score based on its own hypotheses. These LLM scores are used only to re-rank the BO-proposed candidates and select a small subset to evaluate. In other words, the LLM serves as an additional, knowledge-driven acquisition critic on top of classical BO, rather than replacing the surrogate model or directly generating candidate points.
>
> **Q2: Why not let the LLM propose candidate points directly, as in LLAMBO/BORA-style methods?**
>
> A2: Our framework does allow the LLM to generate and evolve its own hypotheses across iterations—but we deliberately avoid letting the LLM directly propose numerical candidate points for BO evaluation. This choice is motivated by two practical considerations:
>
> First, in real optimization workflows (especially scientific ones), hallucinated or out-of-domain proposals from an LLM can be difficult to detect and may introduce uncontrolled failure modes. By grounding all proposed points in the GP + qLogEI backbone, we ensure a reliable lower bound on performance: every evaluated point remains a valid BO candidate regardless of LLM behavior, which is important for reproducibility and trustworthy experimentation.
>
> Second, many candidate-generation approaches that rely solely on LLM sampling (e.g., LLAMBO[1], BORA[2] variants) are more susceptible to contamination effects and implicit prior knowledge leaking into the search space. Using the GP to generate all numerical proposals avoids this concern while still allowing the LLM’s hypotheses to influence ranking and selection.
>
> We acknowledge that GP-based proposal mechanisms face known scalability limitations in very high dimensions, and we have already noted this in our discussion of limitations **(See Appendix H)**.
>
> ---
>
> [1]: Yang, J., Li, Z., Shi, S., Wang, Z., & Chen, Y. (2024). LLAMBO: Large Language Models for Multi-Step Bayesian Optimization. arXiv:2402.03921.
>
> [2]: Cissé, A., Evangelopoulos, X., Gusev, V. V., & Cooper, A. I. (2025). Language-Based Bayesian Optimization Research Assistant (BORA). arXiv:2501.16224.
>
> **Q3: Is the knowledge graph actually useful？Evidence from Ablations**
>
> A3: We appreciate the reviewer’s thoughtful question. In the current dry-experiment setting (small synthetic landscapes and limited evaluation rounds), the benefits of the knowledge graph are indeed modest. These tasks—especially Hartmann—contain little meaningful domain structure, so classical BO already performs near its ceiling, and injecting additional priors cannot help much.
>
> That said, the knowledge system is indeed used and tested. **Figure 7 of the paper provides a targeted ablation (+Notes vs. –Notes)** across both chemical and mathematical tasks. These results show:
>
> - **Chemistry tasks (Suzuki, Direct Arylation):**
>   Adding structured notes—which include distilled graph relations and retrieved knowledge—yields consistent improvements.
>   - Suzuki shows an approximately 2% boost in early performance.
>   - Direct Arylation, a much more challenging and failure-prone reaction system, benefits substantially in mid-to-late stages. (As noted in Reviewer 5, Q2, Direct Arylation is far more rugged than Suzuki, so knowledge becomes especially valuable.)
> - **Mathematical task (Hartmann-6D):**
>   The same structured notes hurt performance. Since Hartmann has no real-world semantic structure, the injected priors constrain exploration and slow progress.
>
>   This outcome is expected: it demonstrates that the knowledge module is not merely redundant—it has domain-dependent effects, improving results only when meaningful priors exist.
>
> Importantly, the knowledge graph is designed as a **pluggable architectural component** for real scientific workflows—where mechanistic relationships (e.g., reagent–solvent compatibility, feasible parameter co-occurrences) are critical and classical BO has no mechanism to represent them. Synthetic benchmarks cannot capture these real-world structures, which is why the gains appear more clearly in chemistry tasks.

---

> ### Author Response · Authors · 2025-11-20
> **Reply to Reviewer ZUuJ - Part 2/4**
>
> **Q4: Why use GSM8K for the RL post-training stage? How does it relate to optimization reasoning ?**
>
> A4: We appreciate the question. GSM8K was **not** chosen because it matches BO, nor is it part of our core pipeline. Its purpose was simply to probe a hypothesis supported by recent reasoning-focused RL works [1, 2]: **improving general multi-step reasoning often transfers to downstream decision-making tasks,** including heuristics used during optimization.
>
> We therefore used a lightweight GSM8K-based RL stage as an exploratory test of this transferability. It appears only as an auxiliary model in Figure 8, and the final model used throughout the paper is **QwQ-Plus**, without RL or GSM8K involvement.
>
> We agree with the reviewer that, in practice, domain-specific RL or task-aligned post-training would be more appropriate. Indeed, for real applications—especially robotics or edge-deployment scenarios—specialized RL tuning may be beneficial. Our experiment here was only intended to demonstrate that such adaptation is feasible, not to claim GSM8K is optimal for BO.
>
> ---
>
> [1] Guo, D. (2025). DeepSeek-R1 incentivizes reasoning in LLMs through reinforcement learning. Nature, 645, 633–638.
>
> [2] Muennighoff, N. (2025). s1: Simple test-time scaling. arXiv preprint arXiv:2501.19393.
>
> **Q5: Data-contamination risk when LLMs act as the acquisition function?**
>
> A5: We share the reviewer’s concern. Even though the LLM does not generate candidate points, its role in ranking BO proposals can still introduce indirect contamination risks if any real-world benchmark overlaps with its pretraining corpus. To mitigate this, we intentionally adopted a conservative design: the GP/qLogEI backbone always determines the numerical candidate set, and the LLM only re-orders a small, fixed pool of BO-valid points. This substantially reduces the possibility of contamination, but it cannot completely eliminate it.
>
> Importantly, the optima for real chemical reactions are highly specific combinations of substrates, ligands, bases, solvents, and quantitative loadings. Such configurations are extremely unlikely to appear in pretraining corpora, and even frontier models cannot generate the correct substrate–ligand–base–solvent pairing for Suzuki or Direct Arylation reactions when queried directly. Because these optima are effectively absent from the training distribution, an LLM cannot simply recall them when scoring BO proposals.
>
> We initially explored adding verification modules or secondary “sanity-check agents” to detect hallucinated knowledge or potential leakage, but these mechanisms significantly increased latency, resource usage, and complexity for readers, so we did not include them in the present version. We would welcome any references or techniques the reviewer recommends for further reducing contamination risk in LLM-guided optimization pipelines.
>
> **Q6: Lack of comparison to stronger LLM-based BO methods or more advanced BO / tree-based baselines**
>
> A6: Thank you for the suggestion. Our framework is intentionally different from LLAMBO/BORA-style methods: those approaches allow the LLM to directly generate numerical candidates, whereas our design restricts the LLM to an acquisition-function role to avoid hallucinated or out-of-domain proposals. Because of this design choice, we focus on baselines that modify only the acquisition layer while keeping the BO bakebone fixed.
>
> We have already added several stronger classical BO baselines (including TuRBO) and are running additional ablations during the rebuttal period (see Reviewer 5, A1). Additional comparisons will be incorporated into the camera-ready version.
>
> **Q7: Different initialization strategies may bias the results; should all methods use the same initialization?**
>
> A7: Initialization indeed has a strong impact on BO. Classical BO baselines start with Sobol/random points, whereas our method uses LLM-generated hypotheses. This difference is intentional rather than incidental: the initialization stage is precisely where we expect reasoning to influence the search.
>
> That said, we acknowledge the reviewer’s concern. Our ablations at Figure 6 confirm that both the initialization phase and the later BO refinement behave as expected, and the advantage persists even when we restrict the LLM’s influence. Since the GP + qLogEI backbone is identical across VBO and RBO, the observed differences arise from how each method conducts early search rather than from external contamination or implementation inconsistencies.

---

> ### Author Response · Authors · 2025-11-20
> **Reply to Reviewer ZUuJ - Part 3/4**
>
> **Q8: Are 10 tasks sufficient for validating an optimizer paper? Please report the standard deviation or confidence intervals.**
>
> A8: Our goal is not to propose a new standalone optimizer, but to explore how LLM reasoning can be integrated into the BO pipeline. That said, we agree that reporting variability is important, and we have included standard deviations for all results in Table 1 in the revised version.
>
> **Q9: If the LLM uses its own confidence to select new points, how does this affect exploration?**
>
> A9: In our design, LLM chose candidates based on its evolving hypotheses. Across iterations, the LLM may maintain multiple competing hypotheses and naturally alternates between confirming them (exploitation) and probing inconsistencies (exploration).
>
> To illustrate this behavior more concretely, we include below a representative example from Suzuki Benchmark. The LLM starts with multiple chemically plausible hypotheses (iodide systems with electron-rich ligands; bromide systems with optimized ligands; bulky-ligand synergies). As iterations proceed, the model updates these hypotheses based on observed outcomes and adjusts the corresponding confidence levels. This yields a transparent form of “reasoning-based” exploration, where the model actively re-evaluates its beliefs rather than greedily exploiting a single mode.
>
> ```json
> {
>   "comment": "The highest yields stem from iodide electrophiles paired with CsF base and THF solvent. The nucleophile B(O)O and bulky triphosphine ligands (e.g., CC(C)(C)P(C(C)(C)C)C1=CC=C(N(C)C)C=C1) significantly enhance yields. Bromide systems underperform unless optimized with suitable ligands and solvents.",
>   "hypotheses": [
>     {
>       "strategy": "Iodide with Electron-Rich Ligands",
>       "rationale": "Electron-rich ligands stabilize palladium intermediates, enhancing transmetallation efficiency with iodide. Testing ligands with strong electron-donating groups to boost reactivity.",
>       "confidence": "high",
>       "parameter_sets": [
>         {
>           "Electrophile_SMILES": "IC1=CC=C(N=CC=C2)C2=C1",
>           "Nucleophile_SMILES": "CC1=CC=C(N(C2CCCCO2)N=C3)C3=C1B(O)O",
>           "Ligand_SMILES": "P(C1=CC=CC=C1)(C2=CC=CC=C2)C3=CC=CC=C3",
>           "Base_SMILES": "[Cs+].[F-]",
>           "Solvent_SMILES": "C1COCC1"
>         },
>         {
>           "Electrophile_SMILES": "IC1=CC=C(N=CC=C2)C2=C1",
>           "Nucleophile_SMILES": "CC1=CC=C(N(C2CCCCO2)N=C3)C3=C1B(O)O",
>           "Ligand_SMILES": "CC1=CC=CC=C1P(C2=CC=CC=C2C)C3=CC=CC=C3C",
>           "Base_SMILES": "[Cs+].[F-]",
>           "Solvent_SMILES": "C1COCC1"
>         }
>       ]
>     },
>     {
>       "strategy": "Bromide with THF and Optimized Ligand",
>       "rationale": "Bromide reactivity improves with THF solvent and a ligand balancing electronic effects and steric hindrance to enhance transmetallation.",
>       "confidence": "medium",
>       "parameter_sets": [
>         {
>           "Electrophile_SMILES": "BrC1=CC=C(N=CC=C2)C2=C1",
>           "Nucleophile_SMILES": "CC1=CC=C(N(C2CCCCO2)N=C3)C3=C1B(O)O",
>           "Ligand_SMILES": "[c-]1(P(C2=CC=CC=C2)C3=CC=CC=C3)cccc1.[c-]4(P(C5=CC=CC=C5)C6=CC=CC=C6)cccc4.[Fe+2]",
>           "Base_SMILES": "[Cs+].[F-]",
>           "Solvent_SMILES": "C1COCC1"
>         }
>       ]
>     }
> }
> ```
>
> This multi-hypothesis structure induces exploration not through randomness, but through **explicit hypothesis** revision grounded in the model’s own reasoning trace.
>
> To make this dynamic clearer, we include below a confidence-evolution snapshot drawn directly from one real Suzuki optimization run.
>
> Table: Confidence Evolution Table Across Trials
> | **Hypothesis Strategy**                     | Trial 0-5 Confidence | Trial 6-15 Confidence | Trial 16-29 Confidence |
> |--------------------------------------------|----------------------|-----------------------|------------------------|
> | Iodide with electron-rich ligands           | High                 | Medium                | Medium                 |
> | Bromide with CCCC ligand                    | N/A                  | Medium                | High                   |
> | Bulky nucleophiles with optimal ligands     | Medium               | High                  | High                   |
> | Sulfone electrophile optimization           | Low                  | Low                   | Low                    |
> | Bromide with P(C1=CC...) ligand             | Medium               | High                  | Medium                 |
> | Chloride/bulky ligand systems               | Medium               | Low                   | Discarded              |
>
> The optimization converged on bromide as the top electrophile, with CCCC ligand and bulky nucleophiles becoming the cornerstone of high-yield conditions. Confidence solidified around these parameters while peripheral strategies were phased out.

---

> ### Author Response · Authors · 2025-11-20
> **Reply to Reviewer ZUuJ - Part 4/4**
>
> **Q10: What is the Experiment Compass?**
>
> A10: The `Experiment Compass` is a lightweight, task-level abstraction that provides the LLM with minimal contextual information needed to reason about the optimization problem.
>
> Example: Ackley-2D(Synthetic Benchmark)
>
> ```json
> {
>     "name": "Bayesian Optimization on Anonymous Multimodal Benchmark Function",
>     "constraint": "Parameters must be in [-32.768, 32.768] range.",
>     "parameter_definitions": [
>         {
>             "display_name": "x1",
>             "data_type": "float",
>             "bounds": [
>                 -32.768,
>                 32.768
>             ]
>         },
>         {
>             "display_name": "x2",
>             "data_type": "float",
>             "bounds": [
>                 -32.768,
>                 32.768
>             ]
>         }
>     ],
>     "target": {
>         "name": "Objective Function Value",
>         "direction": "minimize"
>     }
> }
> ```
>
> **Q11: Have the authors considered RL post-training with more relevant tasks (e.g., related to BO).**
>
> A11: We agree that RL post-training on tasks more closely aligned with BO is a promising direction. In fact, one broader motivation of Reasoning BO is to explore whether general reasoning improvement or domain-specific RL can enhance downstream optimization. Our RL experiment was therefore intended only as a preliminary feasibility probe, not as a core component of the method.
>
> For this paper, we deliberately avoided RL on domain-specific scientific tasks because our benchmarks fall entirely in the “dry-experiment’’ category, where task-specific RL can easily raise contamination or leakage concerns.
>
> **Q12: Is LogEI used for all baselines?**
>
> A12: As described in Section 4.1, both Vanilla BO and Reasoning BO use the **qLogEI** acquisition function, Analytic-EI use the Analytic LogEI acquisition function. However, CMA-ES is a fundamentally different algorithm and therefore does not use LogEI. All acquisition‐related configurations and mathematical details are provided in Appendix E.
>
> **Q13: What performance gains do the knowledge graph and database provide?**
>
> A13: Their role and expected impact are the same as discussed in **Q3**: in the small-scale dry benchmarks used in the paper, the improvement is naturally limited, while their primary purpose is to serve as interpretable, pluggable priors for real-world, knowledge-rich optimization. Please refer to our response in **A3** for details.
>
> ---
>
> **Closing Remark:**
>
> We sincerely thank the reviewer for the constructive and encouraging feedback. Due to space limits, several clarifications and additional results are provided in the corresponding appendices, and we kindly invite you to refer to them for fuller context. We hope our responses have addressed your concerns, and we warmly welcome any further suggestions or discussion. Several of your points overlap with feedback from other reviewers, and the corresponding responses may offer additional clarification.

---

> ### Author Response · Authors · 2025-11-28
> **Hoping to continue the discussion**
>
> Dear Reviewer,
>
> Thank you for your insightful comments and suggestions. We have submitted responses addressing the points you raised, including clarifications and revisions to improve the presentation and completeness of the work. Should you have any further questions or require additional clarification, please do not hesitate to ask.
>
> We also note that our responses to the other reviewers have been positively received and have further solidified their support for our work. You may find their specific points and our detailed replies informative.
>
> We look forward to your updated review and remain open to further discussion. Thank you very much.

---

### Official Review · Reviewer_uEhn · 2025-10-31

**Soundness:** 2
**Presentation:** 2
**Contribution:** 3
**Rating:** 2
**Confidence:** 3

**Summary:**

This paper proposes Reasoning Bayesian Optimization (Reasoning BO), a framework that integrates LLMs with Bayesian optimization to enhance sample selection. The idea is to leverage the reasoning capabilities and contextual knowledge of LLMs to guide BO. Specifically, the BO process generates a set of candidate points, and the LLM selects a subset of promising candidates for evaluation. The LLM is aided by additional textual descriptions and domain knowledge. The authors explore several variants of this approach, including versions augmented with external knowledge bases and fine-tuned reasoning models. Experiments across a range of optimization problems demonstrate that the proposed method can outperform standard BO baselines in certain settings.

**Strengths:**

The approach presents a novel integration of LLM reasoning with classical BO, showing measurable improvements on multiple benchmarks. The underlying concept is interesting and opens up many promising directions for future research and further improvements.
The proposed framework is flexible and general enough to be applied across diverse optimization domains, suggesting potential for scalability and cross-domain applicability. This is beneficial as this is a weakness of similar methods, which may not adapt to different circumstances in the same manner.
The authors provide many ablations on the model structure and design. There are many components that go into reasoning BO and comparing the effects of these changes is an important aspect of understanding the design.

**Weaknesses:**

The clarity and flow of the paper is poor. Several implementation choices are mentioned but not detailed, making it difficult to reproduce or fully understand the system. The paper mentions SFT on an unspecified dataset and GRPO on GSMK. It is not clear if this went into the final model. The lack of clarity here is a big weakness. A clearer, more structured presentation would strengthen the work substantially.

The experimental section lacks consistency. The benchmarks and baselines vary across tables and do not provide a complete picture of the results, complicating interpretation.

The paper discussed data leakage as a concern but does not sufficiently address this aspect. It is unclear how much the results are motivated by data leakage. I think further analysis of the additional data inputted and the data used for fine-tuning would improve the paper.

**Questions:**

The paper does not mention any limitations of reasoning BO. Are there limitations on data types, number of evaluations, evaluations that fail? Does reasoning BO perform worse than classical methods in some cases?

Can you provide an ablation on the additional information provided to the model? How does it perform without additional information or in scenarios where there is not additional information.

The results on Ackley, Rosenbrock, and Hartmann are strong. However, I do not see information about what additional data is provided to the model for these cases or what relevant information would be justifiable to provide to the model. Can you justify why the model would perform so well without additional information?

Can you provide more tests on more function spaces? The limited evaluation and lack of baselines makes the results less convincing. Providing a larger-scale test would help negate concerns that there is too much data leakage or unfair additional information passed to the model.

---

> ### Author Response · Authors · 2025-11-20
> **Reply to Reviewer uEhn - Part 1/3**
>
> **Q1: Clarity of description, SFT/GRPO, and insufficient implementation details**
>
> A1: Thank you for raising this important point. We acknowledge that the current draft is not sufficiently self-contained, especially regarding Algorithm 1 and the interactions among its components. Because the system involves multiple modules (Experiment Compass, knowledge-storage structures), many definitions and examples were placed in the appendix, which understandably interrupts the flow of the method section. In the revision, we will restructure the presentation so that Algorithm 1, all symbols, and all core interfaces are fully defined within the main text.
>
> Regarding why SFT and GRPO are used: SFT is employed solely to teach the model to output structured JSON following our predefined schema (**as noted in Appendix F**), enabling reliable extraction of experiment traces. In early exploratory runs, we observed that RL-trained models often exhibited weaker instruction-following ability and would fail to produce valid JSON in long optimization trajectories, causing the loop to fail. Adding a small amount of format-only SFT data significantly stabilizes schema adherence and prevents these interruptions. GRPO is applied to improve the model’s multi-step reasoning consistency—its purpose is not domain specialization, but rather to stabilize the reasoning pipeline across tasks. We will make this motivation explicit.
>
> Finally, we will clarify which models are actually used in the final system. Except for the baseline comparison in Figure 8, all experimental results use QWQ-Plus as the reasoning model. The RL-14B variant appears only in a small, isolated ablation. Similarly, the two knowledge-storage modules (vector DB and knowledge graph) are optional and are not required for the core pipeline. We will ensure that the revised version clearly separates the main algorithm from auxiliary explorations so that readers can fully understand which components contribute to which results.
>
> **Q2：Lack of consistency across benchmarks and baselines**
>
> A2: Thank you for the question. We do use a unified baseline set—Vanilla BO, CMA-ES, Analytic EI, and Random Search—but for each ablation we include only the baselines relevant to the component under investigation. Adding every baseline to every figure would create redundancy and obscure the purpose of each comparison. If there is a specific pairing the reviewer would find helpful, we are happy to run it during the rebuttal period.
>
> **Q3: Concerns about data leakage and unclear discussion of the additional information or SFT data.**
>
> A 3.1:  Additional input information (Experiment Compass)
>
> We acknowledge that the data-flow description could be clearer. The **only task-level input used by the main Reasoning-BO pipeline is the Experiment Compass**, which provides variable definitions, bounds, types, and a short natural-language description. The Compass only provides high-level contextual hints. (**Structure see Appendix F**). These are task-level abstractions and contain no labels or optimization trajectories. To make this explicit, below is the entire Experiment Compass used for 2D Ackley:
>
> ```json
> {
>     "name": "Bayesian Optimization on Anonymous Multimodal Benchmark Function",
>     "constraint": "Parameters must be in [-32.768, 32.768] range.",
>     "parameter_definitions": [
>         {
>             "display_name": "x1",
>             "data_type": "float",
>             "bounds": [
>                 -32.768,
>                 32.768
>             ]
>         },
>         {
>             "display_name": "x2",
>             "data_type": "float",
>             "bounds": [
>                 -32.768,
>                 32.768
>             ]
>         }
>     ],
>     "target": {
>         "name": "Objective Function Value",
>         "direction": "minimize"
>     }
> }
> ```
>
> A 3.2: SFT data
>
> The SFT stage is not part of the core pipeline and is used only for an optional variant (RL-14B). Its sole purpose is to enforce correct JSON formatting so that the BO loop can reliably parse candidate points—**not** to teach the model any task knowledge.
>
> The SFT dataset was generated from a small number of very early pilot runs and contains only format demonstrations (JSON schema demonstrations). The RL-14B model fine-tuned in this way is evaluated only on two synthetic mathematical benchmarks, and we verified that none of the SFT data overlaps with or resembles these tasks.
>
> The primary model used for all headline results (QWQ-Plus) is **not fine-tuned on any task-specific data at all.**
> Therefore, the SFT process does not introduce any data-leakage risk, and all optimization signals still come solely from BO evaluations.

---

> ### Author Response · Authors · 2025-11-20
> **Reply to Reviewer uEhn - Part 2/3**
>
> **Q4: Are there limitations on data types, evaluation budgets, or failed evaluations? Can Reasoning BO perform worse than classical methods?**
>
> A4: Thank you for raising this important question. As discussed in **Appendix H**, Reasoning BO does have several practical limitations.
>
> First, the method inherits the context-window constraints of modern LLMs. Although our single-turn design avoids multi-turn accumulation, very long optimization campaigns may still approach the context limit and degrade reasoning quality.
> Second, the framework relies on strong instruction-following ability. Smaller models (3B–7B) frequently fail to maintain the strict Insight-Object schema or produce valid hypotheses, and even light SFT/RL can impair general instruction adherence. This is why our main experiments rely on 14B/32B models rather than smaller ones.
>
> Failure cases primarily occur when (i) the insight chain becomes extremely long in high-dimensional analytic functions, or (ii) variables contain complex structures (e.g., long SMILES strings), resulting in invalid candidates or partially missing hypotheses. These issues do not prevent eventual convergence, and in real wet-lab optimization they are further mitigated because experiments occur step-by-step rather than through dozens of uninterrupted iterations. Nonetheless, they highlight that the robustness of Reasoning BO remains tied to LLM capability.
>
> Importantly, there are scenarios where Reasoning BO can perform worse than classical BO. As shown in **Figure 7** of the paper, enabling database/notes integration on the mathematical Hartmann function reduces performance relative to both “without notes’’ and vanilla BO. This is expected: in knowledge-sparse synthetic functions, structured notes restrict exploration without providing meaningful priors, slowing progress. This aligns with our main message that reasoning-augmented BO is most beneficial in **knowledge-rich real-world tasks** rather than purely synthetic benchmarks.
>
> Finally, we note that modern frontier models exhibit strong context stability; in our chemistry experiments we did not observe context-window failures. Our practical recommendation is simply to employ the strongest available model, which substantially reduces formatting errors and improves overall robustness.
>
> **Q5: Additional information ablation & Explanation of results on Ackley, Rosenbrock, and Hartmann**
>
> A5: Thank you for the question. As clarified in **Q3**, the main Reasoning-BO pipeline uses **only** the minimal Experiment Compass, which provides variable names, types, and bounds—and no labels, trends, or structural hints. Thus, for Ackley, Rosenbrock, or Hartmann, **there is no additional information to ablate.**
>
> Regarding why the model performs well without extra priors: as discussed in **Reviewer 5 FLYi, Q3**, the strong early-iteration performance on Ackley/Rosenbrock/Levy arises largely from the **intrinsic structure of these functions** and the model’s ability to generate **higher-quality initial hypotheses** (e.g., symmetric or central candidates), which typical random/sobol-init BO does not exploit.
>
> Ackley at $x^* = (0, 0, ..., 0)$
>
> And Rosenbrock/Levy at $x^* = (1, ..., 1)$
>
> Such symmetric and easily characterizable structures make it straightforward for the LLM to propose “central” or “canonical” candidates during early iterations, providing stronger initialization than random sampling.
>
> For example, one insight object from 2D Ackley is:
>
> ```json
> {
>   "comment": "The initial sampling strategy combines boundary exploration to characterize flat outer regions with central sampling to target the global minimum. Corner points assess extreme parameter interactions, while center-edge pairs evaluate the transition from steep valley to flat areas, ensuring a balance between exploration and exploitation.",
>   "hypotheses": [
>     {
>       "strategy": "Corner Exploration",
>       "confidence": "medium",
>       "parameter_sets": [
>         {"x1": -32.768, "x2": -32.768},
>       ]
>     },
>     {
>       "strategy": "Central and Edge Midpoint Sampling",
>       "confidence": "high",
>       "parameter_sets": [
>         {"x1": 0.0, "x2": 0.0},
>         {"x1": 32.768, "x2": 0.0}
>       ]
>     }
>   ]
> }
> ```
> The parameter set $(x_1, x_2) = (0, 0)$ is the exact global minimizer, and the model assigns it high confidence in iteration 1. This illustrates that Reasoning BO can provide strong initialization on structured analytic functions—though this behavior is orthogonal to the 2D vs. 15D convergence-speed question.

---

> ### Author Response · Authors · 2025-11-20
> **Reply to Reviewer uEhn - Part 3/3**
>
> **Q6: Request for more functions, larger-scale testing, and more baselines**
>
> We appreciate the reviewer’s suggestion. Our primary goal in this work is to study Reasoning BO in knowledge-enhanced settings rather than to exhaustively benchmark every function class. Each configuration requires 10 repeated runs, and the overall computational cost is substantial, so we focused on a representative set of functions that allow us to probe the core idea clearly. We fully agree that expanding to broader function families would be valuable, and we would be happy to discuss which specific classes or baselines the reviewer considers most informative, so that we can prioritize them in future experiments.
>
> ---
>
> **Closing Remark:**
>
> We sincerely thank the reviewer for the constructive and encouraging feedback. Due to space limits, several clarifications and additional results are provided in the corresponding appendices, and we kindly invite you to refer to them for fuller context. We hope our responses have addressed your concerns, and we warmly welcome any further suggestions or discussion. Several of your points overlap with feedback from other reviewers, and the corresponding responses may offer additional clarification.

---

> ### Author Response · Authors · 2025-11-28
> **Hoping to continue the discussion**
>
> Dear Reviewer,
>
> Thank you for your insightful comments and suggestions. We have submitted responses addressing the points you raised, including clarifications and revisions to improve the presentation and completeness of the work. Should you have any further questions or require additional clarification, please do not hesitate to ask.
>
> We also note that our responses to the other reviewers have been positively received and have further solidified their support for our work. You may find their specific points and our detailed replies informative.
>
> We look forward to your updated review and remain open to further discussion. Thank you very much.

---

### Official Review · Reviewer_RRPq · 2025-11-01

**Soundness:** 2
**Presentation:** 2
**Contribution:** 2
**Rating:** 2
**Confidence:** 3

**Summary:**

This paper proposes a method for integrating knowledge into Bayesian optimization.
The main idea is to use a large language model (with reasoning capabilities after post-training) to (a) generate initialization and (b) score the generated candidates by the acquisition function and pick the most promising candidates.

**Strengths:**

. The empirical results on three biology datasets seems to be very competitive, which shows that BO successfully integrates the knowledge in LLMs.

**Weaknesses:**

1. The method description needs to improve.
The main issue is that the description of the proposed method, particularly Algorithm 1, is extremely vague and confusing due to the lack of proper definitions.
    - What are compass \\(C\\), overview \\(O\\), insights \\(I_t\\), and reasoning data \\(D_t\\)?
    No context is provided.
    There is no way to understand this part of the main paper without looking at the examples in Appendix C.
    Even so, the examples in the appendix are excessive long to fully grasp.
    - It is also not clear what is exactly stored in the database and/or knowledge graph.
    Again, it is largely due to the lack of proper definitions.
    What data get stored in the database, and what data get stored in the knowledge graph?

1. Also, after skimming through the example prompts in the appendix, I feel like the proposed method (with so many complicated components and prompts) is specifically designed for the biology tasks.
It is not entire clear if the methodology in this paper is general enough to be transferred to other domains.
(I don't see example prompts and example LLM outputs for the synthetic benchmarks used in the experiments.)
I am not surprised that the strongest empirical results are obtained on three biology datasets (Table 1), which the empirical results on other benchmarks are a bit mixed.
Also, using LLMs on these biology datasets probably makes more sense because biology knowledge is rich during LLM pretraining.

1. Plots in the experiments do not entirely make sense.
    - E.g., the caption in Figure 3 says the y-axis is the best observed objective value.
    Thus, the curves are supposed to be monotonic.
    Why do the curves oscillate so much?
    - Why does reasoning BO converge much faster on 15D Ackley than 2D Ackley?
    For 2D Ackley, reasoning BO takes around 20 iterations to first reach the global optimum; see Figure 3.
    However, reasoning BO finds the global optimum in fewer than 10 iterations on 15D Ackley; see Figure 5.

**Questions:**

1. In each BO iteration, the acquisition function generates \\(5\\) candidates, and then the LLM scores the candidates to pick the best \\(3\\).
I find these numbers very arbitrary.
It would be good to include some ablation studies on this, e.g., change the number of candidate proposals.

---

> ### Author Response · Authors · 2025-11-20
> **Reply to Reviewer RRPq - Part 1/4**
>
> **Q1: Method description**
>
> A1: Thank you for pointing this out. Some key definitions were placed in the appendix due to space limitations, which may make Algorithm 1 less self-contained than intended. In the revision, we will integrate the necessary definitions directly into the main text.
>
> To clarify here: both the vector database and knowledge graph operate at the task level, they serve distinct purposes. The vector database stores verified reusable pieces of information distilled during the optimization process. The knowledge graph, in contrast, stores structured triples such as ⟨entity, relation, entity⟩ describing task-specific priors or mechanistic relationships. In chemistry tasks, for example, the graph captures relations like “reagent–solvent compatibility” which are essential for guiding reasoning. Beyond enabling autonomous optimization, these representations are intentionally designed to assist human scientists and to keep experts meaningfully involved in the loop, which aligns with the original motivation behind Reasoning-BO.
>
> We are currently running the additional ablations and preparing the updated manuscript, and we will revise Algorithm 1 to make all components self-contained and easier to follow.
>
> **Q2: Is the method biology-specific or general?**
>
> A2: Thank you for the question. We would like to clarify that **Reasoning BO is not biology-specific.**
>
> The framework is designed to be **domain-agnostic**,  all core interfaces--experiment compass, overview, insights, memory modules--operate purely at the task level. In principle, any optimization problem that classical BO can handle can also be instantiated under the Reasoning BO framework. Biology appears more prominently in the paper simply because biological and chemical optimization problems are highly knowledge-rich, making them natural testbeds for reasoning-augmented BO.
> In practice, the only task-dependent component is the Experiment Compass,  which defines the search space and variable domains. For example, the 2D Ackley optimization used in our experiments is defined by a simple Compass such as:
>
> ```json
> {
>     "name": "Bayesian Optimization on Anonymous Multimodal Benchmark Function",
>     "constraint": "Parameters must be in [-32.768, 32.768] range.",
>     "parameter_definitions": [
>         {
>             "display_name": "x1",
>             "data_type": "float",
>             "bounds": [
>                 -32.768,
>                 32.768
>             ]
>         },
>         {
>             "display_name": "x2",
>             "data_type": "float",
>             "bounds": [
>                 -32.768,
>                 32.768
>             ]
>         }
>     ],
>     "target": {
>         "name": "Objective Function Value",
>         "direction": "minimize"
>     }
> }
> ```
>
> Once this Compass is provided, the **entire pipeline runs identically across domains**. This is why the method transfers without modification to mathematical functions, classical control tasks (e.g., Lunar Lander), and chemical reaction optimization. Importantly, the high-level “overview” texts in non-biology tasks were generated autonomously by the LLM itself, not manually authored, further supporting that the method is not tailored to biological applications.
>
> Finally, the codebase has already been released publicly, and we are preparing a more cleaned and unified version to make it easier for external users to apply Reasoning BO to arbitrary BO tasks.

---

> ### Author Response · Authors · 2025-11-20
> **Reply to Reviewer RRPq - Part 2/4**
>
> **Q3: Clarification of Figure 3, curve oscillation, and the 2D vs 15D Ackley behavior**
>
> A 3.1: Clarification of Figure 3, curve oscillation
>
> Thank you for pointing out this inconsistency. The curves in Figure 3 are **not** best-so-far trajectories. Instead, the y-axis plots the **per-iteration best objective among the three points actually evaluated in that iteration**, selected by the LLM from five BO-proposed candidates. Because the plotted metric is per-iteration rather than cumulative, the curve is not expected to be monotonic.
>
> Each experiment is repeated 10 times with fixed seeds, and the reported curve is the average across these runs. Two factors cause observed oscillation:
>
> 1. **Run-to-run initialization variance.**
>   Even with fixed seeds, the 10 independent runs start in slightly different regions of the search space; averaging their per-iteration values produces mild up-and-down fluctuations—a well-known effect in BO trajectory averaging.
> 2. **Occasional LLM formatting/output errors.**
>   In a small number of iterations, the LLM produces fewer than three valid candidates (e.g., due to malformed JSON in high-dimensional analytic tasks or long SMILES strings in chemistry tasks). When this happens, the per-iteration maximum is temporarily lower, which introduces small dips in the averaged trajectory.
>
> These fluctuations occur only locally and do **not affect convergence**, as the surrogate model remains the main driver of long-term optimization. The choice to use per-iteration metrics is deliberate: it exposes the dynamics of LLM-guided decisions rather than hiding them behind a monotonic best-so-far curve.

---

> ### Author Response · Authors · 2025-11-20
> **Reply to Reviewer RRPq - Part 3/4**
>
> A 3.2: Why Reasoning BO converges faster on 15D Ackley than on 2D Ackley?
>
> We appreciate the reviewer for highlighting this phenomenon. After examining the per-run trajectories, we find that the difference between the 2D and 15D curves arises from standard BO dynamics rather than from any advantage specific to high dimensionality. The tables below summarize representative runs, where task₁, task₂, … denote independent runs under fixed seeds, and trial_index denotes optimization iterations.
>
> 1. **Run-to-run initialization variance (main contributing factor)**
>
> As discussed in **A3.1**, the averaged trajectory is sensitive to heterogeneous initialization across repeated runs.
> In the 2D setting, one seed (task₆) begins in a particularly unfavorable region(Table B), which noticeably delays the rise of the **average** 2D curve. In contrast, the 15D runs do not contain an outlier of similar severity, so their averaged curve appears to improve earlier.
>
> Thus, the 15D curve rises earlier not because high-dimensional Ackley is easier, but because the 2D average is disproportionately pulled down by a single poor run. Importantly, when viewed individually, both 2D and 15D runs ultimately converge to similarly low objective values.
>
> ---
>
> Table A: Per-iteration objective values for Ackley-15D (7 independent runs)
>
> | trial_index | task_1 | task_2 | task_3 | task_4 | task_5 | task_6 | task_7 |
> |-------------|--------|--------|--------|--------|--------|--------|--------|
> | 0 | 0.004053256 | 0.32787485 | 0.868608996 | 0.868608996 | 0.868608996 | 3.625384938 | 2.987913583 |
> | 1 | 0.000400533 | 0.045318646 | 0.045318646 | 0.045318646 | 0.045318646 | 0.868608996 | 1.192516454 |
> | 2 | 0.000400533 | 0.045318646 | 0.01068286 | 0.01068286 | 0.004053256 | 0.13740366 | 0.168032988 |
> | 3 | 1.03E-05 | 0.004053256 | 0.001036346 | 0.004053256 | 0.000400533 | 0.13740366 | 0.019610922 |
> | 4 | 1.03E-06 | 0.000400533 | 0.001036346 | 0.000517285 | 0.000400533 | 0.01068286 | 0.003914072 |
> | 5 | 1.03E-08 | 0.000400533 | 0.000103315 | 0.000400533 | 4.00E-05 | 0.01068286 | 0.000103315 |
> | 6 | 1.03E-12 | 0.000200133 | 1.03E-05 | 0.000103315 | 4.00E-06 | 0.001036346 | 1.03E-05 |
> | 7 | 0 | 0.000103315 | 0.000103315 | 4.00E-06 | 4.00E-07 | 0.000103315 | 0 |
> | 8 | 0 | 0.000100033 | 1.03E-05 | 4.00E-06 | 4.00E-08 | 1.03E-05 | 0 |
> | 9 | 0 | 1.03E-05 | 1.03E-05 | 4.00E-06 | 4.00E-09 | 1.03E-05 | 0 |
>
> ---
>
> Table B: Per-iteration objective values for Ackley-2D (7 independent runs)
>
> | trial_index | task_1 | task_2 | task_3 | task_4 | task_5 | task_6 | task_7 |
> |-------------|--------|--------|--------|--------|--------|--------|--------|
> | 0 | 0.328842206 | 0.030944912 | 0.328842206 | 0 | 0 | **2.637531092** | 0.528416676 |
> | 1 | 0.030944912 | 0.002855055 | 0.136290485 | 0 | 0 | **2.637531092** | 0.030944912 |
> | 2 | 0 | 0 | 0.076511875 | 0.528416676 | 2.637531092 | **2.637531092** | 0.528416676 |
> | 3 | 0 | 0 | 0.045318646 | 0.001420871 | 0.004053256 | **2.637531092** | 0.002855055 |
> | 4 | 0.001420871 | 0.000283109 | 0.014807708 | 0.001420871 | 0.004053256 | **6.593599079** | 0.000283109 |
> | 5 | 0.000141488 | 2.83E-05 | 0.012479254 | 0.000283109 | 0.000400533 | **6.914978163** | 2.637531092 |
> | 6 | 2.83E-05 | 2.83E-05 | 0.002013314 | 2.83E-05 | 0.000400533 | **4.253654027** | 2.83E-05 |
> | 7 | 2.83E-08 | 2.83E-05 | 0.001204793 | 2.83E-07 | 4.00E-05 | **0.206636279** | 2.83E-05 |
> | 8 | 2.83E-09 | 2.83E-06 | 0.000200133 | 2.83E-08 | 0.000283109 | **0.002855055** | 2.83E-06 |
> | 9 | 2.83E-10 | 2.83E-06 | 0 | 4.00E-08 | 0.000141488 | **0.017397574** | 2.83E-07 |
>
> ---
>
> **2. High-dimensional BO often becomes more conservative[1, 2, 3]**
>
> In higher dimensions, broad exploration becomes disproportionately expensive, causing acquisition functions (such as qLogEI in our paper) to adopt more conservative behavior. Once the 15D surrogate identifies a reasonably good basin, it tends to exploit it earlier, while 2D BO continues to explore the landscape more broadly before committing.
> This results in faster improvement in the **averaged** 15D curve. We emphasize that this is a standard BO effect—
> not a special advantage of Reasoning BO, and **not** related to leakage.

---

> ### Author Response · Authors · 2025-11-20
> **Reply to Reviewer RRPq - Part 4/4**
>
> **Ackley’s symmetric structure makes good initialization easy for both 2D and 15D**
>
> This point does not explain the 2D–15D difference, but helps clarify why Reasoning BO performs well on Ackley overall. Ackley's global minimum is located at: $x* = (0, 0, 0..., 0)$.  A highly symmetric and easily “guessable” structure. In fact, inspection of our logs shows that the LLM frequently proposes central candidates already in iteration 1. For example, one `insight` object from 2D Ackley is:
>
> ```json
> {
>   "comment": "The initial sampling strategy combines boundary exploration to characterize flat outer regions with central sampling to target the global minimum. Corner points assess extreme parameter interactions, while center-edge pairs evaluate the transition from steep valley to flat areas, ensuring a balance between exploration and exploitation.",
>   "keywords": "boundary exploration, central valley sampling, edge midpoint, global minimum targeting",
>   "hypotheses": [
>     {
>       "strategy": "Corner Exploration",
>       "rationale": "Sample extreme parameter values at the domain corners to explore the flat outer regions and boundary behavior.",
>       "confidence": "medium",
>       "parameter_sets": [
>         {"x1": -32.768, "x2": -32.768},
>         {"x1": 32.768, "x2": 32.768}
>       ]
>     },
>     {
>       "strategy": "Central and Edge Midpoint Sampling",
>       "rationale": "Directly probe the central valley where the global minimum is located and assess the transition to the flat region via an edge midpoint.",
>       "confidence": "high",
>       "parameter_sets": [
>         {"x1": 0.0, "x2": 0.0},
>         {"x1": 32.768, "x2": 0.0}
>       ]
>     }
>   ]
> }
> ```
>
> The parameter set $(x_1, x_2) = (0, 0)$ is the exact global minimizer, and the model assigns it high confidence in iteration 1. This illustrates that Reasoning BO can provide strong initialization on structured analytic functions—though this behavior is orthogonal to the 2D vs. 15D convergence-speed question.
>
> [1]Snoek et al., “Practical Bayesian Optimization of Machine Learning Algorithms” NIPS 2012
>
> [2]Wang et al., “Bayesian Optimization in High Dimensions via Additive Models” ICML 2013.
>
> [3]Eriksson et al., “Scalable Global Optimization via Local Bayesian Optimization (TuRBO)” NeurIPS 2019
>
> **Q4: Hyperparameter choice and candidate number ablation**
>
> A4: The choice of how many BO-proposed candidates to pass to the LLM is fundamentally a trade-off between diversity and computational cost. Our initial setup followed the design of BORA [4], where the BO backend generates five candidates in each iteration and a higher-level policy selects a subset for actual evaluation. As noted in their paper, “setting $n_{\mathrm{BO}} = 5$ and  $n_{\mathrm{LBO}} $ to a small number empirically showed to offer enough diversity of hypothesized optima locations while maintaining competitive performance.”
>
> We began with the same configuration, but unlike BORA, which uses a hand-crafted heuristic policy and allows LLMs to directly propose new points, our framework deliberately restricts the LLM to a selection-only role. Through early experimentation, we found that evaluating three LLM-selected points provided a favorable compromise: it accelerates practical convergence while avoiding noticeable loss in accuracy, and it substantially reduces the LLM inference overhead compared to larger candidate sets.
> During the rebuttal period, we are running a lightweight ablation on the candidate number and measuring both performance and LLM inference budget. We will include the results and a short discussion of this trade-off in the revision.
>
> [4]: Cissé, A., Evangelopoulos, X., Gusev, V. V., & Cooper, A. I. (2025). Language-Based Bayesian Optimization Research Assistant (BORA). arXiv:2501.16224.
>
> ---
>
> **Closing remark**
>
> We sincerely thank the reviewer for the constructive and encouraging feedback. Due to space limits, several clarifications and additional quantitative results are provided in the corresponding appendices, and we kindly invite you to refer to them for fuller context. We hope our responses have addressed your concerns, and we warmly welcome any further suggestions or discussion. Several of your points overlap with feedback from other reviewers, and the corresponding responses may offer additional clarification.

---

> ### Author Response · Authors · 2025-11-28
> **Hoping to continue the discussion**
>
> Dear Reviewer,
>
> Thank you for your insightful comments and suggestions. We have submitted responses addressing the points you raised, including clarifications and revisions to improve the presentation and completeness of the work. Should you have any further questions or require additional clarification, please do not hesitate to ask.
>
> We also note that our responses to the other reviewers have been positively received and have further solidified their support for our work. You may find their specific points and our detailed replies informative.
>
> We look forward to your updated review and remain open to further discussion. Thank you very much.

---

### Author Response · Authors · 2025-11-22
**Looking Forward to Your Updated Review**

Dear Reviewers,

Thank you again for the thoughtful and constructive feedback. We truly appreciate your time and insights.

In response, we have updated the manuscript to improve clarity and self-containment. Specifically:

- Section 2.2 and Algorithm 1 now include full definitions and mathematical notation.
- The distinction between QWQ-Plus and the RL-14B variant—as well as the roles of SFT/GRPO and knowledge modules—is now clearly stated.
- Figures have been enlarged for readability, and Related Work has been moved to the appendix.
- We also emphasize that limitations and practical constraints are documented in the Appendix I, as some reviewers noted these were easy to miss.

We hope the rebuttal and updated manuscript address the concerns raised. We look forward to your updated review and remain open to further discussions.

Sincerely,

The Authors

---

### Author Response · Authors · 2025-12-01
**Thank All Reviewers**

To all reviewers,

We would like to sincerely thank all reviewers for their thoughtful insights and valuable comments. These discussions have significantly helped us improve the quality and clarity of our manuscript.

It seems that the vacancy of a summarization of the core contributions and the positioning of this framework have caused some misunderstandings. To ensure a clear common ground for discussion, we summarize our **core contributions** here, which have been highlighted in the revised paper:

1. **Reasoning-Integrated BO Framework**: We propose Reasoning BO, a novel framework that embeds a reasoning agent into the Bayesian Optimization loop. Instead of merely generating points, the agent acts as a reasoning-driven acquisition critic, generating and evolving scientific hypotheses to guide the sampling process;
2. **Dynamic Knowledge Management**: We design a multi-agent system (Verifier, Formatter, Notes Agent) coupled with a dual-channel memory (Knowledge Graph & Vector DB). This enables the system to continuously accumulate domain expertise and real-time insights, addressing the "memoryless" limitation of traditional BO;
3. **Superior Performance on Complex Tasks**: We demonstrate that our method significantly outperforms traditional baselines on knowledge-rich tasks. For instance, in the Direct Arylation reaction, Reasoning BO achieved a 60.7% yield compared to 25.2% for traditional BO, highlighting the power of long-context reasoning in optimization.

We are excited that you recognized our contributions. We quote correspondingly as below:

- **On Novelty & Concept**: "This paper introduces a novel idea: using an agentic reasoning model to augment Bayesian optimization... conceptually exciting, and timely." [Reviewer FLYi]; "The underlying concept is interesting and opens up many promising directions for future research." [Reviewer uEhn]; "AFAIK, this is the first work to explicitly investigate whether long-form CoT reasoning can improve BO performance." [Reviewer ZUuJ].
- **On Performance & Results**: "The empirical results on three biology datasets seems to be very competitive, which shows that BO successfully integrates the knowledge in LLMs." [Reviewer RRPq]; "The methodology shows improvement beyond traditional baselines on a variety of benchmarks... capable of high-quality optimization." [Reviewer 6mjx]; "The results consistently favor the proposed method." [Reviewer FLYi].
- **On Generalizability & Structure**: "The proposed framework is flexible and general enough to be applied across diverse optimization domains... beneficial as this is a weakness of similar methods." [Reviewer uEhn]; "The method is adaptable across many different optimization tasks and appears to have a robust structure." [Reviewer 6mjx].

We also appreciate the constructive suggestions, based on which we have significantly improved our manuscript. The major changes during the rebuttal include:

- **New Generalization Benchmarks (BBOB)**: To address concerns about domain specificity and potential leakage, we added comprehensive experiments on the BBOB benchmark (30 iterations, 5D/10D). Reasoning BO achieved an average improvement of +48.5% over Classical BO and +64.7% over CMA-ES, proving its robustness on purely mathematical functions [Response to Reviewer uEhn & 6mjx].
- **New SOTA Baselines (TuRBO & BO-LS)**: We integrated comparisons with **TuRBO** and **BO-LS** on the 15D Ackley task, demonstrating that Reasoning BO maintains superior performance even against specialized high-dimensional optimizers [Response to Reviewer FLYi & 6mjx].
- **Clarified Methodology**: We completely restructured Section 2 & 3, moving key definitions (Experiment Compass, Insights) to the main text and explicitly distinguishing the primary model (**QwQ-Plus**) from the ablation-only RL variant [Response to Reviewer RRPq & ZUuJ].

We are grateful for the positive reception of our updates. As **Reviewer FLYi** noted in their final comment:
> "Thank you for adding a comparison to TuRBO and BO_LS. This result significantly strengthens the paper...
> Overall, this rebuttal addresses all of my concerns, and I continue to think that this paper should be accepted."

We hope that these extensive new experiments and revisions adequately address the concerns of all reviewers. We remain active in the discussion period and are happy to clarify any further details.

Sincerely,

Authors

---

### Meta-Review · Area_Chair_1Vf6 · 2026-01-02

**Summary:**

This paper proposes a method for combining standard Bayesian optimization with reasoning LLMs. Specifically, a standard GP surrogate model and a batch acquisition function (e.g., qLogEI) are used to propose a set of high-utility candidates, and then a reasoning LLM, supplied with additional knowledge/context, is used to score amongst them.

While the idea is a promising new direction for Bayesian optimization, there are several limitations with the paper in its current form:

* Clarity: The notation and description of the algorithm are hard to follow, and the method was only made clear after a back-and-forth conversation with the authors
* Lack of insight on standard benchmark problems: the method performs well on standard synthetic benchmarks, like Ackley, Rosenbrock, and Hartmann, where there is little additional knowledge that could be supplied in an LLM. While the authors provide some hypotheses and anecdotal evidence for this performance, these results call for a more thorough evaluation and a set of ablations.
* Risk for contamination. While the authors do not provide the names of the synthetic test functions to the LLM, auxiliary information about the tasks may still cause some leakage. For example, the bounds of $\pm 32.728$, which are supplied to the LLM, are well-known to be associated with the Ackley function.
* More additional benchmark problems: the authors tested on two “real-world” benchmarks and several synthetic ones. This paper would be strengthened by more real-world benchmarks.

None of these issues is fatal, and I can see this work providing value to a future conference. However, I believe a major revision and another round of reviewing are necessary before it merits publication at a top-tier venue.

**Reviewer Concerns:**

Many of the reviewers were concerned about clarity, scope of problems, leakage, context supplied to the LLM, and baselines/benchmarks. The authors clarified confusion about the notation/algorithm and the context supplied to the LLM, and addressed many (though not all) concerns about leakage. Moreover, they included two new baseline methods during the rebuttal phase. Most of the reviewers did not respond to the rebuttal, which is unfortunate. I believe many of the issues were resolved; however, it is hard to assess the extent to which the clarity concerns have been addressed. Moreover, I think that - even with more reviewer discussion - there would still be concerns about the benchmark problems and possible leakage.

**Reviewer Scores:**

I believe some reviewers may have increased their scores based on how the authors addressed clarity concerns, algorithmic details, and contamination issues. Nevertheless, given the paper's initial low scores and the breadth of concerns, I feel confident in recommending another major round of revision.

- Reviewers RRPq, uEhn, ZUuJ, and tmjx did not engage at all during the discussion period, and so my guess is that they would not have likely changed their score with more discussion.
- FLYi engaged substantially during the discussion period, and concluded that their concerns were addressed. They already increased their score, so more discussion probably wouldn’t have changed the outcome.

---

### Decision · Program_Chairs · 2026-01-26

Reject